# LoDAdaC: a unified local training-based decentralized framework with adaptive gradients and compressed communication

**Wei Liu**                                                                          *lwdsdqqb@gmail.com*
*Department of Mathematical Sciences*
*Rensselaer Polytechnic Institute*

**Anweshit Panda**                                                                  *pandaa2@rpi.edu*
*Department of Computer Science*
*Rensselaer Polytechnic Institute*

**Ujwal Pandey**                                                                    *pandeu@rpi.edu*
*Department of Computer Science*
*Rensselaer Polytechnic Institute*

**Haven Cook**                                                                      *cookh2@rpi.edu*
*Department of Computer Science*
*Rensselaer Polytechnic Institute*

**George M. Slota**                                                                 *slotag@rpi.edu*
*Department of Computer Science*
*Rensselaer Polytechnic Institute*

**Naigang Wang**                                                                    *nwang@us.ibm.com*
*IBM T. J. Watson Research Center*

**Jie Chen**                                                                        *chen.future.jie@gmail.com*
*MIT-IBM Watson AI Lab, IBM Research*

**Yangyang Xu***                                                                    *xuy21@rpi.edu*
*Department of Mathematical Sciences*
*Rensselaer Polytechnic Institute*

**Reviewed on OpenReview:** *https://openreview.net/forum?id=0qoy9usvnm*

## Abstract

In the decentralized distributed learning, achieving fast convergence and low communication cost is essential for scalability and high efficiency. Adaptive gradient methods, such as Adam, have demonstrated strong practical performance in deep learning and centralized distributed settings. However, their convergence properties remain largely unexplored in decentralized settings involving multiple local training steps, such as federated learning. To address this limitation, we propose LoDAdaC, a unified multiple **Lo**cal Training (MLT) **D**ecentralized framework with **Ada**m-type updates and **C**ompressed communication (CC). LoDAdaC accommodates a broad class of optimizers for its local adaptive updates, including AMSGrad, Adam, and AdaGrad; it is compatible with standard (possibly biased) compressors such as low-bit quantization and sparsification. MLT and CC enable LoDAdaC to achieve multiplied reduction of communication cost, while the technique of adaptive updates enables fast convergence. We rigorously prove the combined advantage through complexity analysis. In

---

*Corresponding author

addition, experiments on image classification and GPT-style language model training validate our theoretical findings and show that LoDAdaC significantly outperforms existing decentralized algorithms in terms of convergence speed and communication efficiency.

# 1 Introduction

In decentralized learning, multiple agents collaboratively train a model without a central server, by exchanging information exclusively with immediate (a.k.a. one-hop) neighbors. Compared to centralized distributed learning, decentralized learning has better robustness and scalability. However, communication cost can become a bottleneck in decentralized learning, especially when low-bandwidth or wireless communication is performed. This motivates the design of communication-efficient decentralized algorithms.

Two widely adopted strategies to reduce the communication burden in distributed learning are *Compressed Communication (CC)* and *Multiple Local Training (MLT)*. By CC, the agents transmit compressed information rather than full-precision one, significantly reducing per-round communication cost. Examples include low-bit quantization (Sun et al., 2020; Wang et al., 2018; Bernstein et al., 2018; Alistarh et al., 2017) and sparsification (Koloskova et al., 2019; Stich et al., 2018). CC also provides implicit privacy protection: by transmitting the compressed message, agents inherently obscure precise local data information, thus mitigating potential privacy risks (Kairouz et al., 2021). On the other hand, MLT, which involves performing several local updates per communication round, has gained popularity in various distributed learning settings. Prominent examples include local SGD (Haddadpour et al., 2019), SGD averaging (Zhang et al., 2015), and, notably, Federated Averaging (FedAvg) (McMahan et al., 2017), a widely employed method in federated learning (FL). Empirically, MLT significantly reduces the number of communication rounds required to achieve a target convergence threshold. Moreover, from a privacy perspective, MLT further enhances security by reducing the frequency and amount of sensitive information exchanged among agents, thus limiting potential data leakage (Li et al., 2020; Kairouz et al., 2021).

Both CC and MLT have been explored in vanilla and momentum SGD (Singh et al., 2021; Sun et al., 2022), and it is shown in (Singh et al., 2021) that multiplied reduction of communication can be achieved. However, adaptive (i.e., Adam) stochastic methods (Kingma & Ba, 2014) exhibit significantly faster convergence than vanilla or momentum SGD on training deep learning models and are now the workhorse for training language models. Hence, it is natural to ask the following question:

> *Can* CC *and* MLT *be applied in* decentralized adaptive stochastic *methods to simultaneously achieve multiplied reduction of communication and fast convergence?* (Q)

## 1.1 Contributions

This work provides an affirmative answer to the question (Q). We propose LoDAdaC, a **Lo**cal training-based **D**ecentralized framework with **Ada**ptive gradient updates and **C**ompressed communication. Our local update scheme includes both the vector and matrix variants of AdaGrad (Duchi et al., 2011), Adam (Kingma & Ba, 2014), AMSGrad (Reddi et al., 2016), and the recently proposed Adam-Mini (Zhang et al., 2024). The integration of CC and MLT enables a multiplied reduction of communication cost while adaptive gradient updates further yield fast convergence.

A central technical contribution of our work lies in resolving the core analytical challenge introduced by *the interaction of MLT, CC, adaptive gradient updates, and decentralized communication*: their coupling makes it difficult to derive a unified upper bound on the consensus error and stationarity violation. In particular, local adaptive gradient updates introduce nonlinearity and dynamically varying gradient scaling, which complicate the analysis even in centralized settings (Wang et al., 2022b). When combined with decentralized model aggregation, these properties pose significant technical obstacles to convergence analysis. Our analysis carefully disentangles the coupling, leading to tight convergence guarantees under mild conditions and offering the first such results for this challenging setting. By performing $K$ local updates per communication round and utilizing a compression operator that compresses one unit of message to $1 - \eta$ unit with $\eta \in (0, 1)$, our algorithm needs a total communication cost of $O(\frac{1-\eta}{K\epsilon^4})$ to produce an $\epsilon$-stationary solution, thus yielding

multiplied reduction of communication cost. Notably, the result applies uniformly across different choices of compressors and adaptive updates.

In addition, we conduct numerical experiments on two representative tasks, i.e., image classification and language model training, to validate the effectiveness of LoDAdaC. Though our complexity result has the same order dependence on $\epsilon$ as achieved by existing non-adaptive stochastic methods such as SQuARM-SGD, our numerical results demonstrate that LoDAdaC achieves significant speed up by adaptive gradient updates and significant communication reduction from combining MLT and CC. Specifically, our experiments illustrate that: (i) LoDAdaC equipped with adaptive gradient updates significantly outperforms the baseline decentralized algorithm SQuARM-SGD (that employs momentum gradient update) in terms of convergence speed; (ii) The joint use of MLT and CC reduces the total communication cost dramatically, achieving reductions of over 99% in some scenarios (e.g., see the results yielded by LoDAdaC with $K = 50$ and Top-$k$=30% in Figures 3a and 3b), with nearly no sacrifice of accuracy. These empirical findings align closely with our theoretical results.

## 1.2 Problem formulation and technical assumptions

We consider decentralized nonconvex stochastic optimization in the form of

$$\min_{\boldsymbol{x}\in\mathbb{R}^d} f(\boldsymbol{x}) := \frac{1}{n}\sum_{i=1}^n f_i(\boldsymbol{x}), \text{ with } f_i(\boldsymbol{x}) = \mathbb{E}_{\xi_i\sim\mathcal{D}_i}\left[F_i\left(\boldsymbol{x},\xi_i\right)\right]. \tag{1}$$

Here, $n$ agents, connected via a communication graph $\mathcal{G}$, collectively minimize the objective function $f$ as the average of local functions $\{f_i\}$, each of which is defined as an expectation over a data distribution $\mathcal{D}_i$. Each agent $i \in \{1, 2, \ldots, n\}$ exclusively accesses its local function $f_i$ and stochastic gradients $\nabla F_i(\boldsymbol{x}, \xi_i)$, and collaboration occurs through communication with immediate neighbors.

To perform decentralized computation, each agent $i \in \{1, 2, \ldots, n\}$ maintains a local copy $\boldsymbol{x}_i$ of the decision variable $\boldsymbol{x}$. Let $\mathbf{X} = [\boldsymbol{x}_1, \boldsymbol{x}_2, \ldots, \boldsymbol{x}_n] \in \mathbb{R}^{d\times n}$. The problem (1) can be equivalently reformulated as

$$\min_{\mathbf{X}\in\mathbb{R}^{d\times n}} \frac{1}{n}\sum_{i=1}^n f_i(\boldsymbol{x}_i), \text{ s.t. } \mathbf{X} = \mathbf{XW}, \tag{2}$$

where $\mathbf{W}$ is a mixing (a.k.a. gossip) matrix that governs how agents aggregate local information. Under Assumption 2(iii) given below, imposing the constraint $\mathbf{X} = \mathbf{XW}$ is equivalent to requiring $\boldsymbol{x}_1 = \cdots = \boldsymbol{x}_n$, i.e., $\mathbf{X}$ lies in the consensus subspace.

Throughout the paper, we make the following standard assumptions.

**Assumption 1** *For each $i \in \{1, 2, \ldots, n\}$, the function $f_i$ is L-smooth, i.e., $\|\nabla f_i(\boldsymbol{x}) - \nabla f_i(\boldsymbol{y})\| \leq L\|\boldsymbol{x} - \boldsymbol{y}\|$, for any $\boldsymbol{x}, \boldsymbol{y} \in \mathbb{R}^d$, and $f$ is lower bounded, i.e., $f^* := \min_{\boldsymbol{x}} f(\boldsymbol{x}) > -\infty$.*

**Assumption 2** *For the mixing matrix $\mathbf{W}$, it holds (i) $\mathbf{W}$ is doubly stochastic, i.e., $\mathbf{W} \geq 0$, $\mathbf{W}\mathbf{1} = \mathbf{1}$ and $\mathbf{1}^\top\mathbf{W} = \mathbf{1}^\top$; (ii) $W_{ij} = 0$ if $i$ and $j$ are not neighbors to each other; (iii) $\mathrm{Null}(\mathbf{W} - \mathbf{I}) = \mathrm{span}\{\mathbf{1}\}$ and $\rho := \|\mathbf{W} - \mathbf{J}\|_2 < 1$, where $\mathbf{1}$ is an all-one vector, $\mathbf{I}$ is the identity matrix, and $\mathbf{J} = \frac{\mathbf{1}\mathbf{1}^\top}{n}$.*

Assumption 2 encodes the standard structural conditions required for decentralized averaging. First, the doubly stochastic property ensures that the mixing operation preserves the network average. Second, the condition $W_{ij} = 0$ for any non-neighboring agents $i$ and $j$ enforces that communication occurs only between neighboring nodes in the underlying graph $\mathcal{G}$, thereby respecting the locality of the decentralized architecture. Most importantly, the spectral condition $\rho = \|\mathbf{W} - \mathbf{J}\|_2 < 1$ guarantees contraction toward consensus and serves as the key quantity controlling the consensus-error recursion in our analysis. Intuitively, smaller values of $\rho$ correspond to faster information propagation across the network and hence more rapid agreement among agents. The specific choice of the mixing matrix $\mathbf{W}$ depends on the communication topology, and several standard constructions have been proposed in the literature (Koloskova et al., 2019; Mancino-Ball et al., 2023; Nedić et al., 2018). In particular, Xiao & Boyd (2004) showed that one can design an optimal mixing matrix that minimizes $\rho$ while satisfying the constraints in Assumption 2.

### 1.3 Notations and definitions

We define $[T] = \{0, 1, \ldots, T-1\}$ and use $\|\cdot\|$ to denote the Euclidean norm for vectors and the Frobenius norm for matrices. The spectral norm of a matrix $\mathbf{A}$ is denoted by $\|\mathbf{A}\|_2$. For two vectors $\boldsymbol{a}$ and $\boldsymbol{b}$ of the same dimension, $\frac{\boldsymbol{a}}{\boldsymbol{b}}$ and $\boldsymbol{a} \circ \boldsymbol{b}$ denote componentwise division and multiplication, respectively, while $\sqrt{\boldsymbol{c}}$ applies the square-root operation elementwise to a nonnegative vector $\boldsymbol{c}$. $\mathbf{X}_\perp = \mathbf{X}(\mathbf{I} - \mathbf{J})$ denotes the consensus error matrix and $\overline{\boldsymbol{x}} = \frac{1}{n}\mathbf{X}\mathbf{1}$ for the average of all local decision variables. $\mathbb{E}_t$ takes the expectation over the random samples $\{\xi_i^t\}_{i \in \{1,2,\ldots,n\}}$ conditional on the $t$-th iterate, while $\mathbb{E}$ takes the full expectation.

**Definition 1.1** *We call $\mathcal{Q}$ an $\eta$-compression operator, if it holds $\mathbb{E}_{\mathcal{Q}}\left[\|\boldsymbol{x} - \mathcal{Q}[\boldsymbol{x}]\|^2\right] \leq \eta^2 \|\boldsymbol{x}\|^2$ for some $\eta \in [0,1)$ and all $\boldsymbol{x} \in \mathbb{R}^d$.*

The expectation in the above definition is taken with respect to the internal randomness of the compressor, conditioned on the input $\boldsymbol{x}$. For deterministic compressors, such as Top-$k$ sparsification, the inequality holds deterministically. Our analysis relies only on the contractive-error property in Definition 1.1 and does not require additional assumptions such as unbiasedness. Examples of $\eta$-compression operators include Random-$k$ (Stich et al., 2018), Top-$k$ (Aji & Heafield, 2017), and the rescaled quantizations (Chen et al., 2023a); see more examples in (Chen et al., 2023a; Koloskova et al., 2019). When $\eta = 0$, $\mathcal{Q}$ simplifies to the identity operator.

**Definition 1.2** *We say that $\mathbf{X}$ is an $\epsilon$-stationary point, in expectation, of the decentralized problem (2) if $\mathbb{E}\left[\|\nabla f(\overline{\boldsymbol{x}})\|^2\right] + \mathbb{E}\left[\frac{1}{n}\|\mathbf{X}_\perp\|^2\right] \leq \epsilon^2$.*

This notion jointly controls stationarity of the averaged model and network disagreement; both must be small for decentralized learning to be practically meaningful.

## 2 Related work

In this section, we review existing works on distributed stochastic gradient methods (SGMs) in either a centralized or a decentralized setting for solving nonconvex problems. Additionally, we review methods developed for distributed learning with MLT and CC.

### 2.1 Centralized or decentralized (stochastic) adaptive gradient methods

Adaptive SGMs are among the most popular stochastic algorithms for training nonconvex deep learning models. In practice, adaptive SGMs such as AdaGrad (Duchi et al., 2011), Adam (Kingma & Ba, 2014), and AMSGrad (Reddi et al., 2016) are more effective compared to a nonadaptive SGM.

Efforts have been made to integrate adaptive gradient updates into distributed optimization. Hou et al. (2018) propose a distributed Adam for convex problems, while (Chen et al., 2020; Zhao et al., 2022) introduce locally adaptive algorithms for centralized distributed training. The compressed centralized distributed Adam variants are explored in (Chen et al., 2021; 2023a). A centralized distributed AMSGrad is studied in (Li et al., 2022), and a compressed version is presented in (Wang et al., 2022a). The decentralized Adam variant, DADAM, was introduced in (Nazari et al., 2022), providing convergence results for both convex and nonconvex problems. However, subsequent analysis by (Chen et al., 2023b) reveals that DADAM may not converge to a stationary point in nonconvex settings. To address this limitation, (Chen et al., 2023b; Wang et al., 2025; Liu et al., 2025) propose some other decentralized adaptive gradient methods.

Despite these advancements, distributed learning with multiple local adaptive gradient updates has been explored only in a centralized setting. Xie et al. (2019) propose AdaAlter, which employs local adaptive updates on the client side. Similarly, Reddi et al. (2020) extend FedAvg by incorporating three types of local adaptive gradient updates to improve optimization performance. More recently, FedLADA (Sun et al., 2023) introduces momentum-corrected adaptive updates, and FedAMS, along with its corrected variant FedCAMS (Wang et al., 2022b), stabilizes local AMSGrad updates to ensure convergence. However, decentralized distributed learning with multiple local *adaptive* gradient updates remains unexplored. Though Gao

& Huang (2020) attempt to study a decentralized distributed method with multiple local Adam updates, they conduct analysis only to the case without first-order momentum. In addition, their convergence rate results in Corollary 1 and Corollary 2 are obtained by implicitly assuming $\beta_2 = 0$, i.e., no second-order momentum either.

## 2.2 MLT in distributed learning

MLT is a simple yet remarkably effective communication-saving strategy in distributed learning, where clients perform several local updates—rather than a single one—between successive communication rounds. A foundational method that employs MLT in centralized distributed learning is FedAvg, with numerous extensions including FedAvg with local momentum (Hsu et al., 2019), server momentum (Sun et al., 2024), and adaptive FedAvg (Reddi et al., 2020). Recent theoretical advancements have clarified why MLT effectively reduces communication complexity in centralized distributed learning (Kairouz et al., 2021; Li et al., 2020). These results have been rigorously established across a wide range of local update strategies, including standard SGD (Haddadpour et al., 2019; Spiridonoff et al., 2021; Stich, 2018; Yu et al., 2019), momentum-based methods (Karimireddy et al., 2020; Sun et al., 2024), and adaptive gradient methods (Reddi et al., 2020; Xie et al., 2019).

In decentralized distributed learning, early work has primarily focused on algorithms using simple local SGD updates. For example, Xing et al. (2020) propose a decentralized federated learning framework for medical applications, operating without a central server in a dynamic peer-to-peer network. Similarly, Lalitha et al. (2019) explore decentralized learning using a Bayesian-inspired belief update mechanism over connected networks. Further analyses in (Koloskova et al., 2020; Sun et al., 2022; Wu et al., 2025; Li et al., 2019) have demonstrated that incorporating MLT with multiple local SGD updates can also reduce communication complexity in a decentralized distributed setting.

## 2.3 MLT+CC in distributed learning

Combining MLT and CC, while simultaneously retaining their respective benefits, is notably challenging. In a centralized distributed setting, several recent algorithms successfully integrate these strategies, including CompressedScaffnew (Condat et al., 2022), LoCoDL (Condat et al., 2025), FedCOM (Haddadpour et al., 2021), FedPAQ (Reisizadeh et al., 2020), and Qsparse-Local-SGD (Basu et al., 2019). They leverage the advantage of both MLT and CC, achieving a multiplied reduction of communication complexity.

Despite these developments, few algorithms leveraging MLT+CC have been proposed in the context of decentralized distributed learning. Extending theoretical guarantees to the decentralized setting introduces substantial challenges due to the absence of a central coordinator. Complications arise from network topology constraints, the need for peer-to-peer communication, and heterogeneity in local data and model states. These factors make the convergence analysis significantly more intricate and have historically limited the rigorous understanding of MLT+CC in decentralized settings.

Among decentralized MLT+CC methods, each individual one covers only part of the design space. DFedAvgM from (Sun et al., 2022), SQuARM-SGD from (Singh et al., 2021), and LM-DFL from (Chen et al., 2024) are closely related to our method. DFedAvgM extends decentralized FedAvg by incorporating momentum-based local updates and CC. However, DFedAvgM is unable to reduce the order of total communication rounds through MLT and can only mitigate the effect from local variance of stochastic gradients when no momentum is applied. In addition, using CC will hurt the complexity result of DFedAvgM to obtain an $\epsilon$-stationary point unless the compression error is controlled in $O(\epsilon^4)$, which is a too-restrictive assumption. Without relying on such restrictive assumptions, a general convergence result is established to LM-DFL that incorporates both MLT and CC. However, LM-DFL is also unable to reduce the order of total communication rounds through MLT. SQuARM-SGD achieves multiplied communication reduction, but its analysis is tailored to momentum SGD and additionally assumes a symmetric mixing matrix.

Compared to existing decentralized methods combining MLT and CC, our contribution transcends a specific optimizer instantiation by providing a unified algorithmic framework and a generalizable convergence analysis template. Notably, we establish the first rigorous convergence guarantees for federated learning scenarios

| MLT Methods | CC | AG | #Iter | CommCost | MLTsave | CCsave |
|---|---|---|---|---|---|---|
| PD-SGD (Ge & Chang, 2023) | ✗ | ✗ | $\frac{1}{n\epsilon^4}$ | $\frac{1}{nK\epsilon^4}$ | ✓ | – |
| LSGT (Li et al., 2019) | ✗ | ✗ | $\frac{1}{n\epsilon^4}$ | $\frac{1}{nK\epsilon^4}$ | ✓ | – |
| DFedAvgM (Sun et al., 2022) | ✓ | ✗ | $\max\{\frac{K}{\epsilon^4}, \frac{K\epsilon^4}{s^2}\}^*$ | $\max\{\frac{1-s}{\epsilon^4}, \frac{(1-s)\epsilon^4}{s^2}\}^*$ | ✗ | ✓ |
| SQuARM-SGD (Singh et al., 2021) | ✓ | ✗ | $\frac{1}{n\epsilon^4}$ | $\frac{1-\eta}{nK\epsilon^4}$ | ✓ | ✓ |
| LM-DFL (Chen et al., 2024) | ✓ | ✗ | $\max\{\frac{1}{K^3\epsilon^4}, \frac{K^3 s^4}{\epsilon^4}\}^*$ | $\max\{\frac{1-s}{K^4\epsilon^4}, \frac{(1-s)K^2 s^4}{\epsilon^4}\}^*$ | ✗ | ✗ |
| LoDAdaC (this paper) | ✓ | ✓ | $\frac{1}{n\epsilon^4}$ | $\frac{1-\eta}{nK\epsilon^4}$ | ✓ | ✓ |

Table 1: Comparison between the proposed method and selected approaches that use MLT for nonconvex decentralized distributed learning. "CC" indicates whether compressed communication is employed; "AG" denotes the use of adaptive gradient updates; "#Iter" specifies the number of total iterations (per agent) to obtain an $\epsilon$-stationary point of problem (2), see Definition 1.2; "CommCost" refers to the total communication cost, where each communication round incurs a unit cost in the absence of compression; "MLTsave" indicates whether the number of communication rounds can be theoretically reduced by employing MLT; and "CCsave" reflects whether the total communication cost can be effectively reduced by compression. Here, the $O(\cdot)$ notation is omitted in the table, $\epsilon$ is assumed to be sufficiently small, and the number of local steps $K$ is at most $O(\epsilon^{-1})$. $^*s$ refers to the compression error given in (Sun et al., 2022), satisfying $\mathbb{E}_{\mathcal{Q}}\left[\|\boldsymbol{x} - \mathcal{Q}[\boldsymbol{x}]\|^2\right] \leq s^2 d$.

employing adaptive gradient methods such as Adam. Moreover, our convergence analysis relies exclusively on the contractive-error property of the compression operator, thus naturally extending to accommodate potentially biased compressors, including Top-$k$ sparsification. Lastly, our proof technique explicitly decouples the intricate interactions among adaptive gradient updates, MLT steps, and decentralized communication, effectively addressing the core analytical challenge of simultaneously ensuring stationarity and consensus in decentralized distributed training. Detailed comparisons are summarized in Table 1. We notice that the complexity result of our method is in the same order as that of SQuARM-SGD. However, with adaptive gradient updates, our method is able to achieve significantly faster empirical convergence, in particular for training GPT-style language models; see Section 4.

## 3  Decentralized adaptive methods with MLT and CC

In this section, we introduce a unified decentralized framework that integrates multiple local adaptive gradient updates with compressed communication. It is named LoDAdaC. Also, we provide convergence guarantees for the proposed algorithm under general nonconvex settings.

### 3.1  A unified algorithmic framework

We present the pseudocode of our framework in Algorithm 1. For simplicity, we take a single randomly sampled data point $\xi_i^t$ at each iteration. All our theoretical results remain valid by taking a mini-batch of samples.

In addition to Assumptions 1–2, we make the following assumption, which is standard in the analysis of both distributed and non-distributed adaptive SGMs (Chen et al., 2019; 2023b; Kingma & Ba, 2014; Reddi et al., 2018; Xu et al., 2023).

**Assumption 3** *The random samples $\{\xi_i^t\}_{i,t\geq 0}$ are independent. For each $t$ and $i \in \{1, 2, \ldots, n\}$, it holds $\mathbb{E}_t[\boldsymbol{g}_i^t] = \nabla f_i(\boldsymbol{x}_i^t)$. In addition, there are constants $B$ and $B_\infty$ such that $\|\boldsymbol{g}_i^t\| \leq B$, $\|\boldsymbol{g}_i^t\|_\infty \leq B_\infty$ for any $i \in \{1, 2, \ldots, n\}$ and any $t$, and $\|\nabla f_i(\boldsymbol{x})\| \leq B$, $\|\nabla f_i(\boldsymbol{x})\|_\infty \leq B_\infty$ for all $\boldsymbol{x}$.*

The unbiasedness condition $\mathbb{E}_t\left[g_i^t\right] = \nabla f_i\left(x_i^t\right)$ is standard in the literature of stochastic methods (Lan, 2020). The bounded gradient condition in Assumption 3 is stronger than the bounded-variance assumptions often used for non-adaptive methods; it can be restrictive for modern deep networks with heavy-tailed gradients. Nevertheless, similar assumptions are made in prior work such as (Chen et al., 2023b) for the convergence

analysis of adaptive methods. Extending the present adaptive+MLT+CC analysis under weaker conditions, such as generalized smoothness or bounded-variance assumptions, is an important direction for future work.

---

**Algorithm 1:** A **Lo**cal training-based **D**ecentralized framework with **Ada**ptive gradient updates and **C**ompressed communication (LoDAdaC)

---

1 **Input:** $\alpha > 0$, $0 \le \beta_1 < 1$, $\delta > 0$, $0 \le \gamma \le 1$, a maximum number $T$ of communication rounds, a number $K$ of local training steps per communication round, a $\eta$-compression operator $\mathcal{Q}$, $d$-dimension vector-value functions $\{r_t\}$, and a mixing matrix $\mathbf{W}$;

2 Let $\boldsymbol{x}_1^0 = \boldsymbol{x}_2^0 = \cdots = \boldsymbol{x}_n^0 = \underline{\boldsymbol{x}}_1^0 = \underline{\boldsymbol{x}}_2^0 = \cdots = \underline{\boldsymbol{x}}_n^0 = \boldsymbol{x}^0$, and set $\boldsymbol{m}_i^{-1}$, and $\boldsymbol{u}_i^{-1}$ to $\boldsymbol{0}$ for each $i$.

3 **for** $t = 0, 1, \cdots, TK - 1$ **do**

4      **for** *all agents $i \in \{1, 2, \ldots, n\}$ in parallel* **do**

5          Obtain one random sample $\xi_i^t$ and compute a stochastic gradient $\boldsymbol{g}_i^t \leftarrow \nabla F_i(\boldsymbol{x}_i^t, \xi_i^t)$;

6          Let $\boldsymbol{m}_i^t = \beta_1 \boldsymbol{m}_i^{t-1} + (1 - \beta_1) \boldsymbol{g}_i^t$;

7          Let $\boldsymbol{u}_i^t = r_t(\boldsymbol{g}_i^0, \boldsymbol{g}_i^1, \ldots, \boldsymbol{g}_i^t)$;

8          Update $\boldsymbol{x}_i^{t+\frac{1}{2}} = \boldsymbol{x}_i^t - \alpha \frac{\boldsymbol{m}_i^t}{\sqrt{\boldsymbol{u}_i^{t-1} + \delta}}$;

9          **if** $\mathrm{mod}(t+1, K) = 0$, **then**

10             Set $\underline{\boldsymbol{x}}_i^{t+1} = \underline{\boldsymbol{x}}_i^t + \mathcal{Q}[\boldsymbol{x}_i^{t+\frac{1}{2}} - \underline{\boldsymbol{x}}_i^t]$ and $\boldsymbol{x}_i^{t+1} = \boldsymbol{x}_i^{t+\frac{1}{2}} + \gamma(\sum_{j=1}^n \mathbf{W}_{ji} \underline{\boldsymbol{x}}_j^{t+1} - \underline{\boldsymbol{x}}_i^{t+1})$.

11          **else**

12             Update $\boldsymbol{x}_i^{t+1} = \boldsymbol{x}_i^{t+\frac{1}{2}}$, and $\underline{\boldsymbol{x}}_i^{t+1} = \underline{\boldsymbol{x}}_i^t$.

---

The condition in line 9 of Algorithm 1 indicates that neighbor communication happens every $K$ iterations, namely, $K$ local updates are performed per communication round. In addition, we only need to communicate the compressed vectors to obtain $\sum_{j=1}^n \mathbf{W}_{ji} \underline{\boldsymbol{x}}_j^{t+1}$, as explained below. For each $i = 1, 2, \ldots, n$, let agent $i$ maintain a vector $\boldsymbol{y}_i$ and initialize it as $\boldsymbol{y}_i^0 = \underline{\boldsymbol{x}}_i^0$. Then, for all $t \ge 0$, let $\boldsymbol{y}_i^{t+1} = \boldsymbol{y}_i^t$, if $\mathrm{mod}(t+1, K) \ne 0$, and $\boldsymbol{y}_i^{t+1} = \boldsymbol{y}_i^t + \sum_{j=1}^n \mathbf{W}_{ji} \mathcal{Q}\left[\boldsymbol{x}_j^{t+\frac{1}{2}} - \underline{\boldsymbol{x}}_j^t\right]$ otherwise. This way, we have $\boldsymbol{x}_i^{t+1} = \boldsymbol{x}_i^{t+\frac{1}{2}} + \gamma\left(\boldsymbol{y}_i^{t+1} - \underline{\boldsymbol{x}}_i^{t+1}\right)$ and thus enable the reduction of communication cost by only communicating compressed message.

With appropriate parameter $\beta_1$ and vector function $r_t$, the local update of LoDAdaC in line 8 of Algorithm 1 encompasses several well known optimizers as special cases. As we demonstrate in Section 3.2, our theoretical results apply to all optimizers listed in Table 2.

| Optimizer | Description |
|---|---|
| Vanilla SGD | $\beta_1 = 0$ and $r_t \equiv \boldsymbol{0}$, i.e., $\boldsymbol{u}_i^t = \boldsymbol{0}$ for all $i$ and $t$ |
| Momentum (Heavy-ball) SGD | $\beta_1 \in (0, 1)$ and $r_t \equiv \boldsymbol{0}$, i.e., $\boldsymbol{u}_i^t = \boldsymbol{0}$ for all $i$ and $t$ |
| AMSGrad | $\widehat{\boldsymbol{u}}_i^t = \beta_2 \widehat{\boldsymbol{u}}_i^{t-1} + (1 - \beta_2)\boldsymbol{g}_i^t \circ \boldsymbol{g}_i^t, \quad \boldsymbol{u}_i^t = \max\{\boldsymbol{u}_i^{t-1}, \widehat{\boldsymbol{u}}_i^t\},$ $\widehat{\boldsymbol{u}}_i^{-1} = \boldsymbol{0}$, and $\beta_2 \in (0, 1)$ |
| Adam | $\boldsymbol{u}_i^t = \beta_2 \boldsymbol{u}_i^{t-1} + (1 - \beta_2)\boldsymbol{g}_i^t \circ \boldsymbol{g}_i^t$, with $\beta_2 \in \left[\frac{\sqrt{TK}}{\sqrt{TK}+1}, 1\right)$ |
| Adam-mini | $\boldsymbol{u}_i^t = \beta_2 \boldsymbol{u}_i^{t-1} + (1 - \beta_2)\mathrm{mean}(\boldsymbol{g}_i^t \circ \boldsymbol{g}_i^t)$, with $\beta_2 \in \left[\frac{\sqrt{TK}}{\sqrt{TK}+1}, 1\right)$ |
| Averaged AdaGrad | $\boldsymbol{u}_i^t = \frac{1}{t+1} \sum_{s=0}^t \boldsymbol{g}_i^s \circ \boldsymbol{g}_i^s$ |

Table 2: Representative optimizers of Algorithm 1 with specific selections of $\beta_1$ and $r_t$

## 3.2 Convergence analysis

In this subsection, we establish the convergence rate results of Algorithm 1. We first derive a consensus error bound in Lemma 3.1. This bound is essential because it explicitly characterizes the relationship between

the consensus error and key algorithmic parameters, including the step size $\alpha$, MLT steps $K$, compression error $\eta$ of $\mathcal{Q}$, and the spectral gap $\rho$ of the communication graph. Such a characterization allows us to rigorously analyze how MLT and compression impact the convergence speed. Then we establish a bound in Theorem 3.1 on the objective gradient at averaged points. This bound enables us to show the final convergence rate results of our algorithm with several specific choices of popular adaptive updates. All proofs are given in the appendix.

**Lemma 3.1** *Under Assumptions 1–3, let $0 < \gamma \leq \frac{(1-\rho)(1-\eta^2)}{100}$. Then the sequence $\{x^t\}_{t=0}^{TK-1}$ generated by Algorithm 1 satisfies*

$$\frac{1}{TK} \sum_{t=0}^{TK-1} \mathbb{E}\left[\|\mathbf{X}_\perp^t\|^2\right] \leq \alpha^2 nK^2\overline{C}, \text{ where } \overline{C} := \frac{56}{\gamma(1-\rho)}\left(\frac{80}{\gamma(1-\rho)} + \frac{15}{1-\eta^2}\right)B^2\delta^{-1}. \tag{3}$$

**Theorem 3.1** *Suppose that Assumptions 1–3 hold and $\|u_i^t\|_\infty \leq B_u$ for all $t \geq 0$ and $i \in \{1, 2, \ldots, n\}$, for some $B_u > 0$. Let $\overline{C}$ denote the constant defined in (3) and $\alpha, \gamma > 0$ satisfy*

$$\alpha \leq \frac{\delta}{48L\sqrt{B_u + \delta}}, \quad \gamma \leq \frac{(1-\rho)(1-\eta^2)}{100}. \tag{4}$$

*Then the sequence $\{x^t\}_{t=0}^{TK-1}$ generated by Algorithm 1 satisfies*

$$
\begin{aligned}
\frac{\alpha}{4\sqrt{B_u + \delta}} \sum_{t=0}^{TK-1} \mathbb{E}\left[\|\nabla f\left(\overline{x}^t\right)\|^2\right] &\leq \mathbb{E}\left[f\left(x^0\right) - f^*\right] + \frac{\alpha TK}{8\sqrt{B_u + \delta}}\frac{\alpha^2 L^2\beta_1^2 B^2}{\delta(1-\beta_1)^2} \\
&+ \frac{\alpha\beta_1^2 B_\infty^2}{(1-\beta_1)^2}\left(4\sqrt{B_u + \delta} + \alpha L\right)\mathbb{E}\left[\sum_{t=0}^{TK-1}\frac{1}{n}\sum_{i=1}^{n}\left\|\frac{1}{\sqrt{u_i^{t-2} + \delta}} - \frac{1}{\sqrt{u_i^{t-1} + \delta}}\right\|^2\right] \\
&+ \alpha^2 L\left(\frac{24}{n\delta}TKB^2 + \frac{6L^2}{n\delta}\alpha^2 TnK^3\overline{C}\right) \\
&+ \frac{\alpha L^2}{n}\left(\frac{\sqrt{B_u + \delta}}{\delta} + \frac{1}{2\sqrt{\delta}}\right)\alpha^2 TnK^3\overline{C} + \sum_{t=0}^{TK-1}\frac{\alpha^3\beta_1^2 L^2 B^2}{\delta(1-\beta_1)^2}\left(\frac{\sqrt{B_u + \delta}}{\delta} + \frac{1}{2\sqrt{\delta}}\right).
\end{aligned}
\tag{5}
$$

For each optimizer listed in Table 2, we are able to show the condition $\|u_i^t\|_\infty \leq B_u, \forall t, \forall i$ for some constant $B_u$ and that $\mathbb{E}\left[\sum_{t=0}^{TK-1}\frac{1}{n}\sum_{i=1}^{n}\left\|\frac{1}{\sqrt{u_i^{t-2}+\delta}} - \frac{1}{\sqrt{u_i^{t-1}+\delta}}\right\|^2\right]$ is bounded; see Lemma A.7. Thus by Theorem 3.1, we specify the choice of $\alpha$ and obtain the convergence rate of Algorithm 1 by different ways of defining the second momentum term $u_i^t$ in line 7 of Algorithm 1.

**Theorem 3.2** *Under Assumptions 1–3, let $\delta = O(1)$ be a universal positive constant and $\overline{C}$ be the constant defined in (3). Choose $T$ and $K$ such that $\alpha$ and $\gamma > 0$ satisfy*

$$\alpha = \frac{4\theta\sqrt{n(B_\infty^2 + \delta)}}{\sqrt{TK}} \leq \min\left\{\frac{\delta}{48L\sqrt{B_\infty^2 + \delta}}, 1\right\}, \quad \gamma \leq \frac{(1-\rho)(1-\eta^2)}{100}, \tag{6}$$

*where $\theta = O(1)$ is a constant. Then for the sequence $\{x^t\}_{t=0}^{TK-1}$ generated by Algorithm 1 with any optimizer in Table 2, we have*

$$\frac{1}{TK}\sum_{t=0}^{TK-1}\mathbb{E}\left[\|\nabla f(\overline{x}^t)\|^2 + \frac{1}{n}\|\mathbf{X}_\perp^t\|^2\right] = O\left(\frac{f(x^0) - f^* + 1}{\sqrt{nTK}} + \frac{n}{TK} + \frac{nK\overline{C}}{T}\right). \tag{7}$$

### 3.3 Linear speed up, topology independence, and communication reduction

Based on the convergence rate results in Theorem 3.2, we discuss how the number $n$ of agents, the number $K$ of local updates, and compression ratio $1 - \eta$ affect the iteration complexity and communication complexity of our algorithm to produce an $\epsilon$-stationary point in expectation.

**Linear speed up and topology-independent step size.** By (7) and the definition of $\overline{C}$ in (3), if

$$T = \Omega\left(\frac{n^3 K^3}{(1-\rho)^8 (1-\eta^2)^4}\right), \tag{8}$$

then $\frac{nK\overline{C}}{T} = O(\frac{1}{\sqrt{nTK}})$, $\frac{n}{TK} = O(\frac{1}{\sqrt{nTK}})$, and we obtain

$$\frac{1}{TK} \sum_{t=0}^{TK-1} \mathbb{E}\left[\|\nabla f(\overline{\boldsymbol{x}}^t)\|^2 + \frac{1}{n}\|\mathbf{X}_\perp^t\|^2\right] = O\left(\frac{1}{\sqrt{nTK}}\right). \tag{9}$$

Letting $\tau$ be selected from $\{0, \ldots, TK - 1\}$ uniformly at random, we have from (9) that $\mathbb{E}\left[\|\nabla f(\overline{\boldsymbol{x}}^\tau)\|^2 + \frac{1}{n}\|\mathbf{X}_\perp^\tau\|^2\right] = O\left(\frac{1}{\sqrt{nTK}}\right)$. Hence, to obtain an $\epsilon$-stationary point in expectation, the total number of local iterations per agent is $TK = \Theta\left(\frac{1}{n\epsilon^4}\right)$.

Given $K$, we have $T = \Theta\left(\frac{1}{nK\epsilon^4}\right)$; when $\epsilon = O\left(\frac{(1-\rho)^2(1-\eta^2)}{nK}\right)$, the chosen $T$ will satisfy (8) and the first inequality in (6) holds. Thus in this case, we obtain a linear speed up with respect to $n$, and the step size $\alpha = \Theta(n\epsilon^2)$ and is independent of $\rho$ and $\eta$.

**Multiplied reduction of communication cost.** For a small enough $\epsilon > 0$, we can further reduce the order of communication rounds by picking $K$. Suppose $\epsilon = O\left(\left(\frac{(1-\rho)^2(1-\eta^2)}{n}\right)^{\frac{1}{\nu}}\right)$ for some $\nu \in (0, 1)$. Then we can choose $K = \Theta\left(\epsilon^{-1+\nu}\right)$ and $T = \Theta\left(\frac{\epsilon^{-3-\nu}}{n}\right)$, which satisfies (8). This way, compared to performing a single local update, i.e., $K = 1$, we reduce the number of communication rounds by an order of $\epsilon^{-1+\nu}$. In addition, by using an $\eta$-compression operator, our algorithm only needs $1 - \eta$ of communication amount as compared to using no compression. Therefore, the total communication volume required by our algorithm is $\Theta\left(\frac{1-\eta}{n\epsilon^{3+\nu}}\right)$, achieving multiplied reduction of the total communication cost.

## 4 Numerical experiments

In this section, we demonstrate the efficacy of the proposed framework over a set of numerical experiments. We consider three standard benchmarks, including training a convolutional neural network LeNet5 (LeCun et al., 1998) on the FashionMNIST dataset (Xiao et al., 2017), a ResNet architecture Fixup-ResNet-20 (Zhang et al., 2019) on the CIFAR-10 dataset (Krizhevsky et al., 2009), and a small-scale 10.7M parameter GPT model, from nanoGPT (Andrej, 2022), on the tiny-shakespeare dataset. We will show the performance of LoDAdaC equipped with the following adaptive gradient updates: AdaGrad, Adam, and AMSGrad on homogeneously distributed training data. We will compare LoDAdaC against SQuARM-SGD (Singh et al., 2021), which incorporates compressed communication and local training with a momentum-based SGD. Our methods improve over SQuARM-SGD with the addition of an adaptive update. We provide an additional comparison against DADAM (Nazari et al., 2022), which represents methods with decentralized and compressed communication but always with a single local update per communication round. A final experimental baseline for comparison is CDProxSGT (Yan et al., 2023), which represents non-adaptive methods with no local updates, for the sake of completeness.

We implement all of these methods in PyTorch. For FashionMNIST and CIFAR-10, we run our experiments on a CPU server. This server has two-way 64-core (256 threads) AMD EPYC 7742 CPUs at 2.25GHz and 2TB DDR4 memory. For tiny-shakespeare on nanoGPT, we run the experiments on a separate server with 4 NVIDIA A100 GPUs. Both test systems have Python 3.12.3 and PyTorch 2.7.0+cu126 installed, running on top of Ubuntu 24.04.2 LTS. The code and experimental scripts for our methods are publicly available at https://github.com/DecentralizedMethods/LoDAdaC.

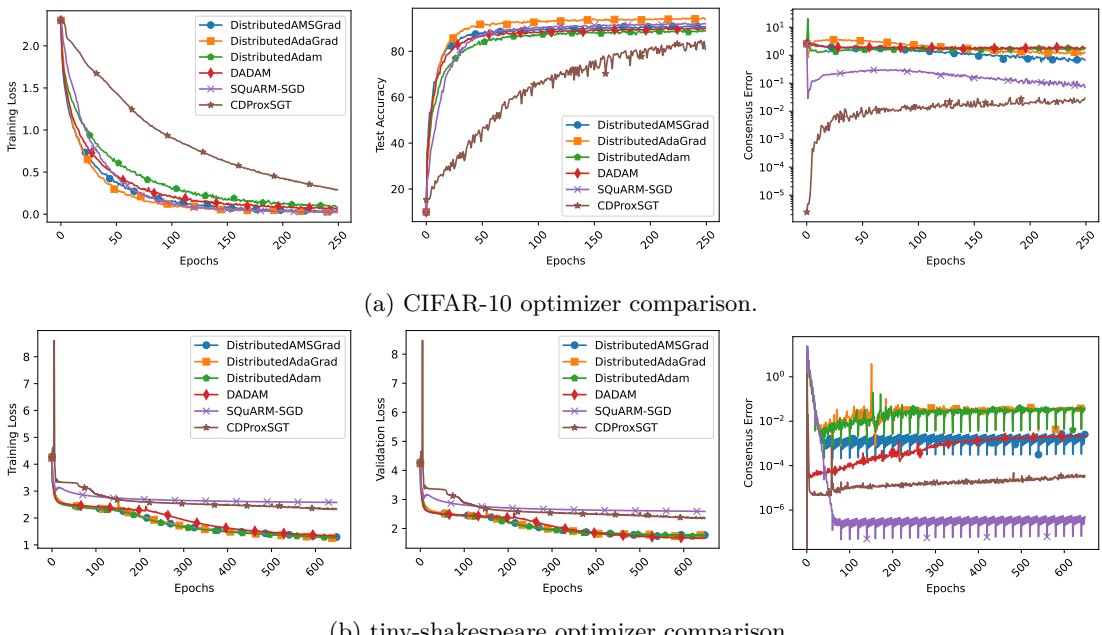

(a) CIFAR-10 optimizer comparison.

(b) tiny-shakespeare optimizer comparison.

Figure 1: **Optimizer Comparison:** Plotted above are the training loss, test accuracy, and consensus error of CIFAR-10 (top) and the training loss, validation loss, and consensus error of tiny-shakespeare (bottom) with training done on all of the various optimizers.

We will compare training loss, test accuracy, and consensus error, as well as validation loss for the GPT model. We will compare these values relative to the number of communication rounds and the communication volume. We ran a significant parametric study, evaluating possible parameters within: local updates $K = [1, 2, 5, 10, 20, 50]$, optimizers = [AdaGrad, Adam, AMSGrad, DADAM, SQuARM-SGD, CDProxSGT], Top-$k$ compression = [30%, 40%, 50%, 60%, None], agents = [4, 9, 16], topology = [ring, 2D-grid], data distribution = [IID, Dirichlet(1.0), Dirichlet(0.5)]. We will show a representative selection of these results below on CIFAR-10 and tiny-shakespeare, with FashionMNIST and the rest of the results appearing in the appendix. We use a batch size of 64 for the CIFAR-10 and FashionMNIST datasets and a batch size of 128 for training the GPT model. We initialize the learning rate to 0.001 for AdaGrad, Adam, and AMSGrad on CIFAR-10 and FashionMNIST and use $\beta$ values of $\beta_1 = 0.9, \beta_2 = 0.999$ for Adam and AMSGrad. We use a learning rate of 0.0001 on tiny-shakespeare. We tune the learning rate to 0.01 for SQuARM-SGD on CIFAR-10 and FashionMNIST and 0.005 on tiny-shakespeare — higher learning rates, such as the recommended learning rates of 0.1 and 0.2 from (Singh et al., 2021), did not converge with a number of tests when using our Top-$k$ compression operator.

## 4.1 Optimizer Comparison

We first compare the performance of different optimizers with $n = 4$ agents with a fixed local updates per communication of $K = 20$ and Top-$k$ compression of 40% and 50% for CIFAR-10 and tiny-shakespeare, respectively. We display these results in Figures 1a and 1b. We compare against SQuARM-SGD, DADAM, and CDProxSGT as our baselines for this set of experiments. Similar results are plotted for other values of $K$ in the appendix in Figure 7.

We observe that SQuARM-SGD is slightly slower to converge on CIFAR-10 with the given hyperparameters, but it achieves a similar test accuracy. CDProxSGT converges significantly slower and does not achieve an equivalent test accuracy in the given number of epochs. The performance of both non-adaptive methods in terms of validation loss is significantly worse on the GPT model, a known issue of training language models with non-adaptive momentum SGD methods (Zhao et al., 2025). We note that across these experiments,

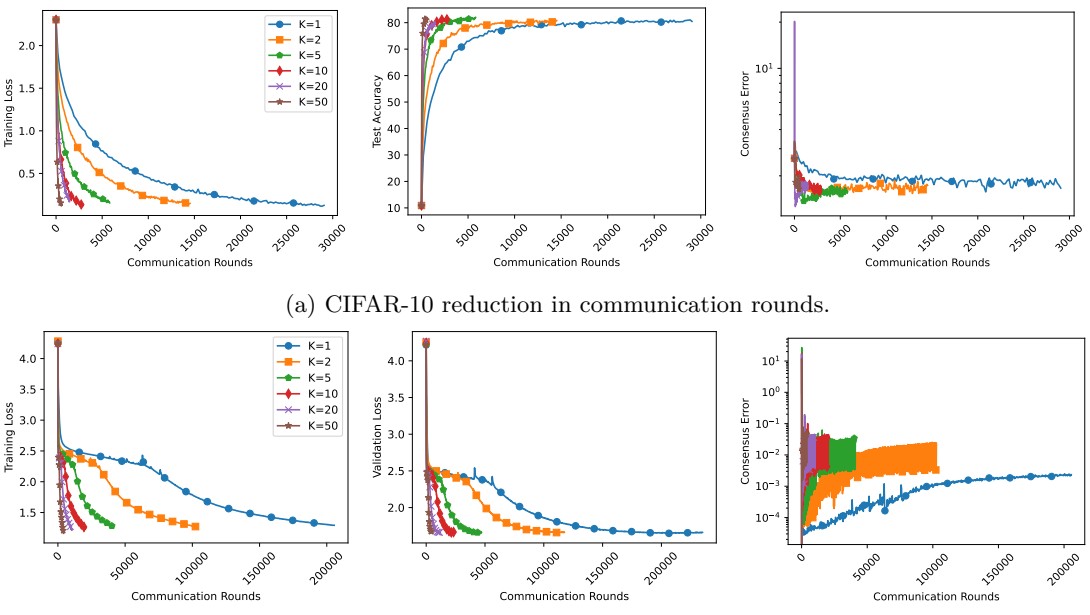

(a) CIFAR-10 reduction in communication rounds.

(b) tiny-shakespeare reduction in communication rounds.

Figure 2: **Number of Local Updates:** Plotted above are the training loss, test accuracy, and consensus error of CIFAR-10 (top) and the training loss, validation loss, and consensus error of tiny-shakespeare (bottom) with training done using the Adam optimizer across a number of possible $K$ values from 1 to 50.

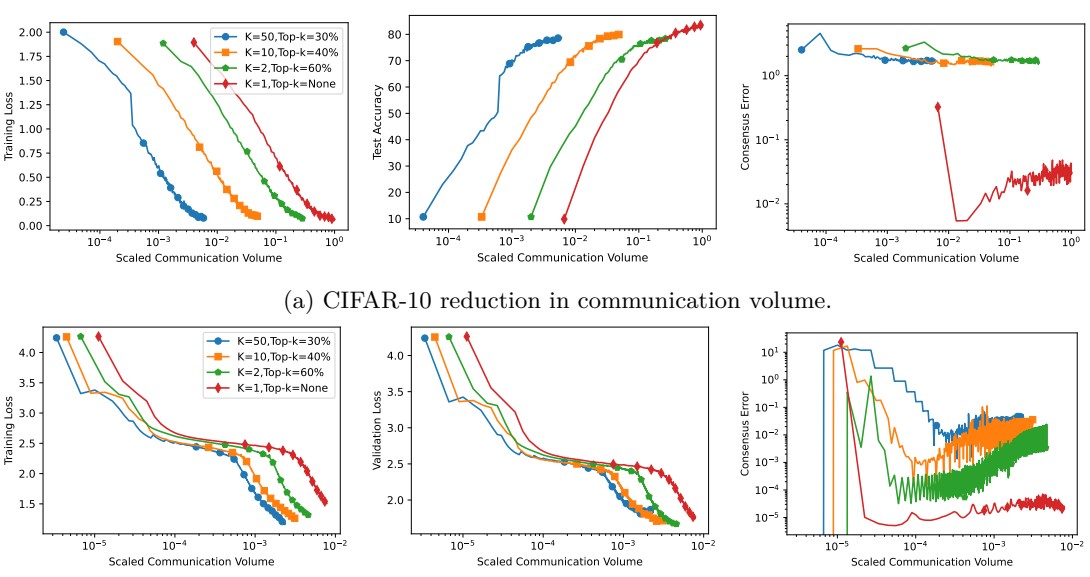

(a) CIFAR-10 reduction in communication volume.

(b) tiny-shakespeare reduction in communication volume.

Figure 3: **Communication Volume:** Plotted above are the training loss, test accuracy, and consensus error of CIFAR-10 (top) and the training loss, validation loss, and consensus error of tiny-shakespeare (bottom) with training done using the Adam optimizer across a number of possible $K$ and top-$k$ values.

AdaGrad and Adam are generally most performant overall, though the performance of all methods contained within our framework is relatively similar. As such, we will focus on Adam in subsequent results.

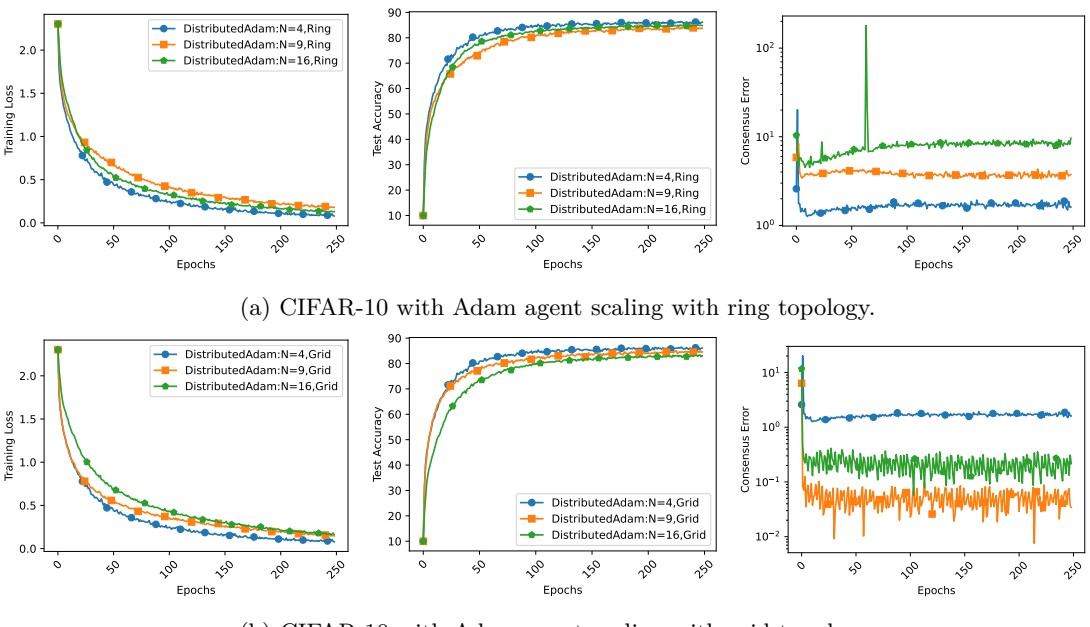

(a) CIFAR-10 with Adam agent scaling with ring topology.

(b) CIFAR-10 with Adam agent scaling with grid topology.

Figure 4: **Larger Agent Counts and Differing Topology:** Plotted above are the training loss, test accuracy, and consensus error of CIFAR-10 with training done using the Adam optimizer when scaling to 4, 9, and 16 agents on ring topology (top) and 2D grid topology (bottom). All experiments were run with $K = 20$ local updates per communication round.

## 4.2 Number of Local Updates

Our next set of experiments analyzes the effect of the number of local updates per communication round on training performance. Figures 2a and 2b give results with Adam training on CIFAR-10 and tiny-shakespeare using Top-$k$ compression of 40% and 50%, respectively. We plot training loss, test accuracy/validation loss, and consensus error against the total number of communication rounds for $K = 1$ to $K = 50$. We run all experiments to the same number of epochs, which gives a reduction in communication rounds proportional to $K$. For CIFAR-10, we note only a 1% loss in maximum test accuracy with $K = 50$ local updates per communication compared to the baseline $K = 1$ instance. On tiny-shakespeare, we likewise observe less than a 1% degradation of minimum validation loss when comparing $K = 50$ to $K = 1$ local updates per communication. We observed similar results with the same experiments on FashionMNIST in Figure 6 in the appendix.

## 4.3 Communication Volume

We continue our experiments by giving the relative proportion of communication volume used by our framework with a selection of $K$ and Top-$k$ values, as compared to a $K = 1$ baseline without compressed communication. Figures 3a and 3b give such a comparison using Adam on CIFAR-10 and tiny-shakespeare. We overall observe a significant reduction in communication volume at a relatively low cost to optimization quality. On CIFAR-10, we observe no loss in quality between $K = 2$ with Top-$k = 60\%$ to $K = 50$ with Top-$k = 30\%$, despite a reduction in communication volume of $50\times$. Comparing $K = 50$ with Top-$k = 30\%$ to the baseline with no compression or local updates, we note that we use only about 0.6% of the communication volume. The maximum test accuracies across all experiments are within a few percent of the baseline. We observe similar results on tiny-shakespeare when considering validation loss instead of test accuracy.

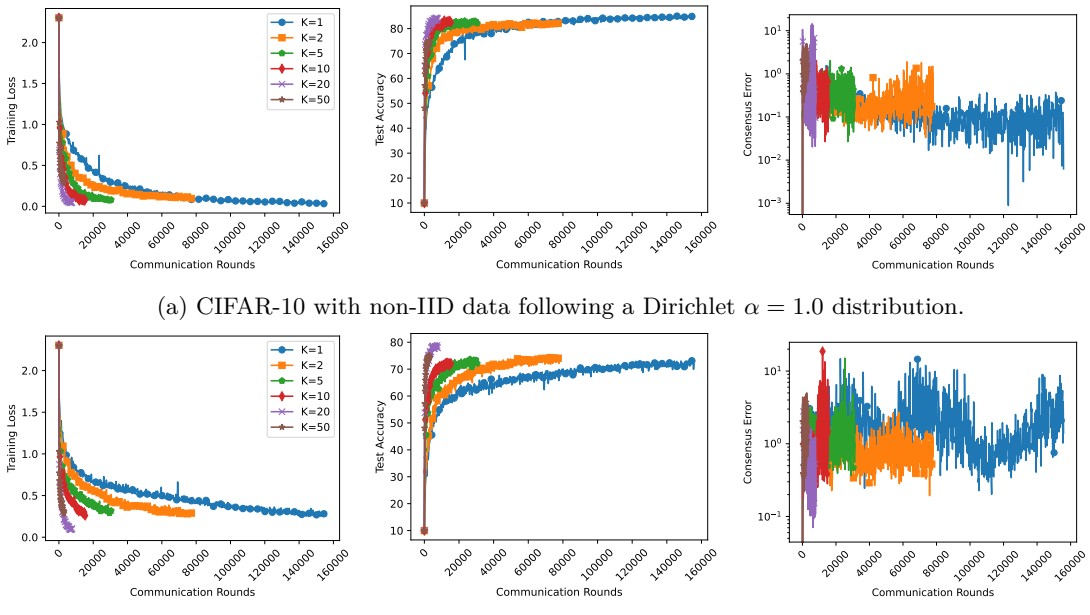

(a) CIFAR-10 with non-IID data following a Dirichlet $\alpha = 1.0$ distribution.

(b) CIFAR-10 with non-IID data following a Dirichlet $\alpha = 0.5$ distribution.

Figure 5: **Non-IID Data:** Plotted above are the training loss, test accuracy, and consensus error when using the Adam optimizer on CIFAR-10 with non-IID data across $K$ values from 1 to 50. We distribute data following a standard Dirichlet distribution process using $\alpha = 1.0$ (top) and $\alpha = 0.5$ (bottom).

### 4.4 Larger Agent Counts and Differing Topology

We demonstrate the linear scaling of our method by running experiments with 4, 9 and 16 agents in a ring and a 2D grid communication topology. Given in Figure 4 are plots of training loss, test accuracy, and consensus error for CIFAR-10 when using Adam optimizer. Similar results for AdaGrad and AMSGrad are displayed in Figure 10 in the appendix. We set $K = 20$ for all tests with Top-$k = 40\%$ compression. The 2D grid topology is defined as $3 \times 3$ for 9 agents and $4 \times 4$ for 16 agents. Note that the ring and grid topologies are equivalent for 4 agents. We observe relatively consistent results across all optimizers, with near-linear scaling in most cases. Minimum achieved training loss and maximum test accuracy are also relatively close, similar to as we previously observed in Figure 2 across varying $K$ values.

### 4.5 Heterogeneous Training Data

Our final experiments examine non-IID (independent and identically distributed) training data, emulating the client drift due to heterogeneity often observed in real decentralized environments. We use a Dirichlet distribution to partition training data. We run experiments with Dirichlet$(\alpha) = [0.5, 1.0]$, using CIFAR-10, Adam optimizer, local updates $K = [1, 2, 5, 10, 20, 50]$, and Top-$k = 40\%$, with results shown in Figure 5. Our experiments were run for approximately $5\times$ the number of communication rounds as the related IID tests shown in Figure 2a. We note that in the less skewed setup with Dirichlet parameter $\alpha = 1.0$, our method achieves convergence and test accuracies similar to those in Figure 2a, though convergence occurs more slowly. Consensus error is also significantly more variable, as expected. However, the tests with larger numbers of local updates still convergence to the same test accuracies of the baselines given by our optimizer comparison in Figure 1a in *significantly fewer* communication rounds. With a more skewed Dirichlet parameter of $\alpha = 0.5$, we note a more reduced rate of convergence, with many tests failing to reach an adequate test accuracy. Further experiments with lower $\alpha$ parameters down to $\alpha = 0.1$ showed correspondingly slower convergence with many tests failing to converge at all. Overall, these tests empirically demonstrate that our method offers some resilience towards heterogeneous training data, though other methods will likely be required if class label distributions are very significantly skewed.

## 5  Conclusions and Discussions

We propose a local training based decentralized algorithmic framework with adaptive local update and compressed communication. The local update of our framework encompasses several well-known optimizers as special cases, including vanilla SGD, momentum SGD, Adam, AMSGrad, AdaGrad, and Adam-Mini, and the established convergence results apply to all these optimizers. To the best of our knowledge, this is the first work to provide theoretical convergence guarantees for adaptive stochastic methods in MLT-based decentralized nonconvex optimization. Our empirical experiments further highlight the compounded benefits of integrating MLT with CC, demonstrating significant reductions in communication overhead without compromising convergence speed or accuracy.

Our current analysis relies on a bounded-gradient assumption, and our language-model experiments are limited to a small-scale transformer benchmark. Extending the framework to weaker assumptions, stronger data heterogeneity, and larger language models is a natural direction for future work.

## Acknowledgements

The authors would like to thank three anonymous reviewers for their valuable comments. This work is partly supported by NSF grant DMS-2208394, ONR grant N000142212573, and also by IBM through the IBM-Rensselaer Future of Computing Research Collaboration.

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

# A    Convergence analysis of compressed decentralized algorithms with multiple local adaptive gradient updates under nonconvex settings

In this section, we give a complete analysis of our decentralized algorithmic framework. We write the updates of Algorithm 1 in the more compact matrix form for all $t \in [TK]$,

$$\mathbf{M}^t = \beta_1 \mathbf{M}^{t-1} + (1 - \beta_1)\mathbf{G}^t, \tag{10}$$

$$\text{Update } \mathbf{U}^t \geq 0, \tag{11}$$

$$\mathbf{Y}^t = \frac{\mathbf{M}^t}{\sqrt{\mathbf{U}^{t-1} + \delta}}, \tag{12}$$

$$\mathbf{X}^{t+\frac{1}{2}} = \mathbf{X}^t - \alpha \mathbf{Y}^t, \tag{13}$$

$$\text{if } \mod(t+1, K) = 0, \text{ then } \underline{\mathbf{X}}^{t+1} = \underline{\mathbf{X}}^t + \mathcal{Q}\left[\mathbf{X}^{t+\frac{1}{2}} - \underline{\mathbf{X}}^t\right], \tag{14}$$

$$\mathbf{X}^{t+1} = \mathbf{X}^{t+\frac{1}{2}} + \gamma \underline{\mathbf{X}}^{t+1}(\mathbf{W} - \mathbf{I}), \tag{15}$$

$$\text{else, } \mathbf{X}^{t+1} = \mathbf{X}^{t+\frac{1}{2}}, \ \underline{\mathbf{X}}^{t+1} = \underline{\mathbf{X}}^t, \tag{16}$$

where

$$\mathbf{G}^t = \left[\boldsymbol{g}_1^t, \boldsymbol{g}_2^t, \ldots, \boldsymbol{g}_n^t\right], \ \mathbf{M}^t = \left[\boldsymbol{m}_1^t, \boldsymbol{m}_2^t, \ldots, \boldsymbol{m}_n^t\right], \ \underline{\mathbf{X}} = \left[\underline{\boldsymbol{x}}_1, \underline{\boldsymbol{x}}_2, \ldots, \underline{\boldsymbol{x}}_n\right],$$
$$\mathcal{Q}\left[\mathbf{X}\right] = \left[\mathcal{Q}[\boldsymbol{x}_1], \mathcal{Q}[\boldsymbol{x}_2], \ldots, \mathcal{Q}[\boldsymbol{x}_n]\right].$$

We let

$$\bar{\boldsymbol{x}} = \frac{1}{n}\mathbf{X}\mathbf{1}, \ \overline{\mathbf{X}} = \mathbf{X}\mathbf{J} = \bar{\boldsymbol{x}}\mathbf{1}^\top, \ \overline{\boldsymbol{m}}^t = \frac{1}{n}\mathbf{M}^t\mathbf{1}, \ \bar{\boldsymbol{y}}^t = \frac{1}{n}\mathbf{Y}^t\mathbf{1}, \ \overline{\mathbf{Y}}^t = \mathbf{Y}\mathbf{J}.$$

First we establish bounds on the sequence $\{\mathbf{M}^t\}$, $\{\mathbf{U}^t\}$ and $\{\mathbf{Y}^t\}$.

**Lemma A.1** *Under Assumption 3, it holds that for any $t \in [TK]$,*

$$\|\mathbf{M}^t\| \leq \left(1 - \beta_1^{t+1}\right)\sqrt{n}B \leq \sqrt{n}B, \quad \|\mathbf{Y}^t\| \leq \sqrt{n}B\delta^{-\frac{1}{2}}, \quad \|\mathbf{Y}_\perp^t\| \leq \sqrt{n}B\delta^{-\frac{1}{2}}, \tag{17}$$

$$\|\boldsymbol{m}_i^t\| \leq B, \quad \|\overline{\boldsymbol{m}}^t\| \leq B, \quad \|\boldsymbol{m}_i^t\|_\infty \leq B_\infty, \forall i \in \{1, 2, \ldots, n\}. \tag{18}$$

*Proof.* From the update of $\boldsymbol{m}$, i.e., $\boldsymbol{m}_i^t = \beta_1 \boldsymbol{m}_i^{t-1} + (1 - \beta_1)\boldsymbol{g}_i^t$, we have that for any $t \geq 0$ and each $i \in \{1, 2, \ldots, n\}$,

$$\|\boldsymbol{m}_i^t\| = \|\beta_1 \boldsymbol{m}_i^{t-1} + (1 - \beta_1)\mathbf{g}_i^t\| \leq \beta_1 \|\boldsymbol{m}_i^{t-1}\| + (1 - \beta_1)\|\mathbf{g}_i^t\| \leq \beta_1 \|\boldsymbol{m}_i^{t-1}\|_\infty + (1 - \beta_1)B,$$

where the second inequality holds from $\|\boldsymbol{g}_i^t\| \leq B$ by Assumption 3. Recursively applying the inequality above and noticing $\boldsymbol{m}_i^{-1} = \mathbf{0}$, we obtain

$$\|\boldsymbol{m}_i^t\| \leq \left(1 + \beta_1 + \beta_1^2 + \ldots + \beta_1^t\right)(1 - \beta_1)B = \left(1 - \beta_1^{t+1}\right)B \leq B.$$

Hence, it holds $\|\overline{\boldsymbol{m}}^t\| \leq B$ and $\|\mathbf{M}^t\| \leq \left(1 - \beta_1^{t+1}\right)\sqrt{n}B$. Now by $\mathbf{U}^t \geq \mathbf{0}$, we immediately have $\|\mathbf{Y}^t\| = \left\|\frac{\mathbf{M}^t}{\sqrt{\mathbf{U}^{t-1}+\delta}}\right\| \leq \sqrt{n}B\delta^{-\frac{1}{2}}$, and $\|\mathbf{Y}_\perp^t\|^2 = \left\|\mathbf{Y}^t - \overline{\mathbf{Y}}^t\right\|^2 = \|\mathbf{Y}^t\|^2 - \left\|\overline{\mathbf{Y}}^t\right\|^2 \leq \frac{nB^2}{\delta}$.

In addition, we have that for any $t \geq 0$ and each $i \in \{1, 2, \ldots, n\}$,

$$\|\boldsymbol{m}_i^t\|_\infty = \|\beta_1 \boldsymbol{m}_i^{t-1} + (1 - \beta_1)\mathbf{g}_i^t\|_\infty \leq \beta_1 \|\boldsymbol{m}_i^{t-1}\|_\infty + (1 - \beta_1)\|\mathbf{g}_i^t\|_\infty \leq \beta_1 \|\boldsymbol{m}_i^{t-1}\|_\infty + (1 - \beta_1)B_\infty,$$

where the second inequality follows from $\|\boldsymbol{g}_i^t\|_\infty \leq B_\infty$ by Assumption 3. Recursively applying the inequality above and noticing $\boldsymbol{m}_i^{-1} = \mathbf{0}$, we obtain

$$\|\boldsymbol{m}_i^t\|_\infty \leq \left(1 + \beta_1 + \beta_1^2 + \ldots + \beta_1^t\right)(1 - \beta_1)B_\infty = \left(1 - \beta_1^{t+1}\right)B_\infty \leq B_\infty.$$

The proof is then completed. $\qquad \square$

The next lemma shows the bound of the consensus error of $\mathbf{X}$.

**Lemma A.2** *Under Assumptions 1–3, let $\widehat{\rho} = 1 - \rho$, $0 < \gamma \leq \frac{(1-\rho)(1-\eta^2)}{100}$, and $\mathcal{Q}$ be an $\eta$-compression operator with $\eta \geq 0$. Then, the following statements hold:*

(i) *For any $r \in [T]$, it holds*

$$
\begin{aligned}
\mathbb{E}\left[\left\|\mathbf{X}_{\perp}^{(r+1)K}\right\|^2\right] &\leq \left(1 + \frac{\gamma\widehat{\rho}}{16}\right)\left(1 - \frac{\gamma\widehat{\rho}}{2}\right)^2 \mathbb{E}\left[\left\|\mathbf{X}_{\perp}^{rK}\right\|^2\right] \\
&+ 4\left(1 + \frac{\gamma\widehat{\rho}}{16}\right)\gamma\left(1 + \frac{2}{\widehat{\rho}}\right)\mathbb{E}\left[\left\|\mathbf{X}^{rK} - \underline{\mathbf{X}}^{(r+1)K}\right\|^2\right] + 2\left(1 + \frac{16}{\gamma\widehat{\rho}}\right)\alpha^2 K^2 n B^2 \delta^{-1}.
\end{aligned}
\tag{19}
$$

(ii) *For any $r \in [T]$, it holds*

$$
\begin{aligned}
\mathbb{E}\left[\left\|\mathbf{X}^{(r+1)K} - \underline{\mathbf{X}}^{(r+2)K}\right\|^2\right] &\leq 4\gamma^2\left(1 + \frac{\gamma\widehat{\rho}}{16}\right)\frac{4}{1-\eta^2}\mathbb{E}\left[\left\|\mathbf{X}_{\perp}^{rK}\right\|^2\right] \\
&+ \frac{3+\eta^2}{4}(1+8\gamma)\left(1 + \frac{\gamma\widehat{\rho}}{16}\right)\mathbb{E}\left[\left\|\mathbf{X}^{rK} - \underline{\mathbf{X}}^{(r+1)K}\right\|^2\right] \\
&+ \left(3\left(1 + \frac{16}{\gamma\widehat{\rho}}\right) + 2\left(1 + \frac{4}{1-\eta^2}\right)\right)\alpha^2 K^2 n B^2 \delta^{-1}.
\end{aligned}
\tag{20}
$$

(iii) *It holds*

$$
\frac{1}{TK}\sum_{t=0}^{TK-1}\mathbb{E}\left[\left\|\mathbf{X}_{\perp}^t\right\|^2\right] \leq \alpha^2 n K^2 \overline{C}, \quad \text{where } \overline{C} := \frac{56}{\gamma\widehat{\rho}}\left(\frac{80}{\gamma\widehat{\rho}} + \frac{15}{1-\eta^2}\right)B^2\delta^{-1}.
\tag{21}
$$

*If in addition $\gamma = \Theta((1-\rho)(1-\eta^2))$, we have $\overline{C} := \Theta\left(\frac{B^2}{(1-\rho)^4(1-\eta^2)^2}\right)$.*

*Proof.* (i) From the update rules (13)–(16), we have, for all $s \in [K-1]$,

$$
\mathbf{X}^{rK+s+1} = \mathbf{X}^{rK+s} - \alpha\mathbf{Y}^{rK+s},
$$

and for the final step in the block of size $K$, we conduct the compressed communication step.

First, using (18) and the inequality $(a+b)^2 \leq (1+\eta_1)a^2 + (1+\eta_1^{-1})b^2$ for any $\eta_1 > 0$, we have

$$
\begin{aligned}
\|\mathbf{X}_{\perp}^{rK+s+1}\|^2 &= \left\|\mathbf{X}_{\perp}^{rK} - \alpha\sum_{i=0}^{s}\mathbf{Y}_{\perp}^{rK+i}\right\|^2 \\
&\leq (1+\eta_1)\|\mathbf{X}_{\perp}^{rK}\|^2 + (1+\eta_1^{-1})\alpha^2\left\|\sum_{i=0}^{s}\mathbf{Y}_{\perp}^{rK+i}\right\|^2, \\
&\leq (1+\eta_1)\|\mathbf{X}_{\perp}^{rK}\|^2 + (1+\eta_1^{-1})K^2\alpha^2 n B^2 \delta^{-1}, \quad \forall s \in [K-1].
\end{aligned}
\tag{22}
$$

Next, we analyze the case of $s = K - 1$. By (15), it holds that

$$
\mathbf{X}_{\perp}^{(r+1)K} = \mathbf{X}^{(r+1)K-\frac{1}{2}} - \mathbf{X}^{(r+1)K}\mathbf{J} + \gamma\underline{\mathbf{X}}^{(r+1)K}(\mathbf{W} - \mathbf{I}).
$$

Noticing $\mathbf{X}^{(r+1)K}\mathbf{J} = \mathbf{X}^{(r+1)K-\frac{1}{2}}\mathbf{J}$ from (14)–(15), and $\mathbf{J}(\mathbf{W} - \mathbf{I}) = \mathbf{0}$, we have

$$
\begin{aligned}
\mathbf{X}_{\perp}^{(r+1)K} &= \mathbf{X}^{(r+1)K-\frac{1}{2}} - \mathbf{X}^{(r+1)K-\frac{1}{2}}\mathbf{J} + \gamma(\underline{\mathbf{X}}^{(r+1)K} - \mathbf{X}^{(r+1)K-\frac{1}{2}}\mathbf{J})(\mathbf{W} - \mathbf{I}) \\
&= \mathbf{X}^{(r+1)K-\frac{1}{2}}(\mathbf{I} - \mathbf{J})((1-\gamma)\mathbf{I} + \gamma\mathbf{W}) + \gamma(\underline{\mathbf{X}}^{(r+1)K} - \mathbf{X}^{(r+1)K-\frac{1}{2}})(\mathbf{W} - \mathbf{I}).
\end{aligned}
\tag{23}
$$

Denote $\widehat{\mathbf{W}} = (1 - \gamma)\mathbf{I} + \gamma\mathbf{W}$. For any $\eta_2 > 0$, it then holds

$$
\left\|\mathbf{X}_\perp^{(r+1)K}\right\|^2
$$
$$
\leq (1 + \eta_2) \left\|\mathbf{X}^{(r+1)K-\frac{1}{2}}(\mathbf{I} - \mathbf{J})\widehat{\mathbf{W}}\right\|^2 + \left(1 + \eta_2^{-1}\right) \left\|\gamma\left(\underline{\mathbf{X}}^{(r+1)K} - \mathbf{X}^{(r+1)K-\frac{1}{2}}\right)(\mathbf{W} - \mathbf{I})\right\|^2
$$
$$
\leq (1 + \eta_2) \left\|\mathbf{X}^{(r+1)K-\frac{1}{2}}(\mathbf{I} - \mathbf{J})\widehat{\mathbf{W}}\right\|^2 + 4\left(1 + \eta_2^{-1}\right)\gamma^2 \left\|\left(\underline{\mathbf{X}}^{(r+1)K} - \mathbf{X}^{(r+1)K-\frac{1}{2}}\right)\right\|^2, \tag{24}
$$

where the second inequality follows from $\|\mathbf{W} - \mathbf{I}\|_2 \leq 2$.

Recalling $\widehat{\rho} = 1 - \rho$ and by $(\mathbf{I} - \mathbf{J})\mathbf{J} = \mathbf{0}$, we have

$$
\left\|\mathbf{X}^{(r+1)K-\frac{1}{2}}(\mathbf{I} - \mathbf{J})\widehat{\mathbf{W}}\right\|
$$
$$
\leq (1 - \gamma)\left\|\mathbf{X}^{(r+1)K-\frac{1}{2}}(\mathbf{I} - \mathbf{J})\right\| + \gamma\left\|\mathbf{X}^{(r+1)K-\frac{1}{2}}(\mathbf{I} - \mathbf{J})\mathbf{W}\right\|
$$
$$
= (1 - \gamma)\left\|\mathbf{X}^{(r+1)K-\frac{1}{2}}(\mathbf{I} - \mathbf{J})\right\| + \gamma\left\|\mathbf{X}^{(r+1)K-\frac{1}{2}}(\mathbf{I} - \mathbf{J})(\mathbf{W} - \mathbf{J})\right\|
$$
$$
\leq (1 - \gamma)\left\|\mathbf{X}^{(r+1)K-\frac{1}{2}}(\mathbf{I} - \mathbf{J})\right\| + \gamma\rho\left\|\mathbf{X}^{(r+1)K-\frac{1}{2}}(\mathbf{I} - \mathbf{J})\right\|
$$
$$
= (1 - \gamma\widehat{\rho})\left\|\mathbf{X}^{(r+1)K-\frac{1}{2}}(\mathbf{I} - \mathbf{J})\right\|. \tag{25}
$$

Substituting (25) into (24), we obtain

$$
\left\|\mathbf{X}_\perp^{(r+1)K}\right\|^2
$$
$$
\leq (1 + \eta_2)(1 - \gamma\widehat{\rho})^2 \left\|\mathbf{X}^{(r+1)K-\frac{1}{2}}(\mathbf{I} - \mathbf{J})\right\|^2 + 4\left(1 + \eta_2^{-1}\right)\gamma^2 \left\|\underline{\mathbf{X}}^{(r+1)K} - \mathbf{X}^{(r+1)K-\frac{1}{2}}\right\|^2. \tag{26}
$$

For the first term in the RHS of (26), using the bound $\|\mathbf{Y}_\perp^t\| \leq \sqrt{n}B\delta^{-\frac{1}{2}}$, we have

$$
\left\|\mathbf{X}^{(r+1)K-\frac{1}{2}}(\mathbf{I} - \mathbf{J})\right\|^2 = \left\|\left(\mathbf{X}^{(r+1)K-1} - \alpha\mathbf{Y}^{(r+1)K-1}\right)(\mathbf{I} - \mathbf{J})\right\|^2
$$
$$
= \left\|\mathbf{X}_\perp^{(r+1)K-1} - \alpha\mathbf{Y}_\perp^{(r+1)K-1}\right\|^2
$$
$$
\overset{(22)}{\leq} (1 + \eta_1)\left\|\mathbf{X}_\perp^{rK}\right\|^2 + \left(1 + \eta_1^{-1}\right)\alpha^2 K^2 nB^2\delta^{-1}. \tag{27}
$$

For the second term in the RHS of (26), using the bound $\|\mathbf{Y}^t\| \leq \sqrt{n}B\delta^{-\frac{1}{2}}$, we have

$$
\left\|\underline{\mathbf{X}}^{(r+1)K} - \mathbf{X}^{(r+1)K-\frac{1}{2}}\right\|^2 = \left\|\underline{\mathbf{X}}^{(r+1)K} - \mathbf{X}^{rK} + \alpha\sum_{i=0}^{K-1}\mathbf{Y}^{rK+i}\right\|^2
$$
$$
\leq (1 + \eta_1)\left\|\underline{\mathbf{X}}^{(r+1)K} - \mathbf{X}^{rK}\right\|^2 + (1 + \eta_1^{-1})\alpha^2 \left\|\sum_{i=0}^{K-1}\mathbf{Y}^{rK+i}\right\|^2
$$
$$
\leq (1 + \eta_1)\left\|\underline{\mathbf{X}}^{(r+1)K} - \mathbf{X}^{rK}\right\|^2 + \left(1 + \eta_1^{-1}\right)\alpha^2 K^2 nB^2\delta^{-1}. \tag{28}
$$

Plugging (27)–(28) back into (26), we arrive at

$$
\left\|\mathbf{X}_\perp^{(r+1)K}\right\|^2
$$
$$
\leq (1 + \eta_2)(1 - \gamma\widehat{\rho})^2(1 + \eta_1)\left\|\mathbf{X}_\perp^{rK}\right\|^2 + 4\left(1 + \eta_2^{-1}\right)\gamma^2(1 + \eta_1)\left\|\underline{\mathbf{X}}^{(r+1)K} - \mathbf{X}^{rK}\right\|^2
$$
$$
+ \left((1 + \eta_2)(1 - \gamma\widehat{\rho})^2 + 4\left(1 + \eta_2^{-1}\right)\gamma^2\right)\left(1 + \eta_1^{-1}\right)\alpha^2 K^2 nB^2\delta^{-1}. \tag{29}
$$

Let $\eta_1 = \frac{\gamma\widehat{\rho}}{16}$ and $\eta_2 = \frac{\gamma\widehat{\rho}}{2}$. By the definition of $\gamma$, $0 < \gamma\widehat{\rho} < 1$, we then have

$$(1+\eta_2)(1-\gamma\widehat{\rho})^2\,(1+\eta_1) \le \left(1+\frac{\gamma\widehat{\rho}}{16}\right)\left(1-\frac{\gamma\widehat{\rho}}{2}\right)^2, \tag{30}$$

$$4\left(1+\eta_2^{-1}\right)\gamma^2(1+\eta_1) \le 4\left(1+\frac{\gamma\widehat{\rho}}{16}\right)\gamma\left(1+\frac{2}{\widehat{\rho}}\right), \tag{31}$$

$$\left((1+\eta_2)(1-\gamma\widehat{\rho})^2 + 4\left(1+\eta_2^{-1}\right)\gamma^2\right)\left(1+\eta_1^{-1}\right) \le 2\left(1+\frac{16}{\gamma\widehat{\rho}}\right). \tag{32}$$

We then complete the proof of (19) by using the inequalities (30)-(32) in (29).

(ii) From (14) and Definition 1.1, for any $\eta_3 > 0$, it holds that

$$\mathbb{E}\left\|\underline{\mathbf{X}}^{(r+2)K} - \mathbf{X}^{(r+1)K}\right\|^2 = \mathbb{E}\left\|\underline{\mathbf{X}}^{(r+2)K-1} + \mathcal{Q}\left[\mathbf{X}^{(r+2)K-\frac{1}{2}} - \underline{\mathbf{X}}^{(r+2)K-1}\right] - \mathbf{X}^{(r+1)K}\right\|^2$$

$$=\mathbb{E}\left\|\underline{\mathbf{X}}^{(r+2)K-1} - \mathbf{X}^{(r+2)K-\frac{1}{2}} + \mathbf{X}^{(r+2)K-\frac{1}{2}} - \mathbf{X}^{(r+1)K} + \mathcal{Q}\left[\mathbf{X}^{(r+2)K-\frac{1}{2}} - \underline{\mathbf{X}}^{(r+2)K-1}\right]\right\|^2$$

$$\le(1+\eta_3)\mathbb{E}\left\|\mathbf{X}^{(r+2)K-\frac{1}{2}} - \underline{\mathbf{X}}^{(r+2)K-1} - \mathcal{Q}\left[\mathbf{X}^{(r+2)K-\frac{1}{2}} - \underline{\mathbf{X}}^{(r+2)K-1}\right]\right\|^2$$

$$\quad + (1+\eta_3^{-1})\mathbb{E}\left\|\mathbf{X}^{(r+2)K-\frac{1}{2}} - \mathbf{X}^{(r+1)K}\right\|^2$$

$$\le(1+\eta_3)\eta^2\mathbb{E}\left\|\mathbf{X}^{(r+2)K-\frac{1}{2}} - \underline{\mathbf{X}}^{(r+2)K-1}\right\|^2 + (1+\eta_3^{-1})\mathbb{E}\left\|\mathbf{X}^{(r+2)K-\frac{1}{2}} - \mathbf{X}^{(r+1)K}\right\|^2. \tag{33}$$

For the second term in the RHS of (33), using the bound $\|\mathbf{Y}^t\| \le \sqrt{n}B\delta^{-\frac{1}{2}}$, we obtain

$$\left\|\mathbf{X}^{(r+2)K-\frac{1}{2}} - \mathbf{X}^{(r+1)K}\right\|^2 = \alpha^2\left\|\sum_{i=0}^{K-1}\mathbf{Y}^{(r+1)K+i}\right\|^2 \le \alpha^2 K^2 nB^2\delta^{-1}. \tag{34}$$

For the first term in the RHS of (33), by $\underline{\mathbf{X}}^{(r+2)K-1} = \underline{\mathbf{X}}^{(r+1)K}$ and $\|\mathbf{Y}^t\| \le \sqrt{n}B\delta^{-\frac{1}{2}}$, for any $\eta_4 > 0$, we have

$$\mathbb{E}\left\|\mathbf{X}^{(r+2)K-\frac{1}{2}} - \underline{\mathbf{X}}^{(r+2)K-1}\right\|^2 = \mathbb{E}\left\|\mathbf{X}^{(r+1)K} - \alpha\sum_{i=0}^{K-1}\mathbf{Y}^{(r+1)K+i} - \underline{\mathbf{X}}^{(r+1)K}\right\|^2$$

$$\le(1+\eta_4)\mathbb{E}\left\|\mathbf{X}^{(r+1)K} - \underline{\mathbf{X}}^{(r+1)K}\right\|^2 + (1+\eta_4^{-1})\mathbb{E}\left\|\alpha\sum_{i=0}^{K-1}\mathbf{Y}^{(r+1)K+i}\right\|^2$$

$$\le(1+\eta_4)\mathbb{E}\left\|\mathbf{X}^{(r+1)K} - \underline{\mathbf{X}}^{(r+1)K}\right\|^2 + (1+\eta_4^{-1})\alpha^2 K^2 nB^2\delta^{-1}. \tag{35}$$

Substituting (34)–(35) into (33), we have

$$\mathbb{E}\left\|\underline{\mathbf{X}}^{(r+2)K} - \mathbf{X}^{(r+1)K}\right\|^2$$

$$\le(1+\eta_3)\eta^2\left((1+\eta_4)\mathbb{E}\left\|\mathbf{X}^{(r+1)K} - \underline{\mathbf{X}}^{(r+1)K}\right\|^2 + (1+\eta_4^{-1})\alpha^2 K^2 nB^2\delta^{-1}\right)$$

$$\quad + (1+\eta_3^{-1})\alpha^2 K^2 nB^2\delta^{-1}. \tag{36}$$

We now bound $\left\|\mathbf{X}^{(r+1)K} - \underline{\mathbf{X}}^{(r+1)K}\right\|^2$. By (23), we have that for any $\eta_5 > 0$,

$$\mathbb{E}\left[\left\|\mathbf{X}^{(r+1)K} - \underline{\mathbf{X}}^{(r+1)K}\right\|^2\right]$$

$$=\mathbb{E}\left[\left\|\left(\underline{\mathbf{X}}^{(r+1)K} - \mathbf{X}^{(r+1)K-\frac{1}{2}}\right)(\gamma(\mathbf{W}-\mathbf{I})-\mathbf{I}) + \gamma\mathbf{X}^{(r+1)K-\frac{1}{2}}(\mathbf{I}-\mathbf{J})(\mathbf{W}-\mathbf{I})\right\|^2\right]$$

$$\le(1+\eta_5)(1+2\gamma)^2\mathbb{E}\left[\left\|\underline{\mathbf{X}}^{(r+1)K} - \mathbf{X}^{(r+1)K-\frac{1}{2}}\right\|^2\right] + (1+\eta_5^{-1})4\gamma^2\mathbb{E}\left[\left\|\mathbf{X}_\perp^{(r+1)K-\frac{1}{2}}\right\|^2\right], \tag{37}$$

where we have used $\mathbf{JW} = \mathbf{J}$ in the equality and $\|\gamma(\mathbf{W} - \mathbf{I}) - \mathbf{I}\|_2 \leq \gamma\|\mathbf{W} - \mathbf{I}\|_2 + \|\mathbf{I}\|_2 \leq 1 + 2\gamma$ and $\|\mathbf{W} - \mathbf{I}\|_2 \leq 2$ in the inequality.

For the first term in the RHS of (37), we know from (28) that for any $\eta_1 > 0$,

$$\left\|\underline{\mathbf{X}}^{(r+1)K} - \mathbf{X}^{(r+1)K-\frac{1}{2}}\right\|^2 \leq (1+\eta_1)\left\|\underline{\mathbf{X}}^{(r+1)K} - \mathbf{X}^{rK}\right\|^2 + \left(1+\eta_1^{-1}\right)\alpha^2 K^2 nB^2\delta^{-1}. \tag{38}$$

For the second term in the RHS of (37), we know from (27) that

$$\left\|\mathbf{X}_\perp^{(r+1)K-\frac{1}{2}}\right\|^2 = \left\|\mathbf{X}^{(r+1)K-\frac{1}{2}}(\mathbf{I}-\mathbf{J})\right\|^2 \leq (1+\eta_1)\left\|\mathbf{X}_\perp^{rK}\right\|^2 + \left(1+\eta_1^{-1}\right)\alpha^2 K^2 nB^2\delta^{-1}. \tag{39}$$

Plugging (38) and (39) into (37), we have

$$\mathbb{E}\left[\left\|\mathbf{X}^{(r+1)K} - \underline{\mathbf{X}}^{(r+1)K}\right\|^2\right]$$
$$\leq \left(1+\eta_5^{-1}\right)4\gamma^2\left((1+\eta_1)\left\|\mathbf{X}_\perp^{rK}\right\|^2 + \left(1+\eta_1^{-1}\right)\alpha^2 K^2 nB^2\delta^{-1}\right)$$
$$+ (1+\eta_5)(1+2\gamma)^2\left((1+\eta_1)\left\|\underline{\mathbf{X}}^{(r+1)K} - \mathbf{X}^{rK}\right\|^2 + \left(1+\eta_1^{-1}\right)\alpha^2 K^2 nB^2\delta^{-1}\right). \tag{40}$$

Combining (40) and (36), we arrive at

$$\mathbb{E}\left\|\underline{\mathbf{X}}^{(r+2)K} - \mathbf{X}^{(r+1)K}\right\|^2$$
$$\leq (1+\eta_3)\eta^2(1+\eta_4)\left(1+\eta_5^{-1}\right)4\gamma^2(1+\eta_1)\left\|\mathbf{X}_\perp^{rK}\right\|^2$$
$$+ (1+\eta_3)\eta^2(1+\eta_4)(1+\eta_5)(1+2\gamma)^2(1+\eta_1)\left\|\underline{\mathbf{X}}^{(r+1)K} - \mathbf{X}^{rK}\right\|^2$$
$$+ (1+\eta_3)\eta^2(1+\eta_4)\left(1+\eta_5^{-1}\right)4\gamma^2\left(1+\eta_1^{-1}\right)\alpha^2 K^2 nB^2\delta^{-1}$$
$$+ (1+\eta_3)\eta^2(1+\eta_4)(1+\eta_5)(1+2\gamma)^2\left(1+\eta_1^{-1}\right)\alpha^2 K^2 nB^2\delta^{-1}$$
$$+ (1+\eta_3)\eta^2(1+\eta_4^{-1})\alpha^2 K^2 nB^2\delta^{-1} + (1+\eta_3^{-1})\alpha^2 K^2 nB^2\delta^{-1}. \tag{41}$$

Let $\eta_3 = \eta_4 = \eta_5 = \frac{1-\eta^2}{4}$, and $\eta_1 = \frac{\gamma\widehat{\rho}}{16}$. By $\gamma \leq \frac{(1-\rho)(1-\eta^2)}{100}$ and $0 < \eta, \widehat{\rho} < 1$, we have

$$(1+\eta_3)\eta^2(1+\eta_4)(1+\eta_5)(1+2\gamma)^2(1+\eta_1) \leq \frac{3+\eta^2}{4}(1+8\gamma)\left(1+\frac{\gamma\widehat{\rho}}{16}\right), \tag{42}$$

$$(1+\eta_3)\eta^2(1+\eta_4)\left(1+\eta_5^{-1}\right)4\gamma^2(1+\eta_1) \leq 4\gamma^2\left(1+\frac{\gamma\widehat{\rho}}{16}\right)\frac{4}{1-\eta^2}, \tag{43}$$

$$(1+\eta_3)\eta^2(1+\eta_4)\left(1+\eta_5^{-1}\right)4\gamma^2\left(1+\eta_1^{-1}\right) \leq 1+\frac{16}{\gamma\widehat{\rho}}, \tag{44}$$

$$(1+\eta_3)\eta^2(1+\eta_4)(1+\eta_5)(1+2\gamma)^2\left(1+\eta_1^{-1}\right) \leq 2\left(1+\frac{16}{\gamma\widehat{\rho}}\right), \tag{45}$$

$$(1+\eta_3)\eta^2(1+\eta_4^{-1}) + (1+\eta_3^{-1}) \leq 2\left(1+\frac{4}{1-\eta^2}\right). \tag{46}$$

We complete the proof of (20) by using the inequalities (42)-(46) in (41).

(iii) Denote $\Omega^r = \mathbb{E}\left[\left\|\mathbf{X}_\perp^{rK}\right\|^2\right] + \mathbb{E}\left[\left\|\mathbf{X}^{rK} - \underline{\mathbf{X}}^{(r+1)K}\right\|^2\right]$. Then, since $0 < \eta < 1$, the two inequalities in (19) and (20) imply

$$\Omega^{k+1} \leq A_0\Omega^k + A_1, \tag{47}$$

where

$$A_0 = \left(1 + \frac{\gamma\widehat{\rho}}{16}\right) \max\left\{ \left(1 - \frac{\gamma\widehat{\rho}}{2}\right)^2 + 4\gamma^2 \frac{4}{1-\eta^2}, 4\gamma\left(1 + \frac{2}{\widehat{\rho}}\right) + \frac{3+\eta^2}{4}(1+8\gamma) \right\} \tag{48}$$

$$A_1 = \left(\frac{80}{\gamma\widehat{\rho}} + \frac{15}{1-\eta^2}\right)\alpha^2 K^2 n B^2 \delta^{-1}. \tag{49}$$

By $\widehat{\rho} = 1 - \rho$, it holds that

$$\gamma \leq \frac{(1-\rho)(1-\eta^2)}{100} \leq \min\left\{ \frac{2\widehat{\rho}(1-\eta^2)}{\widehat{\rho}^2 + 32\widehat{\rho} + 64 + 48\widehat{\rho} + 16\widehat{\rho}\eta^2}, \frac{7\widehat{\rho}(1-\eta^2)}{128 + 2\widehat{\rho}^2(1-\eta^2)} \right\}. \tag{50}$$

Notice that $\gamma \leq \frac{7\widehat{\rho}(1-\eta^2)}{128 + 2\widehat{\rho}^2(1-\eta^2)}$ yields $\left(1 - \frac{\gamma\widehat{\rho}}{2}\right)^2 + 4\gamma^2\frac{4}{1-\eta^2} \leq 1 - \frac{\widehat{\rho}\gamma}{8}$, and $\gamma \leq \frac{2\widehat{\rho}(1-\eta^2)}{\widehat{\rho}^2 + 32\widehat{\rho} + 64 + 48\widehat{\rho} + 16\widehat{\rho}\eta^2}$ implies $4\gamma\left(1 + \frac{2}{\widehat{\rho}}\right) + \frac{3+\eta^2}{4}(1+8\gamma) \leq 1 - \frac{\widehat{\rho}\gamma}{8}$. Thus

$$A_0 \leq \left(1 + \frac{\gamma\widehat{\rho}}{16}\right)\left(1 - \frac{\widehat{\rho}\gamma}{8}\right) \leq 1 - \frac{\gamma\widehat{\rho}}{16} < 1.$$

From $x_i^0 = \overline{x}^0 = \underline{x}_i^0$, $\forall i \in \{1, 2, \ldots, n\}$, we have $\|\mathbf{X}_\perp^0\|^2 = 0$ and $\underline{\mathbf{X}}^0 = \mathbf{0}$. By (36) with $r = -1$ and (46), we have

$$\mathbb{E}\left[\left\|\mathbf{X}^0 - \underline{\mathbf{X}}^K\right\|^2\right] \leq 2\left(1 + \frac{4}{1-\eta^2}\right)\alpha^2 K^2 n B^2 \delta^{-1} \leq \frac{2}{\gamma\widehat{\rho}}A_1.$$

Thus, multiplying both sides of (47) by $A_0^{r-k}$ and summing it over $k = 0, 1, \ldots, r-1$ gives

$$\begin{aligned}
\Omega^r &\leq A_0^{r+1}\Omega^0 + \sum_{k=0}^{r-1} A_0^{r-k}A_1 \leq \Omega^0 + \frac{16}{\gamma\widehat{\rho}}A_1 \\
&= \frac{16}{\gamma\widehat{\rho}}A_1 + \mathbb{E}\left[\left\|\mathbf{X}^0 - \underline{\mathbf{X}}^K\right\|^2\right] \tag{51} \\
&\leq \frac{16}{\gamma\widehat{\rho}}A_1 + 2\left(1 + \frac{4}{1-\eta^2}\right)\alpha^2 K^2 n B^2 \delta^{-1} \leq \frac{18}{\gamma\widehat{\rho}}A_1. \tag{52}
\end{aligned}$$

Summing up the above inequality for all $r = 0, 1, \ldots, T-1$, we obtain

$$\frac{1}{T}\sum_{r=0}^{T-1}\mathbb{E}\left[\left\|\mathbf{X}_\perp^{rK}\right\|^2\right] \leq \frac{18}{\gamma\widehat{\rho}}A_1. \tag{53}$$

Summing (22) with $\eta_1 = 1$ over $s \in \{0, 1, \ldots, K-2\}$, we derive that, for all $r \in [T+1]$,

$$\frac{1}{K}\sum_{s=1}^{K-1}\|\mathbf{X}_\perp^{rK+s}\|^2 \leq 2\|\mathbf{X}_\perp^{rK}\|^2 + 2K^2\alpha^2 n B^2 \delta^{-1}.$$

This implies

$$\frac{1}{K}\sum_{s=0}^{K-1}\|\mathbf{X}_\perp^{rK+s}\|^2 \leq 3\|\mathbf{X}_\perp^{rK}\|^2 + 2K^2\alpha^2 n B^2 \delta^{-1}.$$

Thus, we have

$$\frac{1}{TK}\sum_{t=0}^{TK-1}\mathbb{E}\|\mathbf{X}_\perp^t\|^2 = \frac{1}{T}\sum_{r=0}^{T-1}\frac{1}{K}\sum_{s=0}^{K-1}\mathbb{E}\|\mathbf{X}_\perp^{rK+s}\|^2 \leq \frac{1}{T}\sum_{r=0}^{T-1}\left(3\mathbb{E}\|\mathbf{X}_\perp^{rK}\|^2 + 2K^2\alpha^2 n B^2 \delta^{-1}\right) \leq \frac{56}{\gamma\widehat{\rho}}A_1.$$

This completes the proof. $\qquad\square$

To prove the convergence of our algorithm, we define an auxiliary sequence as follows

$$z^t = \overline{x}^t + \frac{\beta_1}{1-\beta_1}\left(\overline{x}^t - \overline{x}^{t-1}\right), \forall t \in [TK], \tag{54}$$

with $\overline{x}^{-1} = \overline{x}^0$. The lemma below shows the difference of two consecutive $z$-points.

**Lemma A.3** *Let $\{z^t\}$ be defined in (54). It holds that for all $t \in [TK]$,*

$$z^{t+1} - z^t = \frac{\beta_1}{1-\beta_1}\frac{\alpha}{n}\sum_{i=1}^n m_i^{t-1} \circ \left(\frac{1}{\sqrt{u_i^{t-2}+\delta}} - \frac{1}{\sqrt{u_i^{t-1}+\delta}}\right) - \frac{\alpha}{n}\sum_{i=1}^n \frac{g_i^t}{\sqrt{u_i^{t-1}+\delta}}, \tag{55}$$

*where $u_i^{-2} = \mathbf{0}$.*

*Proof.* By (12)–(16) and $(\mathbf{W}-\mathbf{I})\mathbf{J} = \mathbf{0}$, we have

$$\overline{x}^{t+1} = \overline{x}^t - \frac{\alpha}{n}\sum_{i=1}^n \frac{m_i^t}{\sqrt{u_i^{t-1}+\delta}}. \tag{56}$$

Thus by (54), we have

$$z^{t+1} - z^t = \overline{x}^{t+1} - \overline{x}^t + \frac{\beta_1}{1-\beta_1}\left(\overline{x}^{t+1} - \overline{x}^t\right) - \frac{\beta_1}{1-\beta_1}\left(\overline{x}^t - \overline{x}^{t-1}\right)$$

$$= \frac{1}{1-\beta_1}\left(\overline{x}^{t+1} - \overline{x}^t\right) - \frac{\beta_1}{1-\beta_1}\left(\overline{x}^t - \overline{x}^{t-1}\right)$$

$$= \frac{1}{1-\beta_1}\left(-\frac{\alpha}{n}\sum_{i=1}^n \frac{m_i^t}{\sqrt{u_i^{t-1}+\delta}}\right) - \frac{\beta_1}{1-\beta_1}\left(-\frac{\alpha}{n}\sum_{i=1}^n \frac{m_i^{t-1}}{\sqrt{u_i^{t-2}+\delta}}\right)$$

$$= \frac{1}{1-\beta_1}\left(-\frac{\alpha}{n}\sum_{i=1}^n \frac{\beta_1 m_i^{t-1} + (1-\beta_1)g_i^t}{\sqrt{u_i^{t-1}+\delta}}\right) - \frac{\beta_1}{1-\beta_1}\left(-\frac{\alpha}{n}\sum_{i=1}^n \frac{m_i^{t-1}}{\sqrt{u_i^{t-2}+\delta}}\right)$$

$$= \frac{\beta_1}{1-\beta_1}\frac{\alpha}{n}\sum_{i=1}^n m_i^{t-1} \circ \left(\frac{1}{\sqrt{u_i^{t-2}+\delta}} - \frac{1}{\sqrt{u_i^{t-1}+\delta}}\right) - \frac{\alpha}{n}\sum_{i=1}^n \frac{g_i^t}{\sqrt{u_i^{t-1}+\delta}},$$

which is the desired result. $\qquad\square$

**Lemma A.4** *Under Assumptions 1 and 3, it holds that for all $t \in [TK]$,*

$$\frac{1}{n}\sum_{i=1}^n \left\|\left(\nabla f_i\left(x_i^t\right) - \nabla f_i\left(\overline{x}^t\right)\right)\right\|^2 \le \frac{L^2}{n}\|\mathbf{X}_\perp^t\|^2, \tag{57}$$

*and*

$$\|\nabla f\left(z^t\right) - \nabla f(\overline{x}^t)\|^2 \le \frac{\alpha^2 L^2 \beta_1^2 B^2}{\delta(1-\beta_1)^2}. \tag{58}$$

*Proof.* First, by the $L$-smoothness of $f_i$ for each $i \in \{1, 2, \ldots, n\}$ and Young's inequality, we have

$$\frac{1}{n}\sum_{i=1}^n \left\|\left(\nabla f_i\left(x_i^t\right) - \nabla f_i\left(\overline{x}^t\right)\right)\right\|^2 \le \frac{L^2}{n}\sum_{i=1}^n \|x_i^t - \overline{x}^t\|^2,$$

which indicates (57) by the definition of $\mathbf{X}_{\perp}^t$. Also, by the $L$-smoothness of $f$, it follows

$$\|\nabla f\left(\boldsymbol{z}^t\right) - \nabla f(\overline{\boldsymbol{x}}^t)\|^2 \leq L^2\|\boldsymbol{z}^t - \overline{\boldsymbol{x}}^t\|^2 \overset{(54)}{=} \frac{L^2\beta_1^2}{(1-\beta_1)^2}\|\overline{\boldsymbol{x}}^t - \overline{\boldsymbol{x}}^{t-1}\|^2$$

$$\overset{(56)}{=} \frac{L^2\beta_1^2}{(1-\beta_1)^2}\left\|\frac{\alpha}{n}\sum_{i=1}^n \frac{\boldsymbol{m}_i^{t-1}}{\sqrt{\boldsymbol{u}_i^{t-2}+\delta}}\right\|^2 \leq \frac{L^2\beta_1^2}{(1-\beta_1)^2}\frac{\alpha^2}{n}\sum_{i=1}^n\left\|\frac{\boldsymbol{m}_i^{t-1}}{\sqrt{\boldsymbol{u}_i^{t-2}+\delta}}\right\|^2 \leq \frac{\alpha^2 L^2\beta_1^2 B^2}{\delta(1-\beta_1)^2},$$

where the last inequality holds by $\|\boldsymbol{m}_i^{t-1}\|\leq B$ from (18). This completes the proof. $\square$

**Lemma A.5** *Under Assumptions 1–3, it holds that*

$$\sum_{t=0}^{TK-1}\mathbb{E}\left[\left\|\boldsymbol{z}^{t+1}-\boldsymbol{z}^t\right\|^2\right]$$

$$\leq \frac{2\beta_1^2\alpha^2 B_\infty^2}{(1-\beta_1)^2}\mathbb{E}\left[\sum_{t=0}^{TK-1}\frac{1}{n}\sum_{i=1}^n\left\|\frac{1}{\sqrt{\boldsymbol{u}_i^{t-2}+\delta}}-\frac{1}{\sqrt{\boldsymbol{u}_i^{t-1}+\delta}}\right\|^2\right] \tag{59}$$

$$+2\alpha^2\left(\frac{24}{n\delta}TKB^2+\frac{6L^2}{n\delta}\mathbb{E}\left[\sum_{t=0}^{TK-1}\left\|\mathbf{X}_\perp^t\right\|^2\right]+\frac{6}{\delta}\mathbb{E}\left[\sum_{t=0}^{TK-1}\left\|\nabla f(\overline{\boldsymbol{x}}^t)\right\|^2\right]\right).$$

*Proof.* By (55) and Young's inequality, we have

$$\sum_{t=0}^{TK-1}\mathbb{E}\left[\left\|\boldsymbol{z}^{t+1}-\boldsymbol{z}^t\right\|^2\right]$$

$$\leq\mathbb{E}\left[2\sum_{t=0}^{TK-1}\left\|\frac{\beta_1}{1-\beta_1}\frac{\alpha}{n}\sum_{i=1}^n \boldsymbol{m}_i^{t-1}\circ\left(\frac{1}{\sqrt{\boldsymbol{u}_i^{t-2}+\delta}}-\frac{1}{\sqrt{\boldsymbol{u}_i^{t-1}+\delta}}\right)\right\|^2\right]$$

$$+\mathbb{E}\left[2\sum_{t=0}^{TK-1}\left\|\frac{\alpha}{n}\sum_{i=1}^n \frac{\boldsymbol{g}_i^t}{\sqrt{\boldsymbol{u}_i^{t-1}+\delta}}\right\|^2\right]. \tag{60}$$

To bound the first term in the RHS of (60), we obtain from (18) that

$$\sum_{t=0}^{TK-1}\left\|\frac{1}{n}\sum_{i=1}^n \boldsymbol{m}_i^{t-1}\circ\left(\frac{1}{\sqrt{\boldsymbol{u}_i^{t-2}+\delta}}-\frac{1}{\sqrt{\boldsymbol{u}_i^{t-1}+\delta}}\right)\right\|^2$$

$$\leq \sum_{t=0}^{TK-1}\frac{1}{n}\sum_{i=1}^n\left\|\boldsymbol{m}_i^{t-1}\circ\left(\frac{1}{\sqrt{\boldsymbol{u}_i^{t-2}+\delta}}-\frac{1}{\sqrt{\boldsymbol{u}_i^{t-1}+\delta}}\right)\right\|^2$$

$$\leq B_\infty^2\sum_{t=0}^{TK-1}\frac{1}{n}\sum_{i=1}^n\left\|\frac{1}{\sqrt{\boldsymbol{u}_i^{t-2}+\delta}}-\frac{1}{\sqrt{\boldsymbol{u}_i^{t-1}+\delta}}\right\|^2. \tag{61}$$

To bound the second term in RHS of (60), we have

$$2\mathbb{E}\left[\sum_{t=0}^{TK-1}\left\|\frac{1}{n}\sum_{i=1}^{n}\frac{\boldsymbol{g}_i^t}{\sqrt{\boldsymbol{u}_i^{t-1}+\delta}}\right\|^2\right]$$

$$\leq 2\mathbb{E}\left[\sum_{t=0}^{TK-1}\left\|\frac{1}{n}\sum_{i=1}^{n}\frac{\left(\boldsymbol{g}_i^t-\nabla f_i(\boldsymbol{x}_i^t)+\nabla f_i(\boldsymbol{x}_i^t)-\nabla f(\overline{\boldsymbol{x}}^t)+\nabla f(\overline{\boldsymbol{x}}^t)\right)}{\sqrt{\boldsymbol{u}_i^{t-1}+\delta}}\right\|^2\right]$$

$$\leq 6\mathbb{E}\left[\sum_{t=0}^{TK-1}\left\|\frac{1}{n}\sum_{i=1}^{n}\frac{\left(\boldsymbol{g}_i^t-\nabla f_i(\boldsymbol{x}_i^t)\right)}{\sqrt{\boldsymbol{u}_i^{t-1}+\delta}}\right\|^2+\sum_{t=0}^{TK-1}\left\|\frac{1}{n}\sum_{i=1}^{n}\frac{\left(\nabla f_i(\boldsymbol{x}_i^t)-\nabla f_i(\overline{\boldsymbol{x}}^t)\right)}{\sqrt{\boldsymbol{u}_i^{t-1}+\delta}}\right\|^2\right.$$

$$\left.+\sum_{t=0}^{TK-1}\left\|\frac{1}{n}\sum_{i=1}^{n}\frac{\nabla f(\overline{\boldsymbol{x}}^t)}{\sqrt{\boldsymbol{u}_i^{t-1}+\delta}}\right\|^2\right]$$

$$\leq 6\mathbb{E}\left[\sum_{t=0}^{TK-1}\left\|\frac{1}{n}\sum_{i=1}^{n}\frac{\left(\boldsymbol{g}_i^t-\nabla f_i(\boldsymbol{x}_i^t)\right)}{\sqrt{\boldsymbol{u}_i^{t-1}+\delta}}\right\|^2+\sum_{t=0}^{TK-1}\frac{1}{n}\sum_{i=1}^{n}\left\|\frac{\left(\nabla f_i(\boldsymbol{x}_i^t)-\nabla f_i(\overline{\boldsymbol{x}}^t)\right)}{\sqrt{\boldsymbol{u}_i^{t-1}+\delta}}\right\|^2\right.$$

$$\left.+\sum_{t=0}^{TK-1}\frac{1}{n}\sum_{i=1}^{n}\left\|\frac{\nabla f(\overline{\boldsymbol{x}}^t)}{\sqrt{\boldsymbol{u}_i^{t-1}+\delta}}\right\|^2\right]$$

$$=\frac{6}{n^2}\mathbb{E}\left[\sum_{t=0}^{TK-1}\sum_{i=1}^{n}\left\|\frac{\boldsymbol{g}_i^t-\nabla f_i(\boldsymbol{x}_i^t)}{\sqrt{\boldsymbol{u}_i^{t-1}+\delta}}\right\|^2\right]+6\mathbb{E}\left[\sum_{t=0}^{TK-1}\frac{1}{n}\sum_{i=1}^{n}\left\|\frac{\left(\nabla f_i(\boldsymbol{x}_i^t)-\nabla f_i(\overline{\boldsymbol{x}}^t)\right)}{\sqrt{\boldsymbol{u}_i^{t-1}+\delta}}\right\|^2\right]$$

$$+6\mathbb{E}\left[\sum_{t=0}^{TK-1}\frac{1}{n}\sum_{i=1}^{n}\left\|\frac{\nabla f(\overline{\boldsymbol{x}}^t)}{\sqrt{\boldsymbol{u}_i^{t-1}+\delta}}\right\|^2\right]$$

$$\leq\frac{6}{n^2\delta}\mathbb{E}\left[\sum_{t=0}^{TK-1}\sum_{i=1}^{n}\left\|\boldsymbol{g}_i^t-\nabla f_i(\boldsymbol{x}_i^t)\right\|^2\right]+\frac{6}{\delta}\mathbb{E}\left[\sum_{t=0}^{TK-1}\frac{1}{n}\sum_{i=1}^{n}\left\|\left(\nabla f_i(\boldsymbol{x}_i^t)-\nabla f_i(\overline{\boldsymbol{x}}^t)\right)\right\|^2\right]$$

$$+\frac{6}{\delta}\mathbb{E}\left[\sum_{t=0}^{TK-1}\frac{1}{n}\sum_{i=1}^{n}\left\|\nabla f(\overline{\boldsymbol{x}}^t)\right\|^2\right]$$

$$\leq\frac{6}{n^2\delta}4nTKB^2+\frac{6L^2}{n\delta}\mathbb{E}\left[\sum_{t=0}^{TK-1}\left\|\mathbf{X}_\perp^t\right\|^2\right]+\frac{6}{\delta}\mathbb{E}\left[\sum_{t=0}^{TK-1}\left\|\nabla f(\overline{\boldsymbol{x}}^t)\right\|^2\right],\tag{62}$$

where in the last inequality, we have used (57), and the equality holds because

$$\mathbb{E}_t\left\langle\frac{\left(\boldsymbol{g}_i^t-\nabla f_i(\boldsymbol{x}_i^t)\right)}{\sqrt{\boldsymbol{u}_i^{t-1}+\delta}},\frac{\left(\boldsymbol{g}_j^t-\nabla f_j(\boldsymbol{x}_j^t)\right)}{\sqrt{\boldsymbol{u}_j^{t-1}+\delta}}\right\rangle$$

$$=\left\langle\frac{\mathbb{E}_t[\boldsymbol{g}_i^t-\nabla f_i(\boldsymbol{x}_i^t)]}{\sqrt{\boldsymbol{u}_i^{t-1}+\delta}},\frac{\mathbb{E}_t[\boldsymbol{g}_j^t-\nabla f_j(\boldsymbol{x}_j^t)]}{\sqrt{\boldsymbol{u}_j^{t-1}+\delta}}\right\rangle=0,\ \forall i\neq j,$$

from the fact that $\boldsymbol{g}_1^t,\boldsymbol{g}_2^t,\ldots,\boldsymbol{g}_n^t$ are conditionally independent of each other. Plugging (61) and (62) into (60), we complete the proof. $\quad\square$

**Lemma A.6** *Suppose Assumptions 1 and 3 hold, and $\|\boldsymbol{u}_i^t\|_\infty \le B_u$ for all $t \ge 0$ and $i \in \{1, 2, \ldots, n\}$. It holds*

$$\mathbb{E}_t\left[\left\langle \nabla f\left(\boldsymbol{z}^t\right), \frac{1}{n}\sum_{i=1}^n \frac{\boldsymbol{g}_i^t}{\sqrt{\boldsymbol{u}_i^{t-1}+\delta}}\right\rangle\right] \tag{63}$$

$$\ge \frac{1}{2\sqrt{B_u+\delta}}\|\nabla f\left(\overline{\boldsymbol{x}}^t\right)\|^2 - \frac{L^2}{n}\left(\frac{\sqrt{B_u+\delta}}{\delta} + \frac{1}{2\sqrt{\delta}}\right)\|\mathbf{X}_\perp^t\|^2 - \frac{\alpha^2\beta_1^2 L^2 B^2}{\delta(1-\beta_1)^2}\left(\frac{\sqrt{B_u+\delta}}{\delta} + \frac{1}{2\sqrt{\delta}}\right).$$

*Proof.* By Assumption 3, it holds that

$$\mathbb{E}_t\left[\left\langle \nabla f\left(\boldsymbol{z}^t\right), \frac{1}{n}\sum_{i=1}^n \frac{\boldsymbol{g}_i^t}{\sqrt{\boldsymbol{u}_i^{t-1}+\delta}}\right\rangle\right] = \left\langle \nabla f\left(\boldsymbol{z}^t\right), \frac{1}{n}\sum_{i=1}^n \frac{\nabla f_i(\boldsymbol{x}_i^t)}{\sqrt{\boldsymbol{u}_i^{t-1}+\delta}}\right\rangle$$

$$= \left\langle \nabla f\left(\boldsymbol{z}^t\right) - \nabla f(\overline{\boldsymbol{x}}^t), \frac{1}{n}\sum_{i=1}^n \frac{\left(\nabla f_i(\boldsymbol{x}_i^t) - \nabla f(\overline{\boldsymbol{x}}^t)\right)}{\sqrt{\boldsymbol{u}_i^{t-1}+\delta}}\right\rangle$$

$$+ \left\langle \nabla f\left(\boldsymbol{z}^t\right) - \nabla f(\overline{\boldsymbol{x}}^t), \frac{1}{n}\sum_{i=1}^n \frac{\nabla f(\overline{\boldsymbol{x}}^t)}{\sqrt{\boldsymbol{u}_i^{t-1}+\delta}}\right\rangle$$

$$+ \left\langle \nabla f(\overline{\boldsymbol{x}}^t), \frac{1}{n}\sum_{i=1}^n \frac{\left(\nabla f_i(\boldsymbol{x}_i^t) - \nabla f(\overline{\boldsymbol{x}}^t)\right)}{\sqrt{\boldsymbol{u}_i^{t-1}+\delta}}\right\rangle + \left\langle \nabla f(\overline{\boldsymbol{x}}^t), \frac{1}{n}\sum_{i=1}^n \frac{\nabla f(\overline{\boldsymbol{x}}^t)}{\sqrt{\boldsymbol{u}_i^{t-1}+\delta}}\right\rangle. \tag{64}$$

Next we bound each of the four terms in the RHS of (64). For the first term in the RHS of (64), we use Young's inequality and (58) to have

$$\left\langle \nabla f\left(\boldsymbol{z}^t\right) - \nabla f(\overline{\boldsymbol{x}}^t), \frac{1}{n}\sum_{i=1}^n \frac{\left(\nabla f_i(\boldsymbol{x}_i^t) - \nabla f(\overline{\boldsymbol{x}}^t)\right)}{\sqrt{\boldsymbol{u}_i^{t-1}+\delta}}\right\rangle$$

$$\ge -\frac{1}{2n\sqrt{\delta}}\sum_{i=1}^n\left(\left\|\nabla f\left(\boldsymbol{z}^t\right) - \nabla f(\overline{\boldsymbol{x}}^t)\right\|^2 + \left\|\nabla f_i\left(\boldsymbol{x}_i^t\right) - \nabla f_i\left(\overline{\boldsymbol{x}}^t\right)\right\|^2\right)$$

$$\ge -\frac{1}{2n\sqrt{\delta}}\sum_{i=1}^n\left(\left\|\nabla f\left(\boldsymbol{z}^t\right) - \nabla f(\overline{\boldsymbol{x}}^t)\right\|^2 + L^2\left\|\boldsymbol{x}_i^t - \overline{\boldsymbol{x}}^t\right\|^2\right)$$

$$\overset{(58)}{\ge} -\frac{1}{2\sqrt{\delta}}\left(\frac{\alpha^2\beta_1^2 L^2 B^2}{\delta(1-\beta_1)^2} + \frac{L^2}{n}\|\mathbf{X}_\perp^t\|^2\right), \tag{65}$$

where we have used $\boldsymbol{u}_i^{t-1} \ge \mathbf{0}$ in the first inequality. For the second term in the RHS of (64), we have

$$\left\langle \nabla f\left(\boldsymbol{z}^t\right) - \nabla f(\overline{\boldsymbol{x}}^t), \frac{1}{n}\sum_{i=1}^n \frac{\nabla f(\overline{\boldsymbol{x}}^t)}{\sqrt{\boldsymbol{u}_i^{t-1}+\delta}}\right\rangle \tag{66}$$

$$\ge -\frac{\left\|\nabla f\left(\overline{\boldsymbol{x}}^t\right)\right\|^2}{4\sqrt{B_u+\delta}} - \frac{\sqrt{B_u+\delta}}{\delta}\left\|\nabla f\left(\boldsymbol{z}^t\right) - \nabla f(\overline{\boldsymbol{x}}^t)\right\|^2$$

$$\overset{(58)}{\ge} -\frac{\left\|\nabla f\left(\overline{\boldsymbol{x}}^t\right)\right\|^2}{4\sqrt{B_u+\delta}} - \frac{\alpha^2\beta_1^2 L^2 B^2\sqrt{B_u+\delta}}{\delta^2(1-\beta_1)^2}, \tag{67}$$

where the first inequality follows from Young's inequality and $\boldsymbol{u}_i^{t-1} \geq \boldsymbol{0}$. For the third term in the RHS of (64), we have from Young's inequality that

$$
\begin{aligned}
\left\langle \nabla f(\overline{\boldsymbol{x}}^t), \frac{1}{n} \sum_{i=1}^n \frac{(\nabla f_i(\boldsymbol{x}_i^t) - \nabla f(\overline{\boldsymbol{x}}^t))}{\sqrt{\boldsymbol{u}_i^{t-1} + \delta}} \right\rangle & \\
\geq \frac{1}{n} \sum_{i=1}^n & \left( -\frac{\left\| \nabla f\left(\overline{\boldsymbol{x}}^t\right) \right\|^2}{4\sqrt{B_u + \delta}} - \frac{\sqrt{B_u + \delta}}{\delta} \left\| \nabla f_i\left(\boldsymbol{x}_i^t\right) - \nabla f_i\left(\overline{\boldsymbol{x}}^t\right) \right\|^2 \right) \\
\geq \frac{1}{n} \sum_{i=1}^n & \left( -\frac{\left\| \nabla f\left(\overline{\boldsymbol{x}}^t\right) \right\|^2}{4\sqrt{B_u + \delta}} - \frac{L^2 \sqrt{B_u + \delta}}{\delta} \left\| \boldsymbol{x}_i^t - \overline{\boldsymbol{x}}^t \right\|^2 \right) \\
= - & \frac{\left\| \nabla f\left(\overline{\boldsymbol{x}}^t\right) \right\|^2}{4\sqrt{B_u + \delta}} - \frac{L^2 \sqrt{B_u + \delta}}{n\delta} \left\| \mathbf{X}_\perp^t \right\|^2.
\end{aligned}
\tag{68}
$$

Since $\|\boldsymbol{u}_i^t\|_\infty \leq B_u$ for all $t \geq 0$ and $i \in \{1, 2, \ldots, n\}$, the last term in the RHS of (64) can be bounded as

$$
\left\langle \nabla f\left(\overline{\boldsymbol{x}}^t\right), \frac{1}{n} \sum_{i=1}^n \frac{\nabla f\left(\overline{\boldsymbol{x}}^t\right)}{\sqrt{\boldsymbol{u}_i^{t-1} + \delta}} \right\rangle \geq \frac{1}{\sqrt{B_u + \delta}} \left\| \nabla f\left(\overline{\boldsymbol{x}}^t\right) \right\|^2.
\tag{69}
$$

Substituting (65)–(69) into (64) and rearranging terms yields the desired result. □

Now we are ready to show the main convergence result.

**Theorem A.1** *Suppose that Assumptions 1–3 hold, $\mathcal{Q}$ is an $\eta$-compression operator, and $\|\boldsymbol{u}_i^t\|_\infty \leq B_u$ for all $t \geq 0$ and $i \in \{1, 2, \ldots, n\}$. Let $\overline{C}$ denote the constant defined in (21), $\alpha, \gamma > 0$ satisfy*

$$
\alpha \leq \frac{\delta}{48 L \sqrt{B_u + \delta}}, \quad \gamma \leq \frac{(1-\rho)(1-\eta^2)}{100}.
\tag{70}
$$

*Then, it holds*

$$
\begin{aligned}
\frac{\alpha}{4\sqrt{B_u + \delta}} \sum_{t=0}^{TK-1} \mathbb{E}\left[ \left\| \nabla f\left(\overline{\boldsymbol{x}}^t\right) \right\|^2 \right] \leq{}& \mathbb{E}\left[ f\left(\boldsymbol{x}^0\right) - f^* \right] + \frac{\alpha TK}{8\sqrt{B_u + \delta}} \frac{\alpha^2 L^2 \beta_1^2 B^2}{\delta(1-\beta_1)^2} \\
& + \frac{\alpha \beta_1^2 B_\infty^2}{(1-\beta_1)^2} \left( 4\sqrt{B_u + \delta} + \alpha L \right) \mathbb{E}\left[ \sum_{t=0}^{TK-1} \frac{1}{n} \sum_{i=1}^n \left\| \frac{1}{\sqrt{\boldsymbol{u}_i^{t-2} + \delta}} - \frac{1}{\sqrt{\boldsymbol{u}_i^{t-1} + \delta}} \right\|^2 \right] \\
& + \alpha^2 L \left( \frac{24}{n\delta} TKB^2 + \frac{6L^2}{n\delta} \alpha^2 TnK^3 \overline{C} \right) \\
& + \frac{\alpha L^2}{n} \left( \frac{\sqrt{B_u + \delta}}{\delta} + \frac{1}{2\sqrt{\delta}} \right) \alpha^2 TnK^3 \overline{C} + \sum_{t=0}^{TK-1} \frac{\alpha^3 \beta_1^2 L^2 B^2}{\delta(1-\beta_1)^2} \left( \frac{\sqrt{B_u + \delta}}{\delta} + \frac{1}{2\sqrt{\delta}} \right).
\end{aligned}
\tag{71}
$$

*Proof.* By the $L$-smoothness of $f$, we have

$$
f\left(\boldsymbol{z}^{t+1}\right) \leq f\left(\boldsymbol{z}^t\right) + \left\langle \nabla f\left(\boldsymbol{z}^t\right), \boldsymbol{z}^{t+1} - \boldsymbol{z}^t \right\rangle + \frac{L}{2} \left\| \boldsymbol{z}^{t+1} - \boldsymbol{z}^t \right\|^2,
$$

which together with (55) gives

$$
\begin{aligned}
f\left(\boldsymbol{z}^{t+1}\right) \leq{}& f\left(\boldsymbol{z}^t\right) - \alpha \left\langle \nabla f\left(\boldsymbol{z}^t\right), \frac{1}{n} \sum_{i=1}^n \frac{\boldsymbol{g}_i^t}{\sqrt{\boldsymbol{u}_i^{t-1} + \delta}} \right\rangle + \frac{L}{2} \left\| \boldsymbol{z}^{t+1} - \boldsymbol{z}^t \right\|^2 \\
& + \frac{\beta_1}{1-\beta_1} \left\langle \nabla f\left(\boldsymbol{z}^t\right), \frac{\alpha}{n} \sum_{i=1}^n \boldsymbol{m}_i^{t-1} \circ \left( \frac{1}{\sqrt{\boldsymbol{u}_i^{t-2} + \delta}} - \frac{1}{\sqrt{\boldsymbol{u}_i^{t-1} + \delta}} \right) \right\rangle.
\end{aligned}
$$

Take expectation, sum up over $t$, and rearrange terms of the above inequality. Noticing $\boldsymbol{z}^0 = \boldsymbol{x}^0$, we have

$$
\alpha \sum_{t=0}^{TK-1} \mathbb{E}\left[\left\langle \nabla f\left(\boldsymbol{z}^t\right), \frac{1}{n}\sum_{i=1}^{n} \frac{\boldsymbol{g}_i^t}{\sqrt{\boldsymbol{u}_i^{t-1}+\delta}}\right\rangle\right] \leq \mathbb{E}\left[f\left(\boldsymbol{x}^0\right) - f\left(\boldsymbol{z}^{TK}\right)\right] + \frac{L}{2}\mathbb{E}\sum_{t=0}^{TK-1}\left[\left\|\boldsymbol{z}^{t+1}-\boldsymbol{z}^t\right\|^2\right]
$$
$$
+ \frac{\alpha\beta_1}{1-\beta_1}\sum_{t=0}^{TK-1}\mathbb{E}\left[\left\langle \nabla f\left(\boldsymbol{z}^t\right), \frac{1}{n}\sum_{i=1}^{n}\boldsymbol{m}_i^{t-1}\circ\left(\frac{1}{\sqrt{\boldsymbol{u}_i^{t-2}+\delta}} - \frac{1}{\sqrt{\boldsymbol{u}_i^{t-1}+\delta}}\right)\right\rangle\right]. \tag{72}
$$

Below we bound the inner-product terms on the RHS of (72). First,

$$
\frac{\beta_1}{1-\beta_1}\sum_{t=0}^{TK-1}\mathbb{E}\left[\left\langle \nabla f\left(\boldsymbol{z}^t\right), \frac{1}{n}\sum_{i=1}^{n}\boldsymbol{m}_i^{t-1}\circ\left(\frac{1}{\sqrt{\boldsymbol{u}_i^{t-2}+\delta}} - \frac{1}{\sqrt{\boldsymbol{u}_i^{t-1}+\delta}}\right)\right\rangle\right]
$$
$$
= \frac{\beta_1}{1-\beta_1}\sum_{t=0}^{TK-1}\mathbb{E}\left[\left\langle \nabla f\left(\overline{\boldsymbol{x}}^t\right), \frac{1}{n}\sum_{i=1}^{n}\boldsymbol{m}_i^{t-1}\circ\left(\frac{1}{\sqrt{\boldsymbol{u}_i^{t-2}+\delta}} - \frac{1}{\sqrt{\boldsymbol{u}_i^{t-1}+\delta}}\right)\right\rangle\right]
$$
$$
+ \frac{\beta_1}{1-\beta_1}\sum_{t=0}^{TK-1}\mathbb{E}\left[\left\langle \nabla f\left(\boldsymbol{z}^t\right) - \nabla f\left(\overline{\boldsymbol{x}}^t\right), \frac{1}{n}\sum_{i=1}^{n}\boldsymbol{m}_i^{t-1}\circ\left(\frac{1}{\sqrt{\boldsymbol{u}_i^{t-2}+\delta}} - \frac{1}{\sqrt{\boldsymbol{u}_i^{t-1}+\delta}}\right)\right\rangle\right]. \tag{73}
$$

For the first term in the RHS of (73), we use Young's inequality to have

$$
\frac{\beta_1}{1-\beta_1}\sum_{t=0}^{TK-1}\mathbb{E}\left[\left\langle \nabla f\left(\overline{\boldsymbol{x}}^t\right), \frac{1}{n}\sum_{i=1}^{n}\boldsymbol{m}_i^{t-1}\circ\left(\frac{1}{\sqrt{\boldsymbol{u}_i^{t-2}+\delta}} - \frac{1}{\sqrt{\boldsymbol{u}_i^{t-1}+\delta}}\right)\right\rangle\right]
$$
$$
\leq \sum_{t=0}^{TK-1}\frac{1}{8\sqrt{B_u+\delta}}\mathbb{E}\left[\left\|\nabla f\left(\overline{\boldsymbol{x}}^t\right)\right\|^2\right]
$$
$$
+ \sum_{t=0}^{TK-1}\frac{2\beta_1^2\sqrt{B_u+\delta}}{(1-\beta_1)^2}\mathbb{E}\left[\left\|\frac{1}{n}\sum_{i=1}^{n}\boldsymbol{m}_i^{t-1}\circ\left(\frac{1}{\sqrt{\boldsymbol{u}_i^{t-2}+\delta}} - \frac{1}{\sqrt{\boldsymbol{u}_i^{t-1}+\delta}}\right)\right\|^2\right]
$$
$$
\leq \frac{1}{8\sqrt{B_u+\delta}}\sum_{t=0}^{TK-1}\mathbb{E}\left[\left\|\nabla f\left(\overline{\boldsymbol{x}}^t\right)\right\|^2\right]
$$
$$
+ \sum_{t=0}^{TK-1}\frac{2\beta_1^2 B_\infty^2\sqrt{B_u+\delta}}{(1-\beta_1)^2}\frac{1}{n}\sum_{i=1}^{n}\mathbb{E}\left[\left\|\frac{1}{\sqrt{\boldsymbol{u}_i^{t-2}+\delta}} - \frac{1}{\sqrt{\boldsymbol{u}_i^{t-1}+\delta}}\right\|^2\right], \tag{74}
$$

where in the last inequality, we have used $\|\boldsymbol{m}_i^{t-1}\|_\infty \le B_\infty$ by Lemma A.1. For the second term in the RHS of (73), it holds

$$
\frac{\beta_1}{1-\beta_1} \sum_{t=0}^{TK-1} \mathbb{E}\left[\left\langle \nabla f\left(\boldsymbol{z}^t\right) - \nabla f\left(\overline{\boldsymbol{x}}^t\right), \frac{1}{n}\sum_{i=1}^n \boldsymbol{m}_i^{t-1} \circ \left(\frac{1}{\sqrt{\boldsymbol{u}_i^{t-2}+\delta}} - \frac{1}{\sqrt{\boldsymbol{u}_i^{t-1}+\delta}}\right)\right\rangle\right]
$$
$$
\le \sum_{t=0}^{TK-1} \frac{1}{8\sqrt{B_u+\delta}} \mathbb{E}\left[\left\|\nabla f\left(\boldsymbol{z}^t\right) - \nabla f\left(\overline{\boldsymbol{x}}^t\right)\right\|^2\right]
$$
$$
+ \sum_{t=0}^{TK-1} \frac{2\beta_1^2\sqrt{B_u+\delta}}{(1-\beta_1)^2} \mathbb{E}\left[\left\|\frac{1}{n}\sum_{i=1}^n \boldsymbol{m}_i^{t-1} \circ \left(\frac{1}{\sqrt{\boldsymbol{u}_i^{t-2}+\delta}} - \frac{1}{\sqrt{\boldsymbol{u}_i^{t-1}+\delta}}\right)\right\|^2\right]
$$
$$
\le \frac{TK}{8\sqrt{B_u+\delta}} \frac{\alpha^2 L^2 \beta_1^2 B^2}{\delta(1-\beta_1)^2} + \sum_{t=0}^{TK-1} \frac{2\beta_1^2 B_\infty^2 \sqrt{B_u+\delta}}{(1-\beta_1)^2} \frac{1}{n}\sum_{i=1}^n \mathbb{E}\left[\left\|\frac{1}{\sqrt{\boldsymbol{u}_i^{t-2}+\delta}} - \frac{1}{\sqrt{\boldsymbol{u}_i^{t-1}+\delta}}\right\|^2\right], \quad (75)
$$

where in the last inequality, we have used (58) and $\|\boldsymbol{m}_i^{t-1}\|_\infty \le B_\infty$ by Lemma A.1. Plugging (74) and (75) into (73), we obtain

$$
\frac{\alpha\beta_1}{1-\beta_1} \sum_{t=0}^{TK-1} \mathbb{E}\left[\left\langle \nabla f\left(\boldsymbol{z}^t\right), \frac{1}{n}\sum_{i=1}^n \boldsymbol{m}_i^{t-1} \circ \left(\frac{1}{\sqrt{\boldsymbol{u}_i^{t-2}+\delta}} - \frac{1}{\sqrt{\boldsymbol{u}_i^{t-1}+\delta}}\right)\right\rangle\right]
$$
$$
\le \frac{\alpha}{8\sqrt{B_u+\delta}} \sum_{t=0}^{TK-1} \mathbb{E}\left[\left\|\nabla f\left(\overline{\boldsymbol{x}}^t\right)\right\|^2\right] + \frac{\alpha TK}{8\sqrt{B_u+\delta}} \frac{\alpha^2 L^2 \beta_1^2 B^2}{\delta(1-\beta_1)^2}
$$
$$
+ \frac{4\alpha\beta_1^2 B_\infty^2 \sqrt{B_u+\delta}}{(1-\beta_1)^2} \sum_{t=0}^{TK-1} \frac{1}{n}\sum_{i=1}^n \mathbb{E}\left[\left\|\frac{1}{\sqrt{\boldsymbol{u}_i^{t-2}+\delta}} - \frac{1}{\sqrt{\boldsymbol{u}_i^{t-1}+\delta}}\right\|^2\right]. \quad (76)
$$

Now plugging (59), (76) and (63) after taking full expectation into (72) and rearranging terms gives

$$
\left(\frac{3\alpha}{8\sqrt{B_u+\delta}} - \frac{6L\alpha^2}{\delta}\right) \sum_{t=0}^{TK-1} \mathbb{E}[\|\nabla f\left(\overline{\boldsymbol{x}}^t\right)\|^2] \le \mathbb{E}\left[f\left(\boldsymbol{x}^0\right) - f\left(\boldsymbol{z}^{TK}\right)\right] + \frac{\alpha TK}{8\sqrt{B_u+\delta}} \frac{\alpha^2 L^2 \beta_1^2 B^2}{\delta(1-\beta_1)^2}
$$
$$
+ \frac{\alpha\beta_1^2 B_\infty^2}{(1-\beta_1)^2}\left(4\sqrt{B_u+\delta} + \alpha L\right) \mathbb{E}\left[\sum_{t=0}^{TK-1} \frac{1}{n}\sum_{i=1}^n \left\|\frac{1}{\sqrt{\boldsymbol{u}_i^{t-2}+\delta}} - \frac{1}{\sqrt{\boldsymbol{u}_i^{t-1}+\delta}}\right\|^2\right]
$$
$$
+ \alpha^2 L\left(\frac{24}{n\delta}TKB^2 + \frac{6L^2}{n\delta}\mathbb{E}\left[\sum_{t=0}^{TK-1} \|\mathbf{X}_\perp^t\|^2\right]\right)
$$
$$
+ \alpha \sum_{t=0}^{TK-1}\left(\frac{L^2}{n}\left(\frac{\sqrt{B_u+\delta}}{\delta} + \frac{1}{2\sqrt{\delta}}\right) \mathbb{E}[\|\mathbf{X}_\perp^t\|^2] + \frac{\alpha^2\beta_1^2 L^2 B^2}{\delta(1-\beta_1)^2}\left(\frac{\sqrt{B_u+\delta}}{\delta} + \frac{1}{2\sqrt{\delta}}\right)\right).
$$

Plug (21) into the inequality above, notice $\frac{3\alpha}{8\sqrt{B_u+\delta}} - \frac{6L\alpha^2}{\delta} \ge \frac{\alpha}{4\sqrt{B_u+\delta}}$, and rearrange terms. We obtain the desired result and complete the proof. $\qquad\square$

To prove Theorem 3.2, we only need to consider the following three settings of $\{\mathbf{U}^t\}$

$$\text{AMSGrad}: \ \widehat{\mathbf{U}}^t = \beta_2\widehat{\mathbf{U}}^{t-1} + (1-\beta_2)\mathbf{G}^t \circ \mathbf{G}^t \text{ with } \widehat{\mathbf{U}}^{-1} = \mathbf{0}, \ \mathbf{U}^t = \max\left\{\widehat{\mathbf{U}}^t, \mathbf{U}^{t-1}\right\}; \tag{77}$$

$$\text{Adam}: \ \mathbf{U}^t = \beta_2\mathbf{U}^{t-1} + (1-\beta_2)\mathbf{G}^t \circ \mathbf{G}^t; \tag{78}$$

$$\text{AdaGrad}: \ \mathbf{U}^t = \frac{1}{t+1}\sum_{s=0}^{t}\mathbf{G}^s \circ \mathbf{G}^s; \ , \tag{79}$$

where

$$\widehat{\mathbf{U}}^t = \left[\widehat{\boldsymbol{u}}_1^t, \widehat{\boldsymbol{u}}_2^t, \dots, \widehat{\boldsymbol{u}}_n^t\right], \ \mathbf{U}^t = \left[\boldsymbol{u}_1^t, \boldsymbol{u}_2^t, \dots, \boldsymbol{u}_n^t\right]. \tag{80}$$

Notice that when $\beta_1 = 0$ and $\beta_2 = 1$, AMSGrad reduces to the vanilla SGD, and when $\beta_1 \in (0, 1)$ and $\beta_2 = 1$, it reduces to the momentum SGD. Consequently, our theoretical guarantees on AMSGrad naturally extend to these two special cases as well. Adam-Mini (Zhang et al., 2024) can be regarded as a special case of Adam by using a constant scalar (instead of a vector) for each block of variables as the second momentum. Therefore, the results on Adam also hold for Adam-Mini.

Below we bound $\|\boldsymbol{u}_i^t\|$ and the summation of the difference between the consecutive terms in the sequence $\left\{\frac{1}{\sqrt{\mathbf{U}^t+\delta}}\right\}_{t=0}^{TK-1}$ for the three optimizers in (77)-(79).

**Lemma A.7** *Let $\boldsymbol{u}_i^{-2} = \mathbf{0}$ for all $i = 1, 2, \dots, n$. Under Assumption 3, for all $t \geq 0$ and $i \in \{1, 2, \dots, n\}$, the following statements hold.*

*(i) For AMSGrad in (77), it holds $\|\boldsymbol{u}_i^t\|_\infty \leq B_\infty^2$, and*

$$\sum_{t=0}^{TK-1}\frac{1}{n}\sum_{i=1}^{n}\left\|\frac{1}{\sqrt{\boldsymbol{u}_i^{t-2}+\delta}} - \frac{1}{\sqrt{\boldsymbol{u}_i^{t-1}+\delta}}\right\|^2 \leq \frac{d}{\delta}. \tag{81}$$

*(ii) For Adam in (78), it holds $\|\boldsymbol{u}_i^t\|_\infty \leq B_\infty^2$, and*

$$\sum_{t=0}^{TK-1}\frac{1}{n}\sum_{i=1}^{n}\left\|\frac{1}{\sqrt{\boldsymbol{u}_i^{t-2}+\delta}} - \frac{1}{\sqrt{\boldsymbol{u}_i^{t-1}+\delta}}\right\|^2 \leq \frac{TKd(1-\beta_2)^2 B_\infty^4}{\delta^3}. \tag{82}$$

*(iii) For Adagrad in (79), it holds $\|\boldsymbol{u}_i^t\|_\infty \leq B_\infty^2$, and*

$$\sum_{t=0}^{TK-1}\frac{1}{n}\sum_{i=1}^{n}\left\|\frac{1}{\sqrt{\boldsymbol{u}_i^{t-2}+\delta}} - \frac{1}{\sqrt{\boldsymbol{u}_i^{t-1}+\delta}}\right\|^2 \leq \frac{2dB_\infty^4}{\delta^3}. \tag{83}$$

*Proof.* (i) Noticing $\widehat{\boldsymbol{u}}_i^{-1} = \mathbf{0}$ and $\|\boldsymbol{g}_i^t \circ \boldsymbol{g}_i^t\|_\infty \leq B_\infty^2$, we have $\|\widehat{\boldsymbol{u}}_i^t\|_\infty \leq \left(1-\beta_2^{t+1}\right)B_\infty^2$ for each $i \in \{1, 2, \dots, n\}$ and $t \geq 0$. By $\boldsymbol{u}_i^t = \max\{\widehat{\boldsymbol{u}}_i^t, \boldsymbol{u}_i^{t-1}\}$, it holds

$$
\begin{aligned}
\|\boldsymbol{u}_i^t\|_\infty &\leq \max\{\|\widehat{\boldsymbol{u}}_i^t\|_\infty, \|\boldsymbol{u}_i^{t-1}\|_\infty\} \leq \max\{\|\widehat{\boldsymbol{u}}_i^t\|_\infty, \|\widehat{\boldsymbol{u}}_i^{t-1}\|_\infty, \|\boldsymbol{u}_i^{t-2}\|_\infty\} \\
&\leq \max\left\{\|\widehat{\boldsymbol{u}}_i^t\|_\infty, \|\widehat{\boldsymbol{u}}_i^{t-1}\|_\infty, \dots, \|\widehat{\boldsymbol{u}}_i^0\|_\infty, \|\boldsymbol{u}_i^{-1}\|_\infty\right\}, \\
&\leq \max\left\{\left(1-\beta_2^{t+1}\right)B_\infty^2, \left(1-\beta_2^t\right)B_\infty^2, \dots, (1-\beta_2)B_\infty^2, \|\boldsymbol{u}_i^{-1}\|_\infty\right\} = \left(1-\beta_2^{t+1}\right)B_\infty^2,
\end{aligned}
$$

where the equality holds because $\beta_2 \in (0, 1]$ and $\boldsymbol{u}_i^{-1} = \mathbf{0}$.

In addition, we have

$$
\begin{aligned}
&\sum_{t=0}^{TK-1} \frac{1}{n} \sum_{i=1}^{n} \left\| \frac{1}{\sqrt{\boldsymbol{u}_i^{t-2}+\delta}} - \frac{1}{\sqrt{\boldsymbol{u}_i^{t-1}+\delta}} \right\|^2 \\
&\leq \sum_{t=0}^{TK-1} \frac{1}{n} \sum_{i=1}^{n} \left\| \frac{1}{\sqrt{\boldsymbol{u}_i^{t-2}+\delta}} - \frac{1}{\sqrt{\boldsymbol{u}_i^{t-1}+\delta}} \right\|_1 \left\| \frac{1}{\sqrt{\boldsymbol{u}_i^{t-2}+\delta}} - \frac{1}{\sqrt{\boldsymbol{u}_i^{t-1}+\delta}} \right\|_\infty \\
&\leq \sum_{t=0}^{TK-1} \frac{1}{n} \sum_{i=1}^{n} \frac{1}{\sqrt{\delta}} \left\| \frac{1}{\sqrt{\boldsymbol{u}_i^{t-2}+\delta}} - \frac{1}{\sqrt{\boldsymbol{u}_i^{t-1}+\delta}} \right\|_1 \\
&\leq \sum_{t=0}^{TK-1} \frac{1}{n\sqrt{\delta}} \sum_{i=1}^{n} \left( \left\| \frac{1}{\sqrt{\boldsymbol{u}_i^{t-2}+\delta}} \right\|_1 - \left\| \frac{1}{\sqrt{\boldsymbol{u}_i^{t-1}+\delta}} \right\|_1 \right) \\
&= \frac{1}{n\sqrt{\delta}} \sum_{i=1}^{n} \sum_{t=0}^{TK-1} \left( \left\| \frac{1}{\sqrt{\boldsymbol{u}_i^{t-2}+\delta}} \right\|_1 - \left\| \frac{1}{\sqrt{\boldsymbol{u}_i^{t-1}+\delta}} \right\|_1 \right) \\
&\leq \frac{1}{n\sqrt{\delta}} \sum_{i=1}^{n} \left\| \frac{1}{\sqrt{\boldsymbol{u}_i^{-2}+\delta}} \right\|_1 = \frac{d}{\delta},
\end{aligned}
\tag{84}
$$

where $\left\| \frac{1}{\sqrt{\boldsymbol{u}_i^{t-2}+\delta}} - \frac{1}{\sqrt{\boldsymbol{u}_i^{t-1}+\delta}} \right\|_\infty \leq \frac{1}{\sqrt{\delta}}$ holds because $\boldsymbol{u}_i^{t-2} \geq \boldsymbol{0}$ and $\boldsymbol{u}_i^{t-1} \geq \boldsymbol{0}$, and the equality holds because $\boldsymbol{u}_i^{t-1}$ is nondecreasing with $t$ for each $i \in \{1,2,\ldots,n\}$.

(ii) Noticing $\boldsymbol{u}_i^{-1} = \boldsymbol{0}$ and $\|\boldsymbol{g}_i^t \circ \boldsymbol{g}_i^t\|_\infty \leq B_\infty^2$, we have $\|\boldsymbol{u}_i^t\|_\infty \leq \left(1 - \beta_2^{t+1}\right) B_\infty^2$.

For all $t \geq -1$, it holds

$$
\begin{aligned}
&\left\| \frac{1}{\sqrt{\boldsymbol{u}_i^t+\delta}} - \frac{1}{\sqrt{\boldsymbol{u}_i^{t-1}+\delta}} \right\|^2 = \sum_{j=1}^{d} \left| \frac{1}{\sqrt{[\beta_2 \boldsymbol{u}_i^{t-1} + (1-\beta_2)\mathbf{g}_i^t \circ \mathbf{g}_i^t]_j + \delta}} - \frac{1}{\sqrt{[\boldsymbol{u}_i^{t-1}+\delta]_j}} \right|^2 \\
&= \sum_{j=1}^{d} \left| \frac{(1-\beta_2)\left([\boldsymbol{u}_i^{t-1} - \mathbf{g}_i^t \circ \mathbf{g}_i^t]_j\right)}{\sqrt{[\beta_2 \boldsymbol{u}_i^{t-1} + (1-\beta_2)\mathbf{g}_i^t \circ \mathbf{g}_i^t]_j + \delta}\sqrt{[\boldsymbol{u}_i^{t-1}+\delta]_j}\left(\sqrt{[\beta_2 \boldsymbol{u}_i^{t-1} + (1-\beta_2)\mathbf{g}_i^t \circ \mathbf{g}_i^t]_j + \delta} + \sqrt{[\boldsymbol{u}_i^{t-1}+\delta]_j}\right)} \right|^2 \\
&\leq \frac{d(1-\beta_2)^2 B_\infty^4}{\delta^3},
\end{aligned}
$$

where the last inequality follows from $0 \leq [\boldsymbol{u}_i^{t-1}]_j \leq B_\infty^2$ and $0 \leq [\mathbf{g}_i^t \circ \mathbf{g}_i^t]_j \leq B_\infty^2$. Then the desired inequality holds.

(iii) By $\boldsymbol{u}_i^t = \frac{1}{t+1}\sum_{s=0}^{t} \boldsymbol{g}_i^s \circ \boldsymbol{g}_i^s$ and $\|\boldsymbol{g}_i^t \circ \boldsymbol{g}_i^t\|_\infty \leq B_\infty^2$, it holds $\|\boldsymbol{u}_i^t\|_\infty \leq B_\infty^2$. For all $t \geq 1$, it holds

$$
\begin{aligned}
&\left\| \frac{1}{\sqrt{\boldsymbol{u}_i^{t-1}+\delta}} - \frac{1}{\sqrt{\boldsymbol{u}_i^{t-2}+\delta}} \right\|^2 = \sum_{j=1}^{d} \left| \frac{1}{\sqrt{[\frac{t-1}{t}\boldsymbol{u}_i^{t-2} + \frac{1}{t}\mathbf{g}_i^{t-1} \circ \mathbf{g}_i^{t-1}]_j + \delta}} - \frac{1}{\sqrt{[\boldsymbol{u}_i^{t-2}]_j + \delta}} \right|^2 \\
&= \sum_{j=1}^{d} \left| \frac{\frac{1}{t}\left([\boldsymbol{u}_i^{t-2} - \mathbf{g}_i^{t-1} \circ \mathbf{g}_i^{t-1}]_j\right)}{\sqrt{[\frac{t-1}{t}\boldsymbol{u}_i^{t-2} + \frac{1}{t}\mathbf{g}_i^{t-1} \circ \mathbf{g}_i^{t-1}]_j + \delta}\sqrt{[\boldsymbol{u}_i^{t-2}]_j + \delta}\left(\sqrt{[\frac{t-1}{t}\boldsymbol{u}_i^{t-2} + \frac{1}{t}\mathbf{g}_i^{t-1} \circ \mathbf{g}_i^{t-1}]_j + \delta} + \sqrt{[\boldsymbol{u}_i^{t-2}]_j + \delta}\right)} \right|^2 \\
&\leq \frac{d B_\infty^4}{t^2 \delta^3},
\end{aligned}
$$

where the last inequality follows from $0 \le [\boldsymbol{u}_i^{t-2}]_j \le B_\infty^2$ and $0 \le [\mathbf{g}_i^{t-1} \circ \mathbf{g}_i^{t-1}]_j \le B_\infty^2$. Then

$$
\sum_{t=0}^{TK-1} \frac{1}{n} \sum_{i=1}^n \left\| \frac{1}{\sqrt{\boldsymbol{u}_i^{t-2}+\delta}} - \frac{1}{\sqrt{\boldsymbol{u}_i^{t-1}+\delta}} \right\|^2 = \sum_{t=1}^{TK-1} \frac{1}{n} \sum_{i=1}^n \left\| \frac{1}{\sqrt{\boldsymbol{u}_i^{t-2}+\delta}} - \frac{1}{\sqrt{\boldsymbol{u}_i^{t-1}+\delta}} \right\|^2
$$
$$
\le \frac{2dB_\infty^4}{\delta^3}.
$$

The proof is then completed. $\qquad \square$

Now, we prove Theorem 3.2, with its complete statement given as follows.

**Theorem A.2** *Suppose that Assumptions 1–3 hold, and $\mathcal{Q}$ is an $\eta$-compression operator. Let $\delta = O(1)$ be a universal positive constant, $\overline{C}$ be the constant defined in (21), and $\alpha, \gamma > 0$ satisfy*

$$
\alpha = \frac{4\theta\sqrt{n(B_\infty^2+\delta)}}{\sqrt{TK}} \le \min\left\{ \frac{\delta}{48L\sqrt{B_\infty^2+\delta}}, 1 \right\}, \gamma \le \frac{(1-\rho)(1-\eta^2)}{100}, \tag{85}
$$

*where $\theta = O(1)$. Then, the following statements hold.*

(i) *For AMSGrad in (77), it holds*

$$
\frac{1}{TK} \sum_{t=0}^{TK-1} \mathbb{E}\left[ \|\nabla f(\overline{\boldsymbol{x}}^t)\|^2 + \frac{1}{n}\|\mathbf{X}_\perp^t\|^2 \right]
$$
$$
= O\left( \frac{1}{\sqrt{nTK}} \left( f(\boldsymbol{x}^0) - f^* + LB^2B_\infty^2 + LB^2 \right) + \frac{1}{TK} dB_\infty^3 (B_\infty + L + 1) \right. \tag{86}
$$
$$
\left. + \frac{nK\overline{C}}{T} L^2 B_\infty^3 (1 + L + B_\infty) + \frac{n}{TK} L^2 B^2 (1 + B_\infty^4) + \frac{nK\overline{C}}{T} (1 + B_\infty^2) \right).
$$

(ii) *For Adam in (78) with $\beta_2 \in \left[ \frac{\sqrt{TK}}{\sqrt{TK}+1}, 1 \right]$, it holds*

$$
\frac{1}{TK} \sum_{t=0}^{TK-1} \mathbb{E}\left[ \|\nabla f(\overline{\boldsymbol{x}}^t)\|^2 + \frac{1}{n}\|\mathbf{X}_\perp^t\|^2 \right]
$$
$$
= O\left( \frac{1}{\sqrt{nTK}} \left( f(\boldsymbol{x}^0) - f^* + LB^2B_\infty^2 + LB^2 \right) + \frac{1}{TK} dB_\infty^7 (B_\infty + L + 1) \right. \tag{87}
$$
$$
\left. + \frac{nK\overline{C}}{T} L^2 B_\infty^3 (1 + L + B_\infty) + \frac{n}{TK} L^2 B^2 (1 + B_\infty^4) + \frac{nK\overline{C}}{T} (1 + B_\infty^2) \right).
$$

(iii) *For AdaGrad in (79), the relation (87) holds as well.*

*Proof.* From Lemma A.7, it holds that $\|\boldsymbol{u}_i^t\|_\infty \le B_u, \forall t, \forall i$ with $B_u = B_\infty^2$ for all the three optimizers in (77)-(79).

Dividing both sides of (71) by $\frac{\alpha TK}{4\sqrt{B_\infty^2+\delta}} = \theta\sqrt{nTK}$ and rearranging terms, we have

$$
\frac{1}{TK}\sum_{t=0}^{TK-1}\mathbb{E}\left[\|\nabla f\left(\overline{\boldsymbol{x}}^t\right)\|^2\right] \le \frac{1}{\theta\sqrt{nTK}}\left(f(\boldsymbol{x}^0)-f^*\right)+\frac{\alpha^2 L^2\beta_1^2 B^2}{2\delta(1-\beta_1)^2}
$$

$$
+\frac{4\beta_1^2 B_\infty^2\sqrt{B_\infty^2+\delta}}{TK(1-\beta_1)^2}\left(4\sqrt{B_\infty^2+\delta}+\alpha L\right)\mathbb{E}\left[\sum_{t=0}^{TK-1}\frac{1}{n}\sum_{i=1}^{n}\left\|\frac{1}{\sqrt{\boldsymbol{u}_i^{t-2}+\delta}}-\frac{1}{\sqrt{\boldsymbol{u}_i^{t-1}+\delta}}\right\|^2\right]
$$

$$
+4\alpha L\sqrt{B_\infty^2+\delta}\frac{24}{n\delta}B^2+4\sqrt{B_\infty^2+\delta}\frac{\alpha^2 L^2 nK^2\overline{C}}{n}\left(\frac{\sqrt{B_\infty^2+\delta}}{\delta}+\frac{1}{2\sqrt{\delta}}+\frac{6\alpha L}{\delta}\right)
$$

$$
+4\sqrt{B_\infty^2+\delta}\left(\frac{\alpha^2\beta_1^2 L^2 B^2}{\delta(1-\beta_1)^2}\left(\frac{\sqrt{B_\infty^2+\delta}}{\delta}+\frac{1}{2\sqrt{\delta}}\right)\right).
$$

Adding (21) to the above inequality and replacing $\alpha$ by $\frac{4\theta\sqrt{n}\sqrt{B_\infty^2+\delta}}{\sqrt{TK}} \le 1$ in the resulting inequality, we obtain

$$
\frac{1}{TK}\sum_{t=0}^{TK-1}\mathbb{E}\left[\|\nabla f\left(\overline{\boldsymbol{x}}^t\right)\|^2+\frac{1}{n}\|\mathbf{X}_\perp^t\|^2\right] \le \frac{1}{\theta\sqrt{nTK}}\left(f(\boldsymbol{x}^0)-f^*\right)+\frac{16\theta^2 nL^2\beta_1^2 B^2(B_\infty^2+\delta)}{2TK\delta(1-\beta_1)^2}
$$

$$
+\frac{4\beta_1^2 B_\infty^2\sqrt{B_\infty^2+\delta}}{TK(1-\beta_1)^2}\left(4\sqrt{B_\infty^2+\delta}+L\right)\mathbb{E}\left[\sum_{t=0}^{TK-1}\frac{1}{n}\sum_{i=1}^{n}\left\|\frac{1}{\sqrt{\boldsymbol{u}_i^{t-2}+\delta}}-\frac{1}{\sqrt{\boldsymbol{u}_i^{t-1}+\delta}}\right\|^2\right]\tag{88}
$$

$$
+\frac{16\theta}{\sqrt{nTK}}L(B_\infty^2+\delta)\frac{24}{\delta}B^2+\frac{64L^2 nK^2\overline{C}\theta^2(B_\infty^2+\delta)^{\frac{3}{2}}}{TK}\left(\frac{\sqrt{B_\infty^2+\delta}}{\delta}+\frac{1}{2\sqrt{\delta}}+\frac{6L}{\delta}\right)
$$

$$
+\frac{64n\theta^2(B_\infty^2+\delta)^{\frac{3}{2}}}{TK}\left(\frac{\beta_1^2 L^2 B^2}{\delta(1-\beta_1)^2}\left(\frac{\sqrt{B_\infty^2+\delta}}{\delta}+\frac{1}{2\sqrt{\delta}}\right)\right)+\frac{16nK^2\overline{C}\theta^2(B_\infty^2+\delta)}{TK}.
$$

We now substitute the results in Lemma A.7 to the above inequality.

(i) For AMSGrad, we have

$$
\mathbb{E}\left[\sum_{t=0}^{TK-1}\frac{1}{n}\sum_{i=1}^{n}\left\|\frac{1}{\sqrt{\boldsymbol{u}_i^{t-2}+\delta}}-\frac{1}{\sqrt{\boldsymbol{u}_i^{t-1}+\delta}}\right\|^2\right] \le \frac{d}{\delta}=O(d).
$$

(ii) For Adam, we know from $\beta_2 \in \left[\frac{\sqrt{TK}}{\sqrt{TK}+1},1\right]$ that

$$
\mathbb{E}\left[\sum_{t=0}^{TK-1}\frac{1}{n}\sum_{i=1}^{n}\left\|\frac{1}{\sqrt{\boldsymbol{u}_i^{t-2}+\delta}}-\frac{1}{\sqrt{\boldsymbol{u}_i^{t-1}+\delta}}\right\|^2\right] \le \frac{TKd(1-\beta_2)^2 B_\infty^4}{\delta^3}=O(dB_\infty^4).
$$

(iii) For Adagrad, we have

$$
\sum_{t=0}^{TK-1}\frac{1}{n}\sum_{i=1}^{n}\left\|\frac{1}{\sqrt{\boldsymbol{u}_i^{t-2}+\delta}}-\frac{1}{\sqrt{\boldsymbol{u}_i^{t-1}+\delta}}\right\|^2 \le \frac{2dB_\infty^4}{\delta^3}=O(dB_\infty^4).
$$

Therefore, we obtain the desired results. $\qquad\square$

# B  The matrix-form adaptive gradient updates

It should be noted that our theoretical results extend to matrix-form adaptive gradient updates, where the $d$-dimension real-valued functions $\{r_t\}$ are replace by some $d \times d$ dimension real-valued functions $\{\boldsymbol{r}_t\}$. Under this formulation, we update the second-momentum matrices $\mathbf{U}_i$ as $\mathbf{U}_i^t = \boldsymbol{r}_t(\boldsymbol{g}_i^0, \boldsymbol{g}_i^1, \dots, \boldsymbol{g}_i^t)$ and the model parameter by $\boldsymbol{x}_i^{t+\frac{1}{2}} = \boldsymbol{x}_i^t - \alpha \left(\mathbf{U}_i^{t-1} + \delta\right)^{-\frac{1}{2}} \boldsymbol{m}_i^t$.

Our theoretical results apply to the matrix-form adaptive gradient method, provided that $\max_{r,s}\|[\mathbf{U}_i^t]_{rs}\|$ is uniformly bounded for all $i \in \{1, 2, \dots, n\}$ and $t \geq 0$, and that the summation $\sum_{t=0}^{TK-1} \frac{1}{n} \sum_{i=1}^{n} \left\|(\mathbf{U}_i^{t-2} + \delta)^{-\frac{1}{2}} - (\mathbf{U}_i^{t-1} + \delta)^{-\frac{1}{2}}\right\|^2$ remains bounded.

A notable example of this framework is the matrix-form AdaGrad method, where $\mathbf{U}_i^t$ is updated as

$$\mathbf{U}_i^t = \frac{1}{t+1} \sum_{s=0}^{t} \boldsymbol{g}_i \boldsymbol{g}_i^\top, \text{ for each agent } i = 1, \dots, n.$$

For this choice of $\{\mathbf{U}_i^t\}$, it is not difficult to show that $|[\mathbf{U}_i^t]_{rs}| \leq B_\infty^2$ for all $r \in \{1, 2, \dots, d\}$ and $s \in \{1, 2, \dots, d\}$, and

$$\sum_{t=0}^{TK-1} \frac{1}{n} \sum_{i=1}^{n} \left\|(\mathbf{U}_i^{t-2} + \delta)^{-\frac{1}{2}} - (\mathbf{U}_i^{t-1} + \delta)^{-\frac{1}{2}}\right\|^2 \leq \frac{2d^2 B_\infty^4}{\delta^3}. \tag{89}$$

Therefore, our theoretical results extend naturally to the matrix-form AdaGrad method.

# C  Examples of $\eta$-compression operators

In this section, we provide a few concrete examples of compression operators that are $\eta$-compression operators. More examples can be found in (Chen et al., 2023a; Koloskova et al., 2019).

**Example C.1** *QSGD (Alistarh et al., 2017) compresses $\boldsymbol{x} \in \mathbb{R}^d$ by $\mathcal{Q}_{sgd}(\boldsymbol{x}) = \frac{\text{sign}(\boldsymbol{x})\|\boldsymbol{x}\|}{s} \left\lfloor s \frac{|\boldsymbol{x}|}{\|\boldsymbol{x}\|} + \xi \right\rfloor$ where $\xi$ is uniformly distributed on $[0,1]^d$, $s$ is a parameter about compression level. Then $\mathcal{Q}(\boldsymbol{x}) := \frac{1}{\tau} \mathcal{Q}_{ssgd}(\boldsymbol{x})$ with $\tau = \left(1 + \min\left\{d/s^2, \sqrt{d}/s\right\}\right)$ is an $\eta$-compression operator with $\eta = 1 - \frac{1}{\tau}$.*

**Example C.2** *$\mathcal{Q}_{sparse}(\boldsymbol{x})$ (Stich et al., 2018) randomly selects $k$ out of $d$ coordinates from $\boldsymbol{x}$, or the $k$ coordinates with the largest values in magnitude from $\boldsymbol{x}$. Then $\mathcal{Q}_{sparse}(\boldsymbol{x})$ is an $\eta$-compression operator with $\eta = \frac{d-k}{d}$.*

**Example C.3** *$\mathcal{Q}_{gossip}(\boldsymbol{x})$ (Koloskova et al., 2019) sets $\mathcal{Q}_{gossip}(\boldsymbol{x}) = \boldsymbol{x}$ with probability $p \in [0,1]$ and $\mathcal{Q}_{gossip}(\boldsymbol{x}) = 0$ with probability $1 - p$. Then $\mathcal{Q}_{gossip}(\boldsymbol{x})$ is an $\eta$-compression operator with $\eta = 1 - p$.*

# D  Additional Numerical Experiments

We include a suite of additional numerical results in this section. These results were omitted from the main body of the paper for space considerations. We expand on the results presented in the main body in several ways. Figure 6 includes all omitted FashionMNIST results using the same experiment setup as shown in Figures 1, 2, and 3 for CIFAR-10 and tiny-shakespeare. However, we note that we omit CDProxSGT optimizer comparisons in this supplement, as its results were not competitive in most experiments, as observed in Figure 1. SQuARM-SGD otherwise represents methods with non-adaptive updates.

We also include additional experiments using expanded parameter settings. Figure 7 repeats the optimizer comparisons demonstrated in Figure 1 with a wider variety of local update counts. Figure 8 repeats the experiments comparing the effect of local update counts on communication rounds shown in Figure 2 with Adam's update, while comparing AdaGrad's and AMSGrad's adaptive updates as well. Figure 9 compares

varying values of Top-$k$ compression for all three optimizer variants. We do not show training loss or consensus error with these figures for space and clarity, while noting that theses plots would otherwise be consistent with those shown in Figures 1, 2, 3, and 6.

## D.1 FashionNMIST Results

In Figure 6, we plot the results for the same experiments on FashionNMIST as performed and displayed for CIFAR-10 and tiny-shakespeare in Figures 1, 2, 3 in the main body of the paper. For these results, Top-$k$ compression of 30% is used. The primary observations in this figure are consistent as with the prior results. We generally observe little to no degradation of accuracy and loss performance, even when including local updates and Top-$k$ compression. In particular, FashionMNIST suffers less in terms of quality impacts when including compression and minimizing communication relative the two other test datasets.

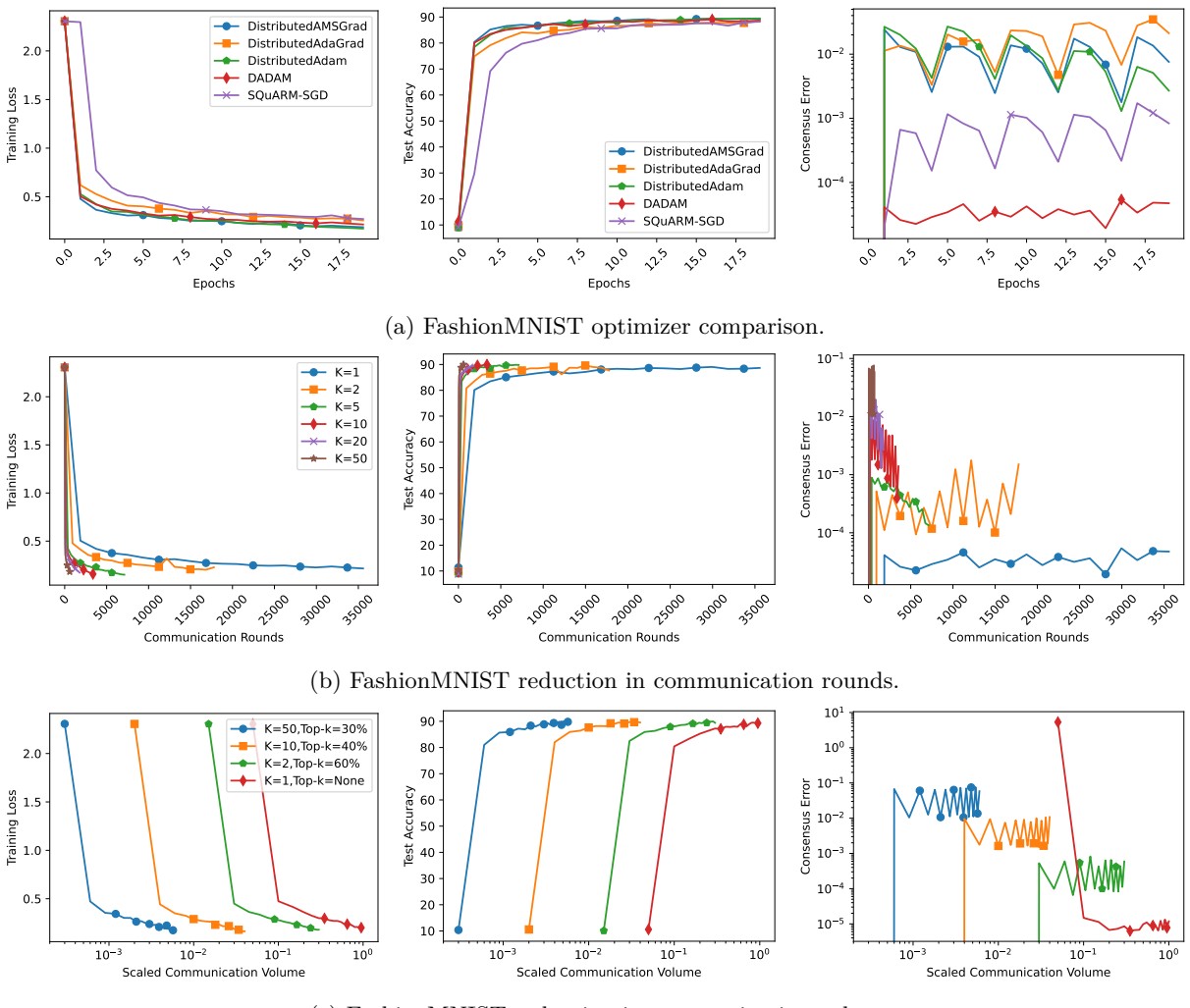

(a) FashionMNIST optimizer comparison.

(b) FashionMNIST reduction in communication rounds.

(c) FashionMNIST reduction in communication volume.

Figure 6: **Convergence performance for FashionMNIST:** Plotted above are the training loss and test accuracy of FashionMNIST. The top row compares optimizer performance with Top-$k$ 30% compression and a local update count of $K = 20$. The middle row demonstrates the reduction in communication rounds based on the number of local updates with Top-$k$ compression of 30%. The bottom row compares the total communication volume scaled relative to the uncompressed baseline with no local updates.

### D.2 Additional Optimizer Comparisons

In the main body of the paper, Figures 1a and 1b compare optimizer performance for a fixed value of $K = 20$. Figure 7 repeats these experiments with additional values of $K = 2, 5, 10, 50$, still using 4 agents. Shown are test accuracy/validation loss with compression values of 30%, 40%, and 50% for FashionMNIST, CIFAR-10, and tiny-shakespeare, respectively. We overall again observe consistent results as with Figure 1, where Adam outperforms other optimizers on FashionMNIST and tiny-shakespeare and AdaGrad outperforms other optimizers on CIFAR-10. Likewise, SQuARM-SGD consistently lacks in generalization performance on the GPT language model with the tiny-shakespeare dataset.

### D.3 Additional Number of Local Updates Comparisons

In Figure 8, the experiments using Adam in Figures 2a, 2b, and 6b are repeated with the AdaGrad and AMSGrad adaptive updates. For each optimizer and benchmark dataset, we plot the accuracy or validation loss that results with local updates of $K = 1, 2, 5, 10, 20, 50$. Again, we use Top-$k$ compression values of 30%, 40%, and 50% for FashionMNIST, CIFAR-10, and tiny-shakespeare. Likewise, we note that accuracy performance is minimally affected by the number of local updates used in these tests.

### D.4 Additional Top-$k$ Compression Comparisons

We fixed Top-$k$ compression values for each dataset for the bulk of the experiments demonstrated thus far, to avoid a parametric explosion in the number of results presented. Figure 9 demonstrates that a different choice of Top-$k$ compression in the same order of what was used has minimal impact on the overall optimizer performance and resultant appearance in plots. In Figure 9, we fix $K = 20$ and vary Top-$k$ to 30%, 40%, 50%, and 60% for each of our optimizers (Adam, AdaGrad, AMSGrad) and dataset (FashionMNIST, CIFAR-10, tiny-shakespeare). We observe a reduction in total used communication volume scaling linearly with Top-$k$ percentages, as expected, while accuracy and validation loss are relatively consistent across all tests.

### D.5 Additional Scaling Comparisons

In Figure 10 we present scaling results for AdaGrad and AMSGrad, as was presented for Adam in Figure 4 in the main body of this paper. We fix $K = 20$ and run with 4, 9, and 16 agents across both ring and 2D grid topologies. As with Adam, we note minimal impact to the maximum achieved test accuracy after 250 epochs, demonstrating near-linear speedup for our approach.

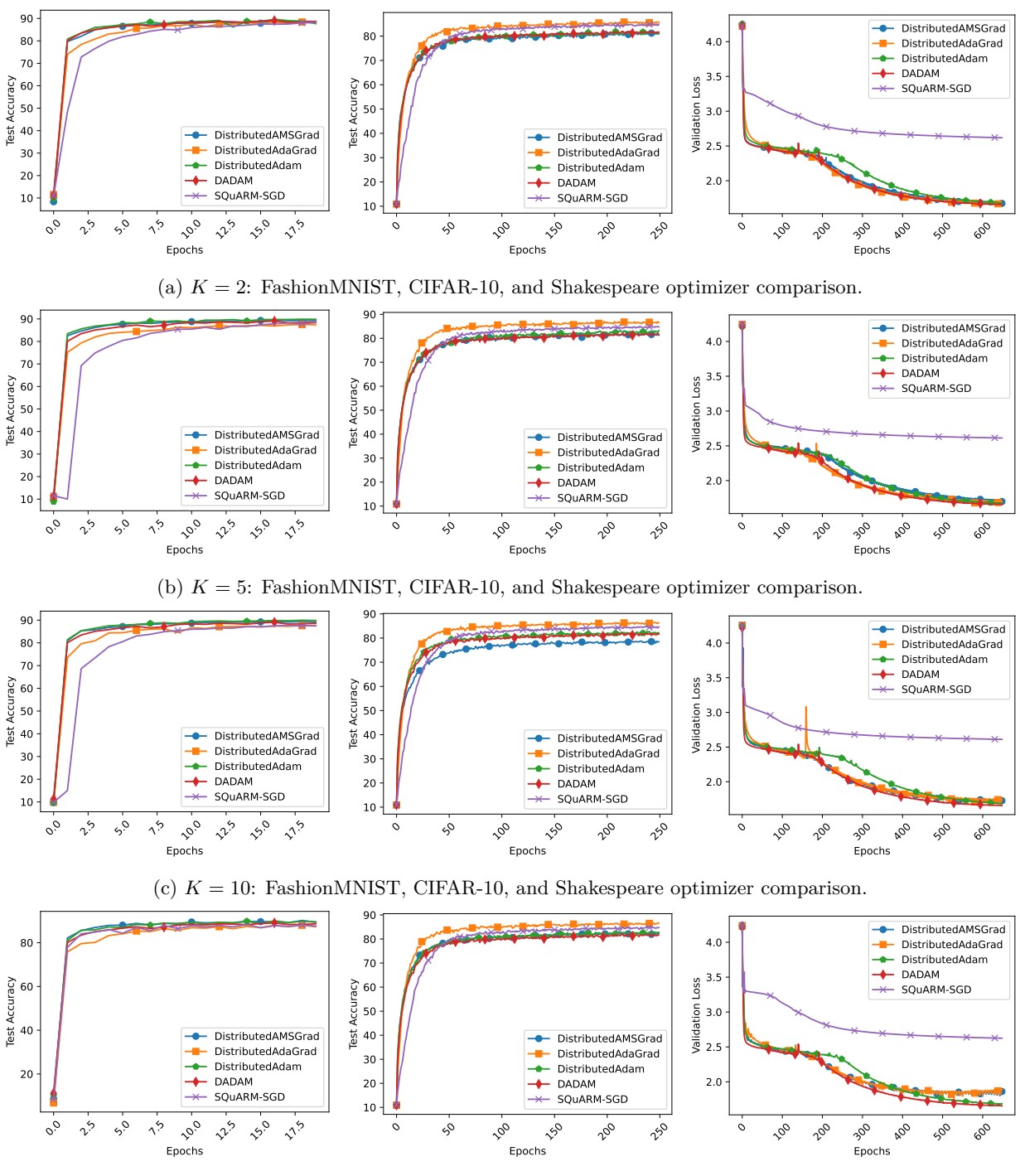

(a) $K = 2$: FashionMNIST, CIFAR-10, and Shakespeare optimizer comparison.

(b) $K = 5$: FashionMNIST, CIFAR-10, and Shakespeare optimizer comparison.

(c) $K = 10$: FashionMNIST, CIFAR-10, and Shakespeare optimizer comparison.

(d) $K = 50$: FashionMNIST, CIFAR-10, and Shakespeare optimizer comparison.

Figure 7: **Additional optimizer comparisons:** Plotted above are accuracy (for FashionMNIST and CIFAR-10) and validation loss (for tiny-shakespeare) for the tested optimizers across a range of local update counts $K$. For each subplot, FashionMNIST is on the right, CIFAR-10 is in the middle, and tiny-shakespeare is on the right.

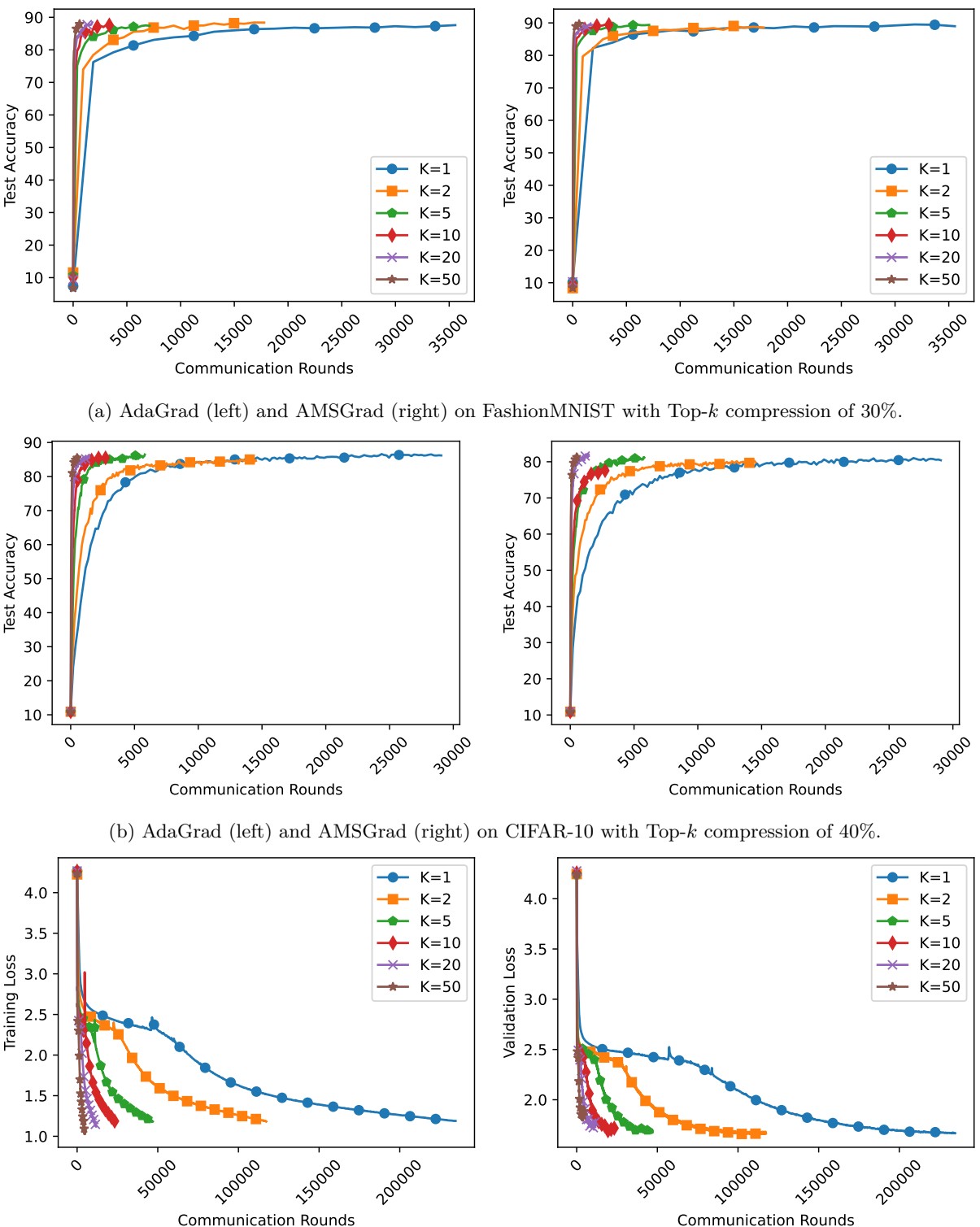

(a) AdaGrad (left) and AMSGrad (right) on FashionMNIST with Top-$k$ compression of 30%.

(b) AdaGrad (left) and AMSGrad (right) on CIFAR-10 with Top-$k$ compression of 40%.

(c) AdaGrad (left) and AMSGrad (right) on tiny-shakespeare with Top-$k$ compression of 50%.

Figure 8: **Additional number of local updates comparisons:** Plotted above are accuracy (for Fashion-MNIST and CIFAR-10) and validation loss (for tiny-shakespeare) using AdaGrad (left) and AMSGrad (right), comparing across a range of local update values $K$.

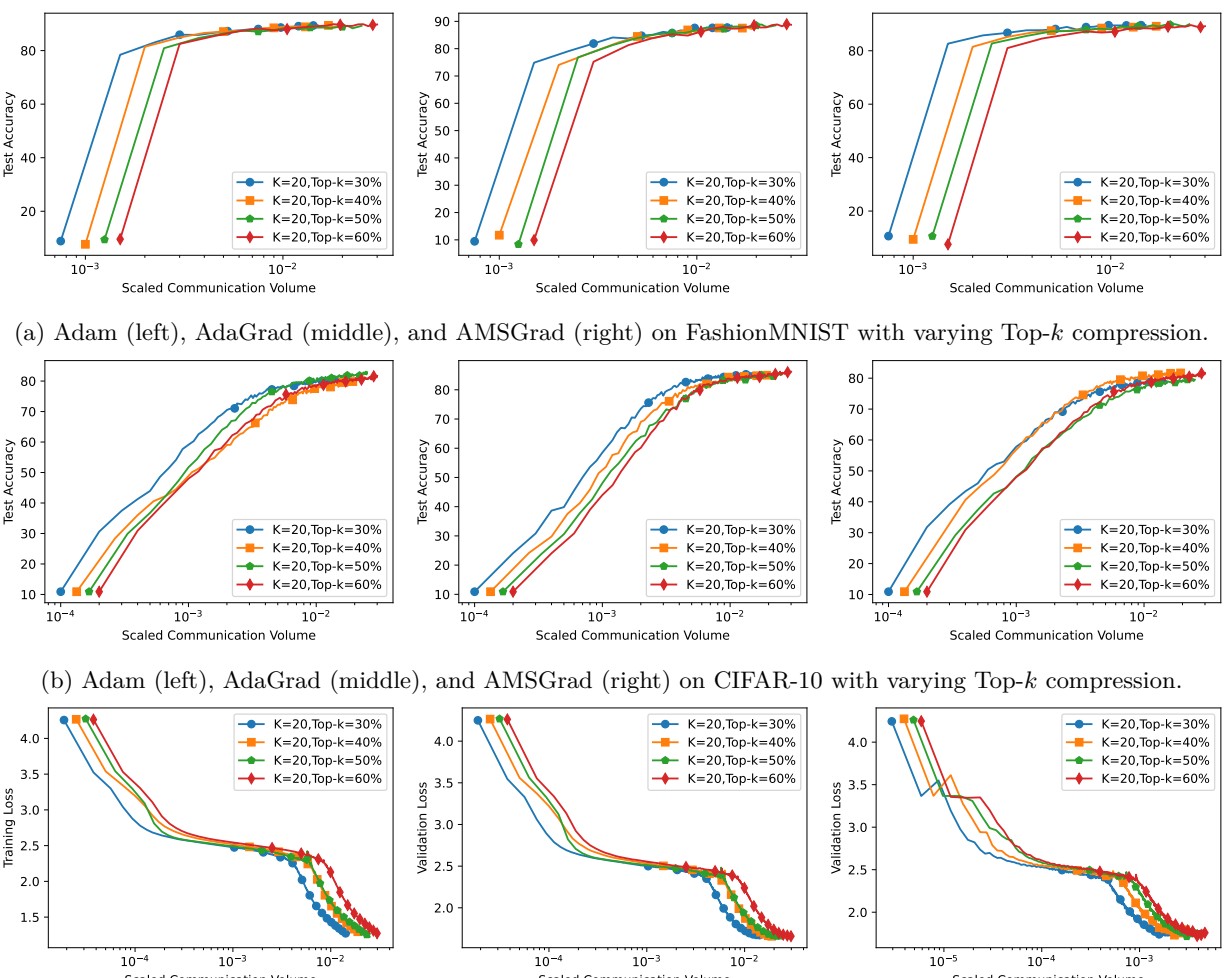

(a) Adam (left), AdaGrad (middle), and AMSGrad (right) on FashionMNIST with varying Top-$k$ compression.

(b) Adam (left), AdaGrad (middle), and AMSGrad (right) on CIFAR-10 with varying Top-$k$ compression.

(c) Adam (left), AdaGrad (middle), and AMSGrad (right) on tiny-shakespeare with varying Top-$k$ compression.

Figure 9: **Additional Top-$k$ compression comparisons:** Plotted above are accuracy (for FashionM-NIST and CIFAR-10) and validation loss (for tiny-shakespeare) using Adam (left), AdaGrad (middle), and AMSGrad (right), comparing across a range of Top-$k$ compression values.

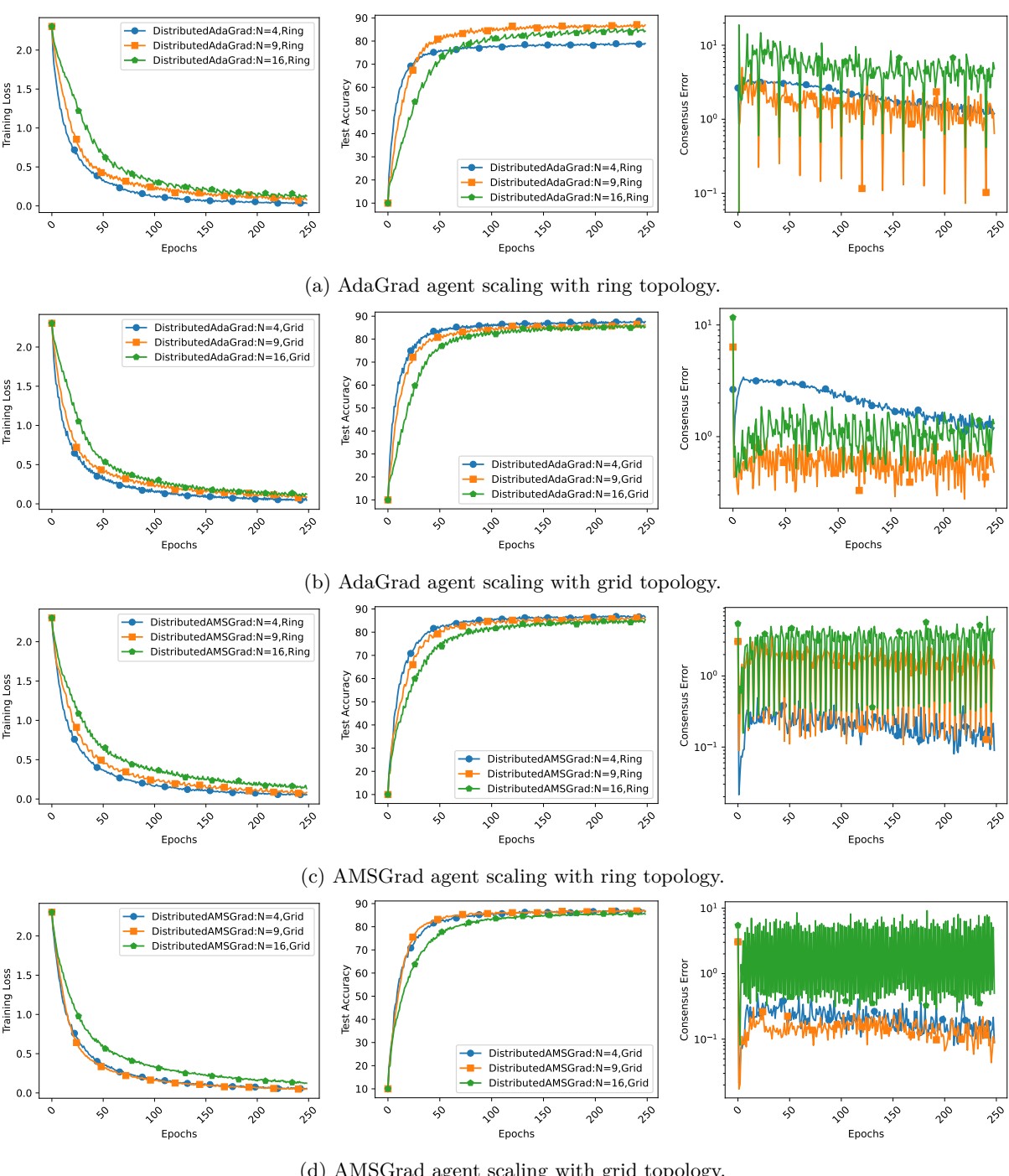

(a) AdaGrad agent scaling with ring topology.

(b) AdaGrad agent scaling with grid topology.

(c) AMSGrad agent scaling with ring topology.

(d) AMSGrad agent scaling with grid topology.

Figure 10: **Additional Scaling Results:** Plotted above are the training loss, test accuracy, and consensus error of CIFAR-10 when scaling to 4, 9, and 16 agents using ring topology and 2D grid topology with the AdaGrad and AMSGrad optimizers. All experiments were run with $K = 20$ local updates per communication round.

