# OpenReview forum: "LoDAdaC: a unified local training-based decentralized framework with adaptive gradients and compressed communication"
_TMLR — Accepted by TMLR_

### Review · Reviewer_r63W · 2025-12-08

**Summary Of Contributions:**

This paper introduces LoDAdaC, a decentralized learning framework that combines Adam-style adaptive optimization, multiple local updates, and compressed communication. By bringing these three ideas together, previously studied mostly in isolation, the method aims to reduce communication cost while keeping fast convergence. The framework supports a broad set of adaptive optimizers (e.g., AdaGrad, Adam, AMSGrad) and standard compressors, including biased ones, in a decentralized setting.

On the theory side, the paper gives a convergence analysis for general nonconvex problems with biased compressors, obtaining an iteration complexity of $O(1/\epsilon^4)$ with explicit dependence on the number of local steps $K$ and the compression ratio $\eta$. Empirically, LoDAdaC performs well on image tasks and Transformer-based language modeling, achieving up to 99% reduction in communication while outperforming momentum-based baselines such as SQuARM-SGD and matching or improving over DADAM.

**Audience:**

Yes

**Audience Explanation:**

This paper targets core challenges in distributed learning: reducing communication overhead while maintaining fast convergence. As research increasingly focuses on training Transformer-based models over bandwidth-constrained networks, frameworks that successfully unify gradient compression with adaptive optimization are highly relevant. Furthermore, the extension of Adam-type updates to the decentralized setting, supported by rigorous complexity analysis, addresses both a theoretical gap and a practical need for researchers working with architectures where adaptive methods are required for stability.

**Broader Impact Concerns:**

No.

**Claims And Evidence:**

Yes

**Claims Explanation:**

The authors provide a mix of theoretical and empirical evidence to support their claims.

1, Theoretical Convergence: They provide a rigorous convergence analysis showing that LoDAdaC achieves an asymptotic convergence rate comparable to state-of-the-art non-adaptive methods ($O(1/\epsilon^4)$), while explicating the dependence on the number of local steps ($K$) and compression ratio ($\eta$).

2, Empirical Validation: Experiments on CIFAR-10, Fashion-MNIST, and Tiny Shakespeare (using a small-scale GPT model) clearly show that the proposed method converges in fewer communication rounds and with less data transfer than baselines like SQuARM-SGD and DADAM.

3, Ablation Studies: The paper includes parameter studies on local step counts and compression rates, supporting the claim that the method is robust across different configurations. While the bounded-gradient assumption is strong and simplifies recent adaptive optimization theory, it is still common in decentralized analyses. The empirical results appear reproducible and distinct enough to validate the core claims of efficiency.

**Requested Changes:**

Major Concerns:

1, Could you clarify why the NanoGPT experiment ($\approx$ 10.7M parameters on Tiny Shakespeare) is described as “large language model training”? This model is several orders of magnitude smaller than what is typically meant by “LLM” today (usually ≥ billions of parameters), so the current wording risks overstating the scale. I recommend replacing “large language model(s)” / “LLM(s)” in the abstract (“e.g., in large language model training”), Introduction (LLMs as the “workhorse” optimizer), and Section 4 with more accurate terms such as “Transformer-based language models” or “small-scale GPT-style language models”, and explicitly acknowledging that the experiments are conducted on a small model.

2, In Definition 1.1, you introduce an η-compression operator and give examples such as Top-$k$ sparsification, which is biased. Could you clarify whether the convergence analysis relies only on the contractive-error property in Definition 1.1, or whether any extra conditions (e.g., unbiasedness/error-feedback behavior) are implicitly needed for such biased compressors? A brief explicit statement here would make the compressor assumptions and the scope of the theory much clearer.

3, In Section 3 you introduce Assumption 3 (bounded gradients) and describe it as “standard” for adaptive SGMs. However, while this assumption is indeed common, it is quite strong and often violated in practice (e.g., heavy-tailed gradients in deep Transformers), and recent analyses of Adam-type methods already work under weaker conditions such as generalized smoothness / bounded variance instead of bounded gradients. Could you soften the wording around “standard” and add a brief discussion that (i) acknowledges Assumption 3 as a strong, often unrealistic assumption, (ii) mentions practical remedies such as gradient clipping, and (iii) situates your analysis relative to these more recent Adam results under weaker assumptions?

4, Could you clarify how you expect LoDAdaC to behave under non-IID (heterogeneous) data, given that all current experiments use homogeneously distributed training data? Real decentralized/federated deployments typically face client drift due to heterogeneity, especially when combining local steps and adaptive updates. It would strengthen the empirical section to include at least one non-IID experiment (e.g., a Dirichlet or label-skewed partition on CIFAR-10); if that is not feasible, I recommend adding a short statement in the problem setup or conclusion acknowledging that the current validation is limited to homogeneous data and briefly discussing how LoDAdaC might interact with client drift and possible remedies.

Minor Concerns:

1, Section 1.1, “LoDAdaC equipped with adaptive gradient updates significantly outperform the baseline…” should be “outperforms”.

2, In the mid of page 5, “siginificantly” should be “significantly”.

3, In the mid of page 11, “Adamini” should be consistently written as “Adam-Mini”.

---

> ### Author Response · Authors · 2026-02-06
>
> $\textbf{Major Concern 1.}$
>
>  Agreed. We will soften the wording around "LLM".
>
> $\textbf{Major Concern 2.}$
>
>  Our convergence analysis relies on the contractive-error property in Definition 1.1 and does NOT require extra conditions. We will clarify that the convergence analysis relies only on the contractive-error property in Definition 1.1.
>
> $\textbf{Major Concern 3.}$
>
>  Agreed. We will soften the wording around "standard'' and explicitly acknowledge that bounded gradients can be unrealistic in modern deep learning
>  (e.g., heavy-tailed gradients in Transformers).
>
> Planned revision (paragraph after Assumption 3):
>  	Assumption 3 is a commonly used technical condition in prior analyses of Adam-type methods in decentralized settings. We acknowledge it can be restrictive
>  	for modern deep networks; in practice, gradient clipping is frequently used to control gradient magnitudes. Extending the present decentralized MLT+CC
>  	analysis to weaker conditions (e.g., generalized smoothness or bounded variance) is an important direction for future work.
>
> $\textbf{Major Concern 4.}$
>
> Agreed.
> We will add a numerical experiment on the non-IID (heterogeneous) data. We ran some limited number of non-IID experiments during this rebuttal period, the results of which can be viewed in our 'Additional Experiments' document at <https://anonymous.4open.science/r/TMLR2-38CE/reviewer_experiments.pdf>.
>
> $\textbf{Minor Concerns 1-3.}$
>
> We will correct all noted typos.

---

### Review · Reviewer_eRb6 · 2026-01-07

**Summary Of Contributions:**

This paper considers a decentralized optimization framework for distributed learning coupled with multiple local updates, compressed communication and adaptive gradient methods, which is close to practical applications and significant for efficient training of large models. Both theoretical results and numerical experiments are provided to validate the advantages of their proposed algorithm LoDAdaC.

**Strengths:**
- This paper states the motivation behind decentralized optimization and the concepts of multiple local updates and compressed communication in a clear and well-structured manner.
- The communication cost $O(\frac{1-\eta}{K\epsilon^2})$ derived for LoDAdaC shows the advantage of the combination and the explicit quantitative control by the parameters of MLC and CC.
- The theoretical results are validated by intensive experiments with LeNet, Resnet, nanoGPT on image or text datasets to show the effects of communications and local updates.

**Weaknesses:**
- Some points in the assumptions and definitions require further clarification and detail.
  - The optimization condition in expression (2): $X = XW$. Is this a typo? It would be beneficial to specify the dimensions of $W$ at the outset for clarity.
  - In Assumption 2, although authors have mentioned construction exist to meet assumption 2, more intuitive insights on the roles of these conditions in the proof should be demonstrated here. The same idea also applies to Definition 1.2.
  - For Definition 1.1, the condition $\mathbb{E}_{\mathcal{Q}} [\||x-\mathcal{Q}\||^2] \leq \eta^2 ||x||^2$ takes the expectation over the set of compression operators, however there's no description about the distribution of $\mathcal{Q}$.
  - Add more captions for (e)(f) in Figure 1: I think Top-k selection is related with compression ratio $\eta$. If it is, please make the connection more clearly. I don't quite get the idea behind (e) and (f).

**Audience:**

Yes

**Audience Explanation:**

The paper explores a decentralized optimization framework that closely aligns with practical applications, particularly in the context of training strategies for language models. The authors also present experiments using small-scale nanoGPT models trained on the Tiny Shakespeare dataset. This study makes contributions to both optimization theory and LLM communities.

**Broader Impact Concerns:**

No ethical concerns.

**Claims And Evidence:**

Yes

**Claims Explanation:**

The theoretical results are supported by detailed proofs and validated by intensive experiments.

**Requested Changes:**

See in Weakness.

---

> ### Author Response · Authors · 2026-02-06
>
> $\textbf{Weakness 1.}$
>
> $X = XW$ is not a typo. Here $X\in\mathbb{R}^{d\times n}$ stacks local models column-wise and $W\in\mathbb{R}^{n\times n}$
> is the mixing matrix. The constraint $X=XW$ characterizes the $\text{\emph{consensus subspace}}$; under Assumption 2(iii), it implies the consensus condition $x_1=\cdots=x_n$.
>
> Planned revision (text insertion after (2)):
> 	Here $X\in\mathbb{R}^{d\times n}$ and $W\in\mathbb{R}^{n\times n}$. Under Assumption 2(iii), the constraint $X=XW$
> 	is equivalent to requiring $x_1=\cdots=x_n$, i.e., $X$ lies in the consensus subspace.
>
> $\textbf{Weakness 2.}$
>
> Agreed. We will expand the explanation of Assumption 2. Intuitively: (i) Double stochasticity ensures mixing preserves the network average. (ii) Sparsity pattern encodes local communication (neighbor-only mixing). (iii) Spectral gap ($\rho=\|W-J\|_2<1$) yields contraction toward consensus and drives the consensus-error recursion.
>
> We will also expand the explanation of Definition 1.2. In decentralized nonconvex optimization, a meaningful solution must achieve $\textbf{both}$ stationarity (for the averaged iterate) and agreement (consensus) across agents.
>
> Planned revision (text insertion after Def. 1.2): This notion jointly controls stationarity of the averaged model and network disagreement; both must be small for decentralized learning to be practically meaningful.
>
> $\textbf{Weakness 3.}$ Thank you---we will clarify that the expectation is taken over the compressor's internal randomness (conditioned on the input).
> For deterministic compressors (e.g., Top-$k$), the inequality holds deterministically (the expectation becomes trivial).
>
> Planned revision (text insertion near Def. 1.1):
> 	The expectation is taken with respect to the internal randomness of the compressor (conditioned on the input $x$).
>
> $\textbf{Weakness 4.}$
>
> Yes, Top-k selection is related with compression ratio $\eta$. Roughly speaking, for a dense vector, keeping only k% largest entries in magnitude will compress the vector to k%. Since largest (in magnitude) entries are kept, we have $\eta \le 1 - k/100$. If the entries of the vector decay rapidly, then $\eta$ will be small.

---

### Review · Reviewer_KdrP · 2026-02-01

**Summary Of Contributions:**

This work proposed a novel approach that combines adaptive optimization algorithms with compressed communication for decentralized distributed learning. Theoretical guarantees and empirical results are provided to justify the effectiveness of this approach.

Strength:
- The paper is well-presented.
- Theoretical results look sound (attention: I did not check all the proofs, but for some that I did read, there is no evident error).
- Empirical experiments are chosen and presented properly.

Weakness:
- Compared to previous works, the contribution seems incremental. Especially, compared to SQuARM-SGD, the only difference is the local update: one uses Momentum-based methods, while the other uses Adaptive methods.
- Some terminologies are inconsistent: e.g., the letter A in LoDAaC, sometimes refers to Adam-type (in the abstract), sometimes refers to adaptive (the second sentence in 1.1. Contributions).
- It is always better to compare algorithms with multiple baselines if possible.

**Audience:**

Yes

**Audience Explanation:**

The paper might interest the distributed learning community.

**Broader Impact Concerns:**

Not available

**Claims And Evidence:**

Yes

**Claims Explanation:**

Theoretical guarantees are well-supported by adequate mathematical arguments. While the baselines in the paper are quite limited (only SQuARM-SGD and DADAM), empirical results do support their claims.

**Requested Changes:**

If the paper is accepted, I recommend that the authors should:
- Publicly provide their implementation of experiments for reproducible purposes.
- I think it is better to replace Adam-type by adaptive in the title. Adam is an instance of adaptive optimization methods.
- Table 2: I am not sure if Adagrad is defined as in the paper. Normally, the running sum is not averaged.
- It is interesting to report the consensus error in the experiments. I am curious whether all the agents in the experiments converge to the same solution or not.

---

> ### Author Response · Authors · 2026-02-06
>
> $\textbf{Weakness 1.}$
>
> We respectfully disagree that the difference is limited to swapping momentum for adaptivity.
> While LoDAdaC shares the general goals of leveraging multiple local steps (MLT) and compressed communication (CC),
> our contribution is a $\text{\emph{unified decentralized framework and analysis}}$ that simultaneously supports:
> (i) multiple local steps, (ii) compressed communication $\text{\emph{including biased compressors}}$, and (iii) a broad class
> of adaptive updates (AdaGrad/Adam/AMSGrad/Adam-Mini, including matrix-form variants), under one proof template.
>
> Beyond the update rule, key distinctions include:
>
> (i) Unified adaptive-family coverage. Our Algorithm 1 and theorems apply to multiple adaptive
> 	optimizers / momentum (Table 2), rather than analyzing one specific instance (SGD itself).
>
> (ii) Compressor generality (bias allowed). Our analysis requires only the contractive-error condition in
> 	Definition 1.1, which accommodates biased operators such as Top-$k$ sparsification without assuming unbiasedness or error-feedback behavior.
>
> (iii) Consensus--adaptivity--local-step coupling. The main technical challenge is to disentangle the intricate interactions among multiple local updates, gradient compression, adaptive gradient scaling, and decentralized mixing to explicitly characterize complexity dependence on the local-step count $K$, compression parameters, and network topology. Our final theoretical results further indicate that adaptive gradient scaling does not affect the order of complexity in theory, yet it significantly accelerates convergence in practical numerical experiments.
>
> We will strengthen the ``Compared to existing MLT+CC methods'' discussion in the Introduction/Related Work and
> add a short paragraph clarifying why the adaptive+MLT+CC coupling is nontrivial in decentralized settings.
>
> $\textbf{Weakness 2.}$
>
> Agreed. We will standardize terminology throughout: (i) Use "adaptive gradient updates" as the umbrella term. (ii) Reserve "Adam'' for the specific optimizer instance only.
>
> $\textbf{Weakness 3.}$
>
> We agree and will expand the baseline set. We added an additional baseline of CDProxSGT, the results of which can be viewed in our 'Additional Experiments' document at <https://anonymous.4open.science/r/TMLR2-38CE/reviewer_experiments.pdf>.
>
> $\textbf{Requested Change 1.}$
>
> We agree and will release the code.
>
> $\textbf{Requested Change 2.}$
>
> Agreed. Adam is one instance of adaptive methods. We will revise the title to use ``adaptive''.
>
> $\textbf{Requested Change 3.}$
>
> Thank you for pointing this out.  We will modify it.
>
> $\textbf{Requested Change 4.}$
>
> We agree this is informative, especially under MLT and compression. We will add plots/tables reporting consensus. Please see the 'Additional Experiments' document linked above for consensus error plots for all experiments reported in the original draft.

---

### Decision · Action_Editor_sVmQ · 2026-03-13

**Recommendation:** Accept with minor revision

**Additional Comments:**

The reviewers are mostly positive about the paper, nevertheless I encourage the authors to submit a revised version of their manuscript taking into account all their comments and the clarifications provided in the rebuttal.

In particular,
* Strengthening the comparison/related works of the framework + assumptions following Weak. 1 of Rev. KdrP and Rev. r63W
* Implement the additional experiments mentioned in the rebuttal and provide a link to the experiments code.
* Clarifications about AdaGrad and Communication matrix (Rev. eRb6 and KdrP)

**Audience:**

Yes

**Audience Explanation:**

Distributed optimization is an important topic when training large models or models with distributed data. It is therefore a important topic for the audience.

**Claims And Evidence:**

Yes

**Claims Explanation:**

The proposed decentralized optimization framework is soundly analyzed theoretically under reasonable assumptions and rather solid experiments validate the soundness of the approach.